# RESFL: An Uncertainty-Aware Framework for Responsible Federated Learning by Balancing Privacy, Fairness and Utility

**Dawood Wasif**[1], **Terrence J. Moore**[2], **Jin-Hee Cho**[1]
[1]Virginia Tech    [2]U.S. Army Research Laboratory
{dawoodwasif, jicho}@vt.edu    terrence.j.moore.civ@army.mil

## Abstract

Federated Learning (FL) has gained prominence in machine learning applications across critical domains, offering collaborative model training without centralized data aggregation. However, FL frameworks that protect privacy often sacrifice fairness and reliability; differential privacy reduces data leakage but hides sensitive attributes needed for bias correction, worsening performance gaps across demographic groups. This work explores the trade-off between privacy and fairness in FL-based object detection and introduces RESFL, an integrated solution optimizing both. RESFL incorporates *adversarial privacy disentanglement* and *uncertainty-guided fairness-aware aggregation*. The adversarial component uses a gradient reversal layer to remove sensitive attributes, reducing privacy risks while maintaining fairness. The uncertainty-aware aggregation employs an *evidential neural network* to weight client updates adaptively, prioritizing contributions with lower fairness disparities and higher confidence. This ensures robust and equitable FL model updates. We demonstrate the effectiveness of RESFL in high-stakes autonomous vehicle scenarios, where it achieves high mAP on FACET and CARLA, reduces membership-inference attack success by 37%, reduces equality-of-opportunity gap by 17% relative to the FedAvg baseline, and maintains superior adversarial robustness. However, RESFL is inherently domain-agnostic and thus applicable to a broad range of application domains beyond autonomous driving.

## 1. Introduction

Federated Learning (FL) has emerged as a promising solution to privacy concerns by enabling decentralized model training, ensuring data remains on local devices. In contrast, only model updates, such as gradients or weight deltas, are shared for aggregation. This paradigm not only reduces the risk of raw data exposure but also supports collaborative learning across heterogeneous and sensitive data silos in domains like healthcare, finance, and smart cities. However, the inherent obfuscation of sensitive attributes such as demographic labels or personal identifiers introduces a critical trade-off: fairness interventions often require direct access to these attributes to detect and correct biases. By withholding sensitive information in pursuit of privacy preservation, FL frameworks inadvertently hamper bias mitigation strategies, leading to disparate model performance across groups defined by age, gender, or ethnicity (Kaplan, 2024; Zhang et al., 2024).

The problem is exacerbated by external uncertainties in real-world data collection and inference, which undermine model confidence and reliability. In safety-critical applications, input data are affected by sensor noise, environmental variability (e.g., lighting, weather, occlusions), and domain shift between training and deployment. Unmodeled, these uncertainties can disproportionately degrade performance for subpopulations, amplifying disparities. For example, under foggy or low-light conditions, object detection models show higher false-negative rates for pedestrians with darker skin tones, compounding risks for vulnerable groups. Quantifying such epistemic uncertainty is therefore essential to ensure equitable reliability across demographic cohorts (Pathiraja et al., 2024). Beyond environmental shift, federated deployments also face adversarial clients that can poison up-

dates or manipulate confidence signals (Kumar et al., 2023), which can inflate group disparity and increasing privacy leakage risk.

While a growing body of work has advanced privacy-preserving techniques, such as differential privacy, secure multi-party computation, and homomorphic encryption, and fairness-aware methods, such as pre-processing transformations, in-processing regularizers, and post-processing adjustments, these solutions frequently optimize one objective at the expense of others. Differential privacy mechanisms can effectively limit membership inference and attribute leakage, but often degrade model utility and exacerbate fairness disparities by obscuring minority data patterns (Sun et al., 2021; Xin et al., 2020). Conversely, fairness-oriented re-weighting or regularization approaches can narrow demographic gaps but may inadvertently expose sensitive information if not carefully integrated. Centralized learning paradigms further magnify these tensions by aggregating unprotected data, while many federated solutions prioritize privacy guarantees without explicitly addressing equitable performance across groups (Chen et al., 2025; Ezzeldin et al., 2023; Yu et al., 2020). Post hoc fairness corrections or hard constraints also struggle to capture the nuanced interplay between privacy protection and bias mitigation in decentralized environments (Kim et al., 2024a).

To address these limitations, we propose RESFL, a domain-agnostic federated learning framework that jointly optimizes privacy and group fairness by integrating two complementary components within a single pipeline: (i) an adversarial representation module with gradient reversal that suppresses sensitive-attribute signals in shared representations, and (ii) an uncertainty-guided aggregation mechanism that leverages evidential uncertainty (via a scale-invariant uncertainty fairness metric (UFM)) to up-weight client updates exhibiting lower inter-group disparity and higher confidence. This unified design yields privacy-preserving, equitable, and reliable updates without sacrificing utility. We validate RESFL on autonomous vehicle (AV) scenarios to demonstrate its effectiveness in safety-critical and diverse environments. Empirically, RESFL delivers strong accuracy while reducing fairness gaps and privacy leakage, and remains robust under distribution shifts (weather variations). Across both FACET and CARLA, it consistently outperforms standard and state-of-the-art FL baselines on utility, fairness, and privacy.

## 2. RELATED WORK

**Federated Learning.** Federated Learning (FL) is a decentralized training paradigm where multiple clients collaboratively train a shared model while keeping data on-device. This mitigates privacy risks of centralized aggregation but introduces challenges, particularly data heterogeneity, as clients typically hold non-IID data (Yang et al., 2023). Early research focused on improving communication efficiency and convergence under heterogeneous (non-IID) client data (Li et al., 2019; Karimireddy et al., 2020; Martinez et al., 2020). However, FL introduces new challenges beyond optimization, including privacy leakage, performance disparities across participants, and fairness across sensitive demographic groups (Wasif et al., 2025).

**Privacy Preservation Techniques.** Preserving user privacy is a core objective in FL. A fundamental privacy–utility tradeoff holds: any mechanism that guarantees nontrivial privacy necessarily incurs some loss in utility (Dinur & Nissim, 2003). Differential Privacy (DP) quantifies this tradeoff by bounding the influence of any individual on the released updates or outputs (Dwork, 2006). Secure computation techniques such as homomorphic encryption (HE) and secure multi-party computation (SMC) protect data during computation and transport, but by themselves do not limit statistical inference from the outputs (Chen et al., 2023; Xu et al., 2021); they are orthogonal to DP and often combined with it for end-to-end protection (Yi et al., 2014; Tran et al., 2023). Recent work also studies perturbation and shuffling for privacy amplification and communication efficiency (Erlingsson et al., 2019; Chen et al., 2024a; Kim et al., 2024b). However, most privacy solutions in FL neglect fairness, risking inequitable outcomes despite strong privacy protections.

**Fairness in Federated Learning.** Fairness in machine learning has been extensively explored in centralized settings, with numerous methods to mitigate biases against underrepresented groups (Hardt et al., 2016; Kairouz et al., 2021; Mehrabi et al., 2021). In FL, researchers distinguish between *client fairness*, which aims for uniform model performance across data-silo clients (Yu et al., 2020; Karimireddy et al., 2020), and *group fairness*, which seeks equitable outcomes across sensitive demographic cohorts despite decentralized data (Kairouz et al., 2021). Traditional fairness

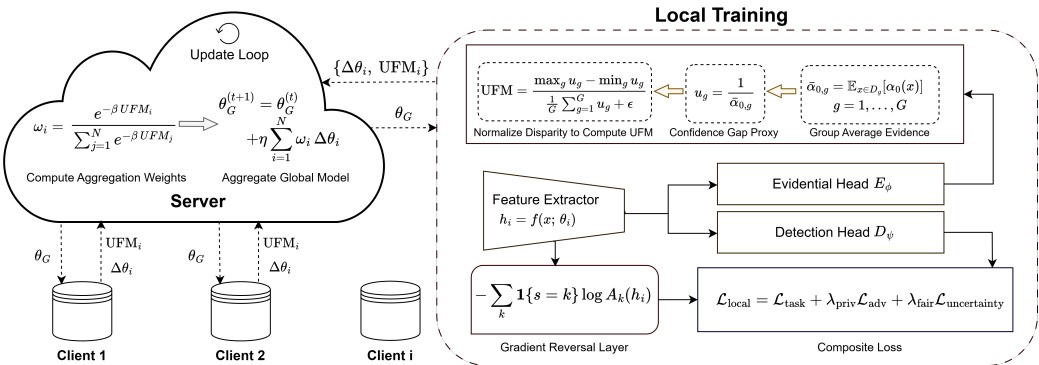

Figure 1: Overview of the `RESFL` framework. *Left:* Server receives $\{\Delta\theta_i, \mathrm{UFM}_i\}$, computes aggregation weights $\omega_i \propto \exp(-\beta\,\mathrm{UFM}_i)$, and updates the global model $\theta_G$. *Right:* Client $i$ computes feature representation, applies a gradient reversal layer for adversarial privacy loss $\mathcal{L}_{\mathrm{adv}}$, and an evidential head for uncertainty-fairness metric $\mathrm{UFM}_i$, then forms the composite loss $\mathcal{L}_{\mathrm{local}}$ to produce update $\Delta\theta_i$.

strategies such as constrained optimization or regularization work well in centralized frameworks (Wu et al., 2018) but falter in FL, since the server lacks direct access to sensitive attributes for bias measurement (McMahan et al., 2017), and client-level equalization does not guarantee demographic parity, underscoring the need for FL-specific fairness mechanisms.

**Privacy-preserving & Fair FL.** Given the inherent tension between privacy and fairness, recent research has explored joint approaches to address both in FL. Differentially private algorithms can worsen fairness disparities by masking minority-group patterns (Zhang et al., 2021), while fairness-aware techniques may increase privacy risks by exposing sensitive attributes. Prior work includes FairDP-SGD and FairPATE for centralized settings (Yaghini et al., 2023), FPFL which enforces group fairness under DP guarantees but at high communication cost (Rodríguez-Gálvez et al., 2021), and two-step schemes that align a privacy-protected model with a fair proxy (Sun et al., 2023; Pujol et al., 2020). Pre- and post-processing defenses like (Pentyala et al., 2022) and (Corbucci et al., 2024) also integrate privacy and fairness but often incur computational overhead or limited scalability (Imteaj et al., 2021). Concurrently, FL methods use epistemic or evidential uncertainty for calibration, reliability, and client personalization, notably in medical imaging e.g., (Chen et al., 2024b; Hendrix et al., 2024; Chen et al., 2024c). Despite these advances, many approaches struggle to scale or balance the trade-offs effectively, motivating our unified `RESFL` framework. Moreover, most FL studies evaluate under benign clients and do not test resilience to adversarial updates or evidence manipulation (Kairouz et al., 2021), leaving the stability of their guarantees unclear.

Building on these insights, our proposed `RESFL` framework overcomes these limitations by integrating privacy preservation and *group fairness* optimization into a single, end-to-end FL algorithm and explicitly evaluates under both benign and adversarial settings.

## 3. METHODOLOGY

This section introduces our integrated privacy-preserving and fairness-aware Federated Learning framework, *responsible FL* (`RESFL`). Our approach tackles two key challenges: (i) preventing sensitive attribute leakage during training to ensure privacy and (ii) mitigating bias in client updates to ensure *group fairness*. To achieve this, we integrate adversarial privacy disentanglement with uncertainty-guided fairness-aware aggregation using an evidential neural network (ENN), enabling the estimates of epistemic uncertainty (Amini et al., 2020). The flow of the `RESFL` algorithm is depicted in Figure 1.

### 3.1. UNCERTAINTY FAIRNESS METRIC (UFM) FOR GROUP-FAIR AGGREGATION

**Evidential Uncertainty Modeling.** In RESFL, each client replaces its standard softmax detection head with an evidential output layer that predicts a nonnegative concentration vector $\boldsymbol{\alpha} = (\alpha_1, \ldots, \alpha_C)$ for $C$ object classes. These concentration parameters parameterize a Dirichlet distribution over the categorical probability simplex, allowing closed-form computation of epistemic uncertainty without resorting to costly Monte Carlo sampling or deep ensembles. Formally, for each input $x$, the evidential head produces

$$p(\mathbf{p} \mid \boldsymbol{\alpha}) = \frac{\Gamma\left(\sum_{c=1}^{C} \alpha_c\right)}{\prod_{c=1}^{C} \Gamma(\alpha_c)} \prod_{c=1}^{C} p_c^{\alpha_c - 1}, \tag{1}$$

where $C$ is the number of considered classes, $\Gamma(\cdot)$ denotes the Gamma function, and $\mathbf{p} = (p_1, \ldots, p_C)$ represents the class probabilities. The total evidence $\alpha_0 = \sum_{c=1}^{C} \alpha_c$ directly yields an analytic estimate of the approximate epistemic variance (Sensoy et al., 2018),

$$\sigma_{\text{epi},c}^2 = \mathbb{E}[p_c]\big(1 - \mathbb{E}[p_c]\big) \cdot \frac{1}{\alpha_0 + 1} = \frac{\alpha_c}{\alpha_0}\left(1 - \frac{\alpha_c}{\alpha_0}\right) \cdot \frac{1}{\alpha_0 + 1} \sim \frac{1}{\alpha_0}, \tag{2}$$

where the first two equalities give the exact Dirichlet predictive variance and the final "$\sim$" indicates asymptotic scaling, i.e., $\text{Var}[p_c] = \Theta(1/\alpha_0)$. This faithfully reflects model confidence: higher $\alpha_0$ implies lower epistemic uncertainty. Raw logits $z_c$ are passed through a softplus-plus-one bias, $\alpha_c = 1 + \text{softplus}(z_c)$, to ensure strict positivity and numerical stability. Training uses a composite Dirichlet negative log-likelihood augmented with a regularization term that penalizes overconfident errors, thereby *encouraging* calibrated uncertainty estimates in practice. At inference, each client computes epistemic variances in a single forward pass, enabling efficient uncertainty assessment on edge devices. This evidential formulation, integrated into the detection pipeline, provides a principled mechanism for quantifying per-detection confidence under data heterogeneity and environmental variability.

UFM is computed solely from the *classification* Dirichlet evidence. For an image $x$, let $\mathcal{P}_\tau(x)$ be post-NMS detections above a fixed score threshold $\tau$. We define the per-image average total evidence (set to 0 if $|\mathcal{P}_\tau(x)| = 0$) and then the group-wise mean:

$$\bar{\alpha}_{0,g} = \mathbb{E}_{x \in \mathcal{D}_g} \left[ \frac{1}{\max(1, |\mathcal{P}_\tau(x)|)} \sum_{d \in \mathcal{P}_\tau(x)} \alpha_0^{(d)} \right]. \tag{3}$$

Here, $\mathcal{D}_g$ is the set of images that contain at least one ground-truth person instance of group $g$; the sum runs over post-NMS detections in $x$, and only detections matched to group-$g$ instances contribute.

**Group-Level Disparity Quantification.** Using $\{\bar{\alpha}_{0,g}\}_{g=1}^{G}$ from Eq. (3), define the inter-group uncertainty gap and the normalized Uncertainty Fairness Metric (UFM) as

$$\Delta_u = \max_g \left( \frac{1}{\bar{\alpha}_{0,g}} \right) - \min_g \left( \frac{1}{\bar{\alpha}_{0,g}} \right), \qquad \text{UFM} = \frac{\Delta_u}{\frac{1}{G} \sum_{g=1}^{G} \frac{1}{\bar{\alpha}_{0,g}} + \epsilon}, \tag{4}$$

with $\epsilon > 0$ for numerical stability. Higher UFM indicates greater disparity; lower UFM indicates better group fairness. See Appendix A.3 for detection-head details.

We formalize UFM as a scale-invariant measure of inter-group epistemic disparity and show (under bounded loss and standard evidential assumptions) that controlling it tightens confidence-adjusted group generalization terms (Appendix B, Thm. B.1, Cor. B.2). We then lift this to federated evaluation via a mixture argument (Lemma B.3) and bound the global disparity by an aggregation-weighted combination of client UFMs (Prop. B.4); with UFM-weighted aggregation $\omega_i \propto \exp(-\beta\, \text{UFM}_i)$, the surrogate bound decreases (Cor. B.5), which in turn tightens explicit bounds on downstream group-parity gap functionals $\mathcal{F}$ evaluated at the fixed fairness IoU $\tau_{\text{fair}}$. Under non-degenerate group coverage and standard client sampling, smaller $\beta$ behaves like uniform averaging, while larger $\beta$ emphasizes clients with tighter per-group disparities, linking the theory to practice and motivating UFM-guided aggregation for global fairness.

**Aggregation Weighting Mechanism.** On the server side, we aggregate client updates using a fairness-aware weighting scheme that dynamically adjusts to reported UFM values. Given each client $i$'s update $\Delta\theta_i$ and corresponding $\text{UFM}_i$ (see Figure 1), we assign weights via a temperature-scaled exponential:

$$\omega_i = \frac{\exp\big(-\beta\,\text{UFM}_i\big)}{\sum_{j=1}^{N} \exp\big(-\beta\,\text{UFM}_j\big)}\,, \tag{5}$$

where $\beta > 0$ controls the sharpness of fairness prioritization. As $\beta \to 0$, weights approach uniform averaging, while larger $\beta$ concentrates updates on clients with minimal uncertainty disparity. The global model is then updated by:

$$\theta_G^{(t+1)} = \theta_G^{(t)} + \eta \sum_{i=1}^{N} \omega_i\,\Delta\theta_i, \tag{6}$$

ensuring that contributions from clients exhibiting both high confidence and equitable performance are amplified. This continuous reweighting adapts to temporal shifts and data heterogeneity, promoting robust convergence with reduced fairness gaps and preserved accuracy. We gate Eq. (5) with a deterministic confidence check: if a client's validation accuracy is below a fixed floor, we set its reported fairness statistic $u_i \leftarrow b$ (the clipped worst value), forcing $\omega_i \approx 0$; this blocks uniformly low-confidence clients from receiving high weight and limits the influence of poisoned or low-confidence updates.

## 3.2. ADVERSARIAL PRIVACY DISENTANGLEMENT VIA GRADIENT REVERSAL

To mitigate sensitive attribute leakage during federated training, we augment the feature extractor $f(x;\theta) : \mathcal{X} \to \mathbb{R}^d$, which maps input data to a latent representation $h$, with an adversarial classifier $A(h;\phi) : \mathbb{R}^d \to [0,1]^K$. The adversary is trained to predict the sensitive attribute label $s \in \{1,\ldots,K\}$ from $h$, while the feature extractor is jointly optimized to make this prediction as difficult as possible, thus encouraging the learned representation to be invariant to $s$. During training, the classifier parameters $\phi$ seek to minimize the cross-entropy over the joint distribution of inputs and labels, while the feature extractor parameters $\theta$ are trained to maximize this same objective, thereby removing attribute-relevant signals. Concretely, we embed a Gradient Reversal Layer (GRL) $\mathcal{R}_{\lambda_{\text{adv}}}$ between $f$ and $A$, which acts as the identity in the forward pass but multiplies incoming gradients by $-\lambda_{\text{adv}}$ in the backward pass. The resulting adversarial minimax objective is expressed as:

$$\min_{\theta} \max_{\phi} \mathbb{E}_{(x,s)\sim\mathcal{D}_i}\Big[ -\lambda_{\text{adv}} \sum_{k=1}^{K} \mathbf{1}\{s=k\} \log A_k\big(\mathcal{R}_{\lambda_{\text{adv}}}(f(x;\theta)); \phi\big)\Big], \tag{7}$$

where $\mathcal{D}_i$ denotes the local dataset of client $i$ and $\mathbf{1}\{s=k\}$ is the indicator for class $k$ and $A_k(\cdot)$ denotes the $k$-th output probability. While we do not claim $(\varepsilon,\delta)$–DP guarantees, our objective has an information-theoretic interpretation: letting $H = f(X;\theta)$ denote the learned representation and $S$ the sensitive attribute, maximizing $\mathcal{L}_{\text{adv}}$ reduces the mutual information $I(H;S)$; by Fano's inequality, as $I(H;S) \to 0$ any attribute-inference attack $\hat{S} = g(H)$ is driven to chance level, i.e., accuracy $\approx 1 - \frac{1}{K}$ (Appendix C).

Once the adversarial classifier is optimally trained for a fixed feature extractor, we obtain the induced privacy loss for $\theta$ by substituting the worst-case classifier parameters $\phi^*(\theta) = \arg\max_\phi \mathcal{L}_{\text{adv}}(\theta,\phi)$. The privacy-preserving gradient step for the feature extractor is then driven by:

$$\mathcal{L}_{\text{priv}}(\theta) = \lambda_{\text{adv}} \mathbb{E}_{(x,s)\sim\mathcal{D}_i}\Big[\sum_{k=1}^{K} \mathbf{1}\{s=k\} \log A_k\big(f(x;\theta); \phi^*(\theta)\big)\Big], \tag{8}$$

which, when differentiated through the GRL, enforces that feature representations $h$ become invariant to $s$. In practice, we interleave updates of $\phi$ (maximization) and $\theta$ (minimization) within each local SGD step, ensuring that the learned representation provably suppresses sensitive attribute information while retaining utility for the primary detection task.

Table 1: Comparison of FL algorithms on the FACET dataset in detection performance (mAP), fairness ($|1-\text{DI}|$, $\Delta$EOP), privacy (MIA, AIA SR), and robustness (BA AD, DPA EODD). Arrows in headers indicate whether higher ($\uparrow$) or lower ($\downarrow$) values are better. Results are mean$_{\pm\text{std}}$ over 3 seeds.

| Algorithm | Utility | Fairness | | Privacy Attacks | | Robustness Attacks | |
|---|---|---|---|---|---|---|---|
| | Overall mAP ($\uparrow$) | $|1-\text{DI}|$ ($\downarrow$) | $\Delta$EOP ($\downarrow$) | MIA SR ($\downarrow$) | AIA SR ($\downarrow$) | BA AD ($\downarrow$) | DPA EODD ($\downarrow$) |
| FedAvg | $0.6378_{\pm 0.006}$ | $\mathbf{0.2159_{\pm 0.006}}$ | $0.2362_{\pm 0.007}$ | $0.3341_{\pm 0.010}$ | $0.4431_{\pm 0.011}$ | $0.3125_{\pm 0.009}$ | $0.0792_{\pm 0.005}$ |
| FedAvg-DP ($\epsilon = 1$) | $0.4612_{\pm 0.008}$ | $0.3945_{\pm 0.003}$ | $0.2879_{\pm 0.009}$ | $\mathbf{0.2364_{\pm 0.011}}$ | $0.2627_{\pm 0.006}$ | $0.3097_{\pm 0.005}$ | $0.1396_{\pm 0.008}$ |
| FairFed | $\mathbf{0.7013_{\pm 0.006}}$ | $0.2496_{\pm 0.006}$ | $0.2562_{\pm 0.007}$ | $0.4409_{\pm 0.012}$ | $0.5256_{\pm 0.012}$ | $0.4139_{\pm 0.013}$ | $\mathbf{0.0566_{\pm 0.004}}$ |
| PrivFairFl-Pre | $0.6154_{\pm 0.006}$ | $0.2504_{\pm 0.006}$ | $0.2659_{\pm 0.007}$ | $0.3875_{\pm 0.010}$ | $0.4038_{\pm 0.010}$ | $0.3238_{\pm 0.009}$ | $0.0953_{\pm 0.006}$ |
| PrivFairFl-Post | $0.6119_{\pm 0.006}$ | $0.2718_{\pm 0.006}$ | $0.2505_{\pm 0.006}$ | $0.2872_{\pm 0.009}$ | $0.3159_{\pm 0.009}$ | $0.3212_{\pm 0.009}$ | $0.0937_{\pm 0.006}$ |
| PUFFLE | $0.4192_{\pm 0.008}$ | $0.3721_{\pm 0.007}$ | $0.2976_{\pm 0.007}$ | $0.2725_{\pm 0.009}$ | $0.2909_{\pm 0.009}$ | $\mathbf{0.1439_{\pm 0.008}}$ | $0.1360_{\pm 0.008}$ |
| PFU-FL | $0.3952_{\pm 0.009}$ | $0.3356_{\pm 0.007}$ | $0.3446_{\pm 0.008}$ | $0.2409_{\pm 0.009}$ | $\mathbf{0.2546_{\pm 0.009}}$ | $0.2612_{\pm 0.009}$ | $0.1459_{\pm 0.008}$ |
| **Ours (RESFL)** | $\mathbf{0.6654_{\pm 0.005}}$ | $\mathbf{0.2287_{\pm 0.005}}$ | $\mathbf{0.1959_{\pm 0.006}}$ | $\mathbf{0.2093_{\pm 0.006}}$ | $\mathbf{0.1832_{\pm 0.005}}$ | $0.1692_{\pm 0.007}$ | $\mathbf{0.0674_{\pm 0.004}}$ |

### 3.3. JOINT OPTIMIZATION OF PRIVACY AND FAIRNESS

In each client's training loop, RESFL minimizes a composite loss that balances detection accuracy, attribute obfuscation, and uncertainty-based bias control. Formally, each client solves:

$$\mathcal{L}_{\text{local}}(\theta, \phi) = \mathcal{L}_{\text{task}}(\theta) + \lambda_{\text{priv}} \, \mathcal{L}_{\text{adv}}(\theta, \phi) + \lambda_{\text{fair}} \, \mathcal{L}_{\text{uncertainty}}(\theta), \qquad (9)$$

where $\lambda_{\text{priv}}$ scales the gradient reversal adversarial loss to limit information leakage, and $\lambda_{\text{fair}}$ weights the evidential uncertainty term to reduce group disparity. By selecting $(\lambda_{\text{priv}}, \lambda_{\text{fair}})$ along the convex envelope of evaluated tradeoff points, practitioners obtain models that meet target privacy and fairness thresholds without unnecessary sacrifice of either objective.

After local updates, each client computes its $\text{UFM}_i$ and sends both the parameter update $\Delta\theta_i$ and $\text{UFM}_i$ to the server. The server then aggregates via:

$$\theta_G^{(t+1)} = \theta_G^{(t)} + \eta \sum_{i=1}^{N} \frac{\exp\left(-\beta \, \text{UFM}_i\right)}{\sum_{j=1}^{N} \exp\left(-\beta \, \text{UFM}_j\right)} \, \Delta\theta_i, \qquad (10)$$

where $\beta$ controls the fairness weight. This alternating sequence of local composite-loss minimization and fairness-aware aggregation (Algorithm 1 in Appendix D) drives the global model to converge with robust detection, provable attribute privacy, and equitable treatment across sensitive groups.

## 4. EXPERIMENTAL RESULTS & ANALYSES

We evaluate RESFL in an autonomous vehicle (AV) context using the FACET dataset and CARLA simulator to capture demographic variation and environmental perturbations. Given the safety-critical and privacy-sensitive nature of AV perception, we pose the following key questions: (1) *To what extent does RESFL balance utility, privacy, and fairness under standard AV operating conditions? (2) How resilient is RESFL to increased uncertainty caused by environmental variations such as changing weather and lighting?*

### 4.1. EXPERIMENTAL SETUP

**Datasets.** The FACET benchmark (Gustafson et al., 2023) comprises 32,000 real-world images with over 50,000 person instances annotated for perceived skin tone on the ten-level Monk Skin Tone (MST) scale (MST = 1 lightest to MST = 10 darkest, see Figure 5 in Appendix E); we average multiple annotations per instance, discretize back to the ten MST levels, partition into ten cohorts, and split into four IID client shards to simulate cross-device heterogeneity and demographic variation without sharing raw data. Using the CARLA simulator (Dosovitskiy et al., 2017), we collect 6,000 clear-weather frames (600 per MST level) for fine-tuning and 7,800 evaluation frames across three urban layouts (Town01, Town03, and Town05) under clear, foggy, and rainy conditions at five intensities (0%, 25%, 50%, 75%, 100%). Each walker blueprint available in CARLA is manually assigned to a corresponding MST label through visual inspection based on appearance and attributes. Pedestrian bounding boxes are extracted via connected-component analysis on semantic segmentation masks, which serves as our ground truth (see Appendix E for details).

**Comparing Schemes.** We benchmark `RESFL` against standard and state-of-the-art federated learning methods. **FedAvg** serves as the canonical baseline, performing weighted averaging of client updates. We evaluate **FedAvg-DP (per-example DP-SGD)**, which adds calibrated Gaussian noise to per-example clipped local gradients under a fixed privacy budget ($\epsilon = 1$). **FairFed** (Ezzeldin et al., 2023) dynamically adjusts aggregation weights to penalize cross-client performance gaps. **PrivFairFL** (Pentyala et al., 2022) incorporates fairness constraints either before aggregation (**PrivFairFL-Pre**) or after local updates (**PrivFairFL-Post**). **PUFFLE** (Corbucci et al., 2024) unifies noise injection with fairness regularization in a joint optimization framework. Finally, **PFU-FL** (Sun et al., 2023) employs adaptive weighting to balance privacy, fairness, and utility objectives.

**Metrics.** We measure detection accuracy using mean Average Precision (mAP), which averages per-class AP to capture both object localization and classification quality. For fairness, we report the absolute disparate impact deviation $|1 - \mathrm{DI}|$, quantifying the ratio of favorable outcome rates between the most- and least-advantaged groups, and the equality of opportunity gap $\Delta\mathrm{EOP}$, the absolute difference in true positive rates across cohorts. Privacy is assessed by Membership Inference Attack Success Rate (MIA SR) and Attribute Inference Attack Success Rate (AIA SR), where lower values indicate stronger confidentiality. Robustness is measured by Byzantine Accuracy Degradation (BA AD), the relative per-condition mAP drop between clean and attacked runs, and Data Poisoning Attack Equalized Odds Difference Deviation (DPA EODD), the rise in fairness disparity, under a fixed protocol with a constant Byzantine-client fraction each round (sign-flip, $\ell_2$-bounded) and a constant poisoning fraction over a specified block of local epochs.

**Experimental Configuration.** We implement `RESFL` using a modified YOLOv8 backbone with an evidential concentration-vector head. We run a standard federated setup with $K = 4$ clients and $T = 100$ communication rounds; in each round every client trains for one local epoch (batch size 64) using SGD (momentum 0.9, weight decay $1\mathrm{e}^{-4}$) with an initial learning rate of 0.001, decayed by 0.1 at epochs 50 and 75. The FACET dataset (32k images) is split into four equal i.i.d. subsets, and within each client shard we keep a fixed 90/10 train/validation split, using the validation portion only to compute per-client $\mathrm{UFM}_i$ and apply the mAP@50–95 floor of 0.30 in the aggregation gate. CARLA fine-tuning uses 6k neutral-weather frames and evaluates on 7.8k frames across 13 weather conditions. We set $\lambda_{\mathrm{priv}} = 0.1$, $\lambda_{\mathrm{fair}} = 0.01$, and aggregation temperature $\beta = 2.0$, and average results over three random seeds. All hyperparameters were selected via an extensive grid search (more details in Appendix F and G).

## 4.2. TRADE-OFF ANALYSIS ON THE FACET DATASET

In this experiment, we evaluate the performance of various FL algorithms on the FACET dataset. Our objective is to compare the overall trade-offs of each method in a controlled setting. Table 1 reports results for baselines in Section 4.1, and our proposed `RESFL`, while Figure 6 illustrates per-skin-tone mAP distributions. `RESFL` attains 0.6654 mAP, close to FairFed (0.7013), and exceeds PUFFLE and PFU-FL. It maintains consistent accuracy across all ten MST cohorts via uncertainty-guided weighting. It yields $|1 - \mathrm{DI}| = 0.2287$ and $\Delta\mathrm{EOP} = 0.1959$, improves privacy (MIA 0.2093, AIA 0.1832) over FedAvg and its DP variants, and remains robust under attacks (BA AD 0.1692; DPA EODD 0.0674). See Appendix H for IID vs. non-IID results with 4 and 8 clients, where `RESFL` maintains strong accuracy with the best fairness–privacy profile. We also note its linear cost and practical scalability in Appendix F.

## 4.3. RESILIENCE ANALYSIS UNDER ADVERSE CONDITIONS IN CARLA

We fine-tune best performing FACET baselines, FedAvg-DP ($\epsilon$=1), FairFed, PUFFLE, and `RESFL`, on 6,000 clear-weather frames and compare on on 7,800 adverse-weather frames under clear, cloud, rain, and fog at five intensities (0–100%) each. Figure 2 reports utility (mAP, higher is better), fairness (mean of $|1-\mathrm{DI}|$ and $\Delta\mathrm{EOP}$, lower is better), and privacy risk (mean MIA/AIA success rate, lower is better). Under clear weather (0%), `RESFL` achieves 0.46 mAP, 0.24 fairness, and 0.17 privacy. As cloud cover increases, contrast and illumination vary but scene geometry remains visible; `RESFL` degrades gently to 0.39 mAP at 100% cloud (vs. 0.22 FedAvg-DP and 0.34 FedAvg), with fairness rising to 0.28 (vs. 0.40 FedAvg-DP) and privacy to 0.20 (vs. 0.38 FedAvg). Rain adds dynamic streaks and partial occlusions; despite this, `RESFL` retains 0.42 mAP, 0.25 fairness, and 0.20 privacy at 100% rain. These trends indicate that uncertainty-guided aggregation with adversar-

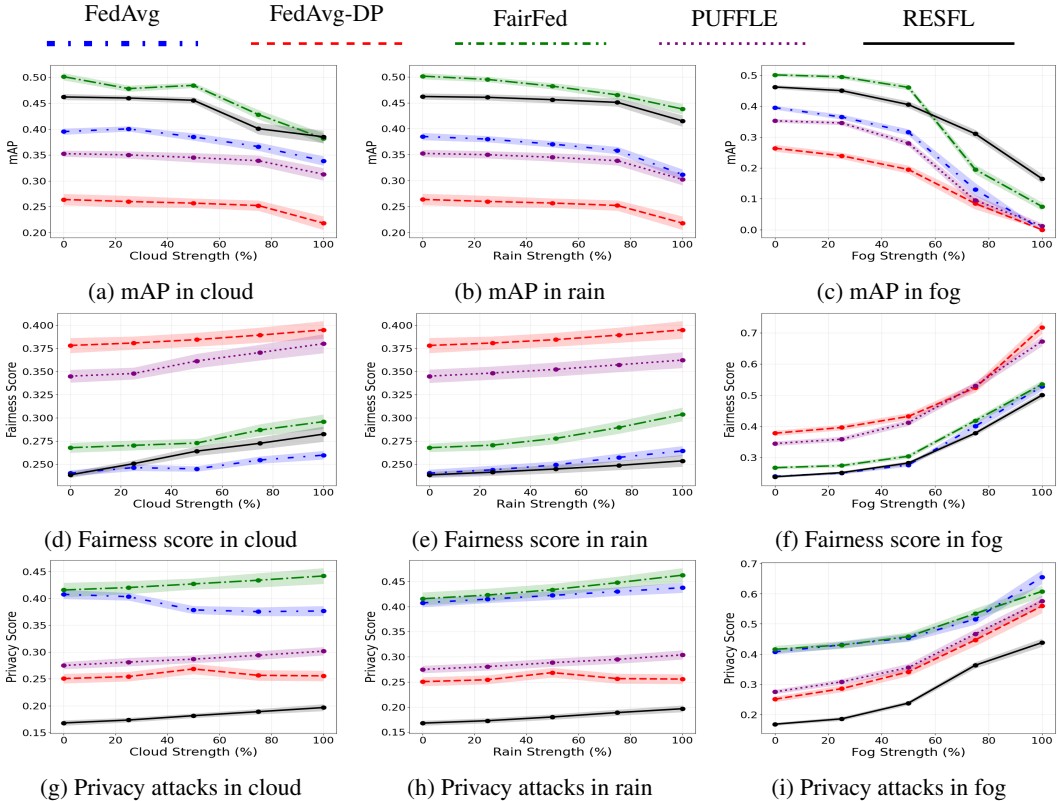

Figure 2: Performance comparison of four state-of-the-art FL methods and RESFL across three weather conditions (cloud, rain, fog) at 0%–100% intensity. Curves show the mean $\pm$ std over three seeds: accuracy is mAP (**higher is better**); fairness score averages $|1 - \mathrm{DI}|$ and $\Delta\mathrm{EOP}$ (**lower is better**) to capture inter-group disparity; privacy score averages MIA and AIA success rates (**lower is better**). RESFL sustains strong mAP while keeping fairness gaps and privacy attack rates low under harsher weather.

ial disentanglement retains discriminative evidence while damping group-specific overconfidence, which helps contain both disparity and attackability in cloud and rain.

Fog is the hardest setting because it hides edges and distant objects, so the useful signal drops sharply. The loss of visibility is uneven across scenes and groups (e.g., small or dark objects disappear first, see Figure 3, which raises $|1-\mathrm{DI}|$ and $\Delta\mathrm{EOP}$ as fog gets denser. With fewer real cues, models fall back on shortcuts and memorized patterns: training images stay relatively more confident than unseen ones, and attribute-related proxies become stronger, so membership/attribute attacks succeed more often. At 100% fog, RESFL keeps 0.17 mAP, 0.50 fairness, and 0.44 privacy, while others drop below 0.10 mAP with fairness/privacy $> 0.50$. This gap reflects the physical limit of severe fog, but RESFL still slows the increase in disparity and attack success through evidence flooring, temperature control, and vacuity masking (Appendix I).

### 4.4. ABLATION STUDY WITH RESFL

We conduct an ablation study to examine the impact of two key hyperparameters in RESFL: the uncertainty-based fairness coefficient ($\lambda_{\mathrm{fair}}$) and the adversarial privacy coefficient ($\lambda_{\mathrm{priv}}$), while keeping the task loss at one. Fixing privacy and sweeping $\lambda_{\mathrm{fair}}$ (first block of Table 2), a lower fairness weight can yield higher mAP. This occurs because increasing $\lambda_{\mathrm{fair}}$ reallocates model capacity toward high-uncertainty or minority slices and applies evidence flooring with group normalization, which redirects gradient mass away from majority slices that dominate average precision and tempers overconfident detections on easy cases, introducing a bias–variance trade-off under finite capacity. Consequently, increasing $\lambda_{\mathrm{fair}}$ lowers $|1 - \mathrm{DI}|$ and $\Delta\mathrm{EOP}$ (both better), but reduces average mAP once fairness pressure pulls learning away from majority segments.

Table 2: RESFL performance under varying uncertainty-based fairness ($\lambda_{\text{fair}}$) and adversarial privacy ($\lambda_{\text{priv}}$) coefficients. Utility is measured as mAP (↑), fairness via $|1 - \text{DI}|$ and $\Delta\text{EOP}$ (↓), and privacy via MIA/AIA success rates (↓). Results are mean ± std over 3 seeds.

| Coefficients | | Utility (↑) | Fairness (↓) | | Privacy (↓) | |
| --- | --- | --- | --- | --- | --- | --- |
| $\lambda_{\text{fair}}$ | $\lambda_{\text{priv}}$ | mAP | $|1 - \text{DI}|$ | $\Delta\text{EOP}$ | MIA SR | AIA SR |
| 1 | 0 | $0.6278_{\pm 0.006}$ | $\mathbf{0.2258}_{\pm\mathbf{0.005}}$ | $0.2062_{\pm 0.007}$ | $0.3341_{\pm 0.011}$ | $\mathbf{0.1431}_{\pm\mathbf{0.006}}$ |
| 0 | 1 | $0.5856_{\pm 0.008}$ | $0.2571_{\pm 0.006}$ | $0.2846_{\pm 0.008}$ | $\mathbf{0.1025}_{\pm\mathbf{0.011}}$ | $0.1463_{\pm 0.006}$ |
| 0.01 | 1 | $0.6056_{\pm 0.007}$ | $0.2653_{\pm 0.007}$ | $0.3459_{\pm 0.010}$ | $0.1256_{\pm 0.012}$ | $0.1668_{\pm 0.002}$ |
| 0.1 | 1 | $0.6254_{\pm 0.006}$ | $0.2538_{\pm 0.006}$ | $0.2626_{\pm 0.007}$ | $0.1477_{\pm 0.003}$ | $0.1608_{\pm 0.007}$ |
| 1 | 1 | $0.5953_{\pm 0.009}$ | $0.2432_{\pm 0.006}$ | $0.2513_{\pm 0.007}$ | $0.2197_{\pm 0.008}$ | $0.1782_{\pm 0.008}$ |
| 0.1 | 0.01 | $\mathbf{0.6654}_{\pm\mathbf{0.005}}$ | $0.2287_{\pm 0.005}$ | $\mathbf{0.1959}_{\pm\mathbf{0.006}}$ | $0.2093_{\pm 0.006}$ | $0.1832_{\pm 0.005}$ |
| 0.1 | 0.1 | $0.6430_{\pm 0.006}$ | $0.2625_{\pm 0.007}$ | $0.3143_{\pm 0.002}$ | $0.1363_{\pm 0.005}$ | $0.1474_{\pm 0.001}$ |
| 0.1 | 1 | $0.5839_{\pm 0.010}$ | $0.3862_{\pm 0.008}$ | $0.4146_{\pm 0.014}$ | $0.1176_{\pm 0.005}$ | $0.1656_{\pm 0.007}$ |

Holding $\lambda_{\text{fair}} = 0.1$ and varying $\lambda_{\text{priv}}$ (second block of Table 2), moderate privacy pressure reduces memorization and stabilizes uncertainty without erasing discriminative cues, whereas excessive privacy weight pushes features toward overly invariant representations that hurt both accuracy and fairness. The best balance occurs at $\lambda_{\text{fair}} = 0.1$, $\lambda_{\text{priv}} = 0.01$, achieving mAP 0.6654, $|1 - \text{DI}| = 0.2287$, $\Delta\text{EOP} = 0.1959$, MIA 0.2093, and AIA 0.1832 (all mean ± std in Table 2); increasing $\lambda_{\text{priv}}$ beyond 0.01 degrades both detection and group parity by over-suppressing informative, but privacy-sensitive, signals.

### 4.5. Impact of Sensitive Attributes and Cross-Domain Generalization

We only use sensitive attributes for local loss computation and fairness estimation and are never transmitted outside each client. This setting matches regulated domains where client-side demographic labels are routinely stored under consent, such as hospital consortia (race/age), automotive or mobility fleets (driver profiles), and enterprise datasets with gender or ethnicity for compliance reporting. In all such cases, attribute information remains local, and RESFL operates entirely within each silo without additional disclosure.

To study cases where explicit attributes are unavailable, we also train RESFL using the attribute-free UFM (AF-UFM) variants described in Appendix J.2. Label-free training preserves over 90% of the fairness and privacy gains obtained with labeled cohorts, with less than a two-percent drop in utility across all datasets. On the **Adult** and **TweetEval** benchmarks, RESFL and its AF-UFM variants maintain comparable accuracy while achieving markedly lower fairness and privacy scores than all baselines (Table 16). These results show that uncertainty-guided aggregation captures latent heterogeneity even without explicit demographic supervision and that the proposed mechanisms extend beyond visual perception to structured tabular and textual modalities, supporting RESFL as a domain-agnostic route to privacy-aware and fair federated optimization.

## 5. Conclusions

This work introduced RESFL, a unified framework for responsible federated learning that jointly enhances utility, fairness, and parameter privacy under realistic adversarial and environmental conditions. The framework integrates adversarial privacy disentanglement through a gradient reversal mechanism with an evidential uncertainty head that estimates calibrated epistemic uncertainty. A scale-invariant Uncertainty Fairness Metric (UFM) further guides aggregation by weighting clients with lower inter-group uncertainty disparity, yielding updates that are both confident and equitable. Experiments across visual (FACET, CARLA) and non-visual (Adult, TweetEval) domains show that RESFL sustains strong predictive utility while substantially reducing fairness gaps and privacy leakage, confirming its potential as a domain-agnostic foundation for responsible federated optimization.

**Limitations and Future Work.** While RESFL already demonstrates linear scalability across clients, its joint optimization of backbone, adversarial, and evidential modules may still constrain

deployment on highly resource-limited devices. Future work will focus on optimizing these components for lightweight participation and adaptive scheduling across heterogeneous clients. Extensions will refine UFM through vacuity–dissonance decomposition to yield interpretable fairness signals and adaptive temperature schedules that self-balance privacy and fairness. We also plan to strengthen trust by incorporating median-based or consistency-checked UFM reporting for robustness against falsified fairness inputs, evaluate `RESFL` in multimodal and streaming federations, and integrate secure aggregation with calibrated differential privacy to deliver end-to-end verifiable responsible learning.

## ETHICS STATEMENT

This work addresses fairness and privacy in federated learning, with experiments involving demographic attributes such as perceived skin tone. The FACET dataset used in our experiments contains annotations of perceived skin tone collected under informed consent and made publicly available for fairness research. We do not collect any new personal data. Sensitive attributes in our framework are used solely for local fairness estimation and are never transmitted outside each client. While `RESFL` is designed to reduce demographic bias and privacy leakage, we acknowledge that uncertainty-based fairness proxies may not perfectly capture all forms of bias. Deployment in safety-critical systems such as autonomous vehicles should involve domain-specific auditing and regulatory oversight. We release all code and model checkpoints to support transparent evaluation and reproducibility.

## REPRODUCIBILITY STATEMENT

We have taken the following steps to ensure reproducibility. Full implementation details, including hyperparameters, dataset splits, training schedules, and attack protocols, are provided in Appendices F and G. All experiments are averaged over three random seeds with standard deviations reported. The FACET dataset (Gustafson et al., 2023) and CARLA simulator (Dosovitskiy et al., 2017) are publicly available. Code, training scripts, dataset wrappers, and pretrained checkpoints are provided in the supplementary material and are publicly available at `https://github.com/dawoodwasif/RESFL`.

## ACKNOWLEDGEMENTS

This work was supported in part by the National Science Foundation under Award No. 2107450 and by the Army Research Office under Grant No. W911NF-24-2-0241.

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

# Appendix to "RESFL: An Uncertainty-Aware Framework for Responsible Federated Learning by Balancing Privacy, Fairness and Utility"

## A. PRELIMINARIES

This section presents the mathematical foundations and system specifications of our work. We detail the YOLOv8-based object detection model, describe the FL setup in the AV scenario, formalize threat models (privacy, robustness, and fairness attacks), and define evaluation metrics. We also provide a unified overview of the datasets used for training and testing.

### A.1. SYSTEM MODEL: OBJECT DETECTION

Let $I \in \mathbb{R}^{H \times W \times C}$ denote an input image. Our object detection model, derived from YOLOv8, produces a set of detections:

$$\mathcal{P} = \{(b_i, c_i, s_i)\}_{i=1}^{N},$$

where each $b_i \in \mathbb{R}^4$ specifies the bounding box coordinates, $c_i \in \{1, \dots, C\}$ is the predicted class label, and $s_i \in [0, 1]$ is the corresponding confidence score. The overall detection loss is given by:

$$\mathcal{L}_{\text{det}} = \lambda_{\text{cls}} \, \mathcal{L}_{\text{cls}} + \lambda_{\text{loc}} \, \mathcal{L}_{\text{loc}} + \lambda_{\text{conf}} \, \mathcal{L}_{\text{conf}}, \tag{11}$$

where $\mathcal{L}_{\text{cls}}$, $\mathcal{L}_{\text{loc}}$, and $\mathcal{L}_{\text{conf}}$ represent the classification, localization, and confidence losses, respectively, and $\lambda_{\text{cls}}, \lambda_{\text{loc}}, \lambda_{\text{conf}} \in \mathbb{R}^{+}$ are hyperparameters.

### A.2. FEDERATED LEARNING SETUP AND NETWORK MODEL

Consider a set of $N$ clients $\{C_i\}_{i=1}^{N}$, each possessing a local dataset $\mathcal{D}_i \subset \mathbb{R}^{H \times W \times C}$ and a local model with parameters $\theta_i$. A central server maintains the global model $\theta_G$. The FL process begins with the server initializing and distributing $\theta_G^{(0)}$ to all clients. Each client then updates its model by performing local stochastic gradient descent (SGD):

$$\theta_i^{(t+1)} = \theta_i^{(t)} - \eta \nabla \mathcal{L}_i \left( \theta_i^{(t)} \right),$$

where $\eta > 0$ is the learning rate, $t$ is the local iteration index, and $\mathcal{L}_i$ is the local loss (e.g., $\mathcal{L}_{\text{det}}$). The server aggregates the locally updated parameters via FedAvg:

$$\theta_G^{(t+1)} = \sum_{i=1}^{N} \frac{|\mathcal{D}_i|}{\sum_{j=1}^{N} |\mathcal{D}_j|} \theta_i^{(t+1)}.$$

This FL framework preserves data locality (raw data remain on devices); the server receives only model updates and no formal privacy guarantee is implied.

### A.3. UNCERTAINTY QUANTIFICATION VIA EVIDENTIAL REGRESSION

**Evidential head for detection (Dirichlet & NIG).** Our detector uses a decoupled evidential head on top of YOLOv8. For every anchor/location, the *classification* branch outputs a nonnegative concentration vector $\boldsymbol{\alpha} = (\alpha_1, \dots, \alpha_C)$ via $\alpha_c = 1 + \text{softplus}(z_c)$, which parameterizes a Dirichlet over class probabilities. The *localization* branch outputs Normal–Inverse–Gamma (NIG) parameters for each box coordinate $q \in \{x, y, w, h\}$, $(\gamma_q, \nu_q, \alpha_q^{\text{nig}}, \beta_q)$, following (Amini et al., 2020). Concretely:

$$p(\mathbf{p} \mid \boldsymbol{\alpha}) = \text{Dir}(\boldsymbol{\alpha}), \qquad (\mu_q, \sigma_q^2) \sim \text{NIG}\left(\gamma_q, \nu_q, \alpha_q^{\text{nig}}, \beta_q\right).$$

For the Dirichlet, total evidence $\alpha_0 = \sum_{c=1}^{C} \alpha_c$ controls epistemic uncertainty; larger $\alpha_0$ implies lower epistemic variance (Sensoy et al., 2018). For the NIG, the epistemic variance of the mean is $\text{Var}[\mu_q] = \beta_q / (\nu_q(\alpha_q^{\text{nig}} - 1))$.

**Per-detection scalar uncertainties.** We extract two scalar measures:

$$u_{\text{cls}} = \frac{1}{\alpha_0 + 1} \quad \text{(classification epistemic; lower is more confident)},$$

$$u_{\text{box}} = \frac{1}{4} \sum_{q \in \{x,y,w,h\}} \frac{\text{Var}[\mu_q]}{s_q^2} \quad \text{(localization epistemic; normalized, lower is more confident)}.$$

Here $s_x = W$, $s_y = H$, $s_w = W$, $s_h = H$ are image-scale normalizers (width $W$ and height $H$) so that $u_{\text{box}}$ is dimensionless and comparable across resolutions. In practice we compute $\text{Var}[\mu_q] = \beta_q/(\nu_q(\alpha_q^{\text{nig}} - 1 + \epsilon))$ with a small $\epsilon$ for stability.

**What feeds UFM.** Unless otherwise specified, **UFM is computed only from the classification Dirichlet evidence**. For an image $x$ with detections $\mathcal{P}_\tau(x)$ (post-NMS, score $\geq \tau$), define the group-wise mean evidence $\bar{\alpha}_{0,g}$ as in Eq. 3 of the main text. Plug $\{\bar{\alpha}_{0,g}\}_{g=1}^G$ into the UFM definition Eq. 4 to obtain $\text{UFM} = \text{UFM}_{\text{cls}}$.

**Training losses.** The classification branch is trained with the Dirichlet NLL plus an evidential regularizer that penalizes overconfident errors (Sensoy et al., 2018); the localization branch uses the NIG NLL with the regularizer from (Amini et al., 2020). This yields calibrated epistemic estimates for both branches while keeping the UFM definition unambiguous.

## A.4. Threat Model

We study a cross-silo FL setting with an honest-but-curious server that observes client updates and may collude with a subset of clients. Training data remain on-device and are never shared; thus raw data exposure is reduced, but parameter-level inference and sensitive-attribute leakage from updates/models remain possible. Our privacy goal is *parameter privacy*: reduce sensitive-attribute leakage from intermediate representations or model updates. We also evaluate *robustness* to malicious updates and *fairness* against bias amplification. The attacks below instantiate these goals.

### A.4.1. Privacy Attacks

The selected privacy attacks assess whether an adversary can extract sensitive information from federated model updates.

*Membership Inference Attack (MIA):* MIA tests whether a sample $x \in \mathbb{R}^d$ was used in training. An adversarial client $C_a$ trains a shadow model to mimic the global model $M_t$, queries $M_t$ on member/non-member samples, and uses the resulting outputs to train a binary classifier $\mathcal{A}_{\text{MIA}}$ that predicts membership:

$$\mathcal{A}_{\text{MIA}}(x) = \begin{cases} 1, & x \in \mathcal{D}_{\text{train}} \\ 0, & x \notin \mathcal{D}_{\text{train}} . \end{cases}$$

We report the *MIA Success Rate* (binary accuracy):

$$S_{\text{MIA}} = \frac{TP + TN}{TP + TN + FP + FN}, \tag{12}$$

where $TP, TN, FP, FN$ are counts over a balanced member/non-member evaluation set. Higher $S_{\text{MIA}}$ indicates greater privacy leakage.

*Attribute Inference Attack (AIA):* AIA tests whether a sensitive attribute $s \in \mathcal{S}$ can be inferred from update-derived features. The adversary extracts features $I$ from observed gradients/updates and trains $\mathcal{A}_{\text{AIA}}$ to predict $s$:

$$\hat{s} = \mathcal{A}_{\text{AIA}}(I). \tag{13}$$

We report the *AIA Success Rate* as top-1 accuracy over $M$ instances:

$$S_{\text{AIA}} = \frac{1}{M} \sum_{i=1}^{M} \mathbf{1}\{\hat{s}_i = s_i\}. \tag{14}$$

(For binary attributes, $S_{\text{AIA}}$ reduces to standard binary accuracy.)

**Adversary knowledge and data.** We assume an honest-but-curious server that sees per-round updates and the final global model. Attack probes are trained only on model outputs and update metadata computed over a small, non-overlapping slice of the benchmark that we reserve exclusively for attack training (no private client data are used). For FACET, this slice is sampled from unused images disjoint from our train/test splits; for CARLA, it is rendered with independent seeds and kept separate from tuning/evaluation.

**Membership inference (MIA) protocol and reporting.** We adopt a score-based MIA with a single shadow run per method/seed to preserve compute parity. For FACET, we reserve a fixed **5%** of the full benchmark as an *attack-reserve* slice *disjoint* from all train/val/test splits (1,600 images; stratified by MST). For CARLA, we render an additional **600** clear-weather frames with independent seeds (balanced by town and MST) reserved exclusively for attacks and never used in tuning or evaluation. For each method, the adversary (i) trains one shadow model with the same optimizer and schedule on a *subset* of the attack-reserve; (ii) collects per-example attack features for both *members* (sampled from the actual training sets of participating clients) and *non-members* (from the remaining attack-reserve portion) by querying the *target* global model at round $T$: final loss, logit margins, evidential concentration summaries (mean/variance of $\alpha$), and confidence; and (iii) fits a calibrated logistic classifier on a balanced member/non-member training set (attack-reserve split: 70% train, 10% validation for threshold selection, 20% test).

**Attribute inference (AIA) protocol and reporting.** We evaluate an update-aware attacker aligned with the server's view. From each round's client update we extract layerwise meta-features (per-layer $\ell_2$ norm, sign ratio, top-$k$ index histogram of magnitudes, and head-gradient statistics), concatenate across layers, and train a probe to predict the sensitive attribute $s$. Attack training uses a balanced attribute distribution, a 70/10/20 train/val/test split within the attack-reserve, and is strictly disjoint from the model's train/val/test data. AIA is reported as top-1 accuracy (lower is better) with macro-F1 in the additional experiments in Appendix H.

**Supplementary privacy results (FACET, 4 clients, IID).** Tables 13 and 14 summarize the additional MIA/AIA results; values are mean $\pm$ std over three seeds. Privacy Score in the main paper continues to use balanced accuracy for MIA and AIA; the TPR@FPR values are reported here for completeness.

### A.4.2. Robustness Attack

We assess the FL system's resilience to malicious modifications of model updates using a robustness attack.

*Byzantine Attack:* In a Byzantine attack, a subset of clients manipulates their model updates before sending them to the central aggregator. Let $\theta_k$ be the legitimate update from client $k$, and let the adversary introduce a perturbation $\delta_k$, yielding a modified update:

$$\tilde{\theta}_k = \theta_k + \delta_k, \quad \text{with} \quad \|\delta_k\| \gg 0. \tag{15}$$

A sufficiently large $\delta_k$ disrupts training, leading to model divergence or severe performance degradation. The attack's impact is quantified by comparing global model accuracy without malicious interference ($A_{\text{clean}}$) to accuracy under Byzantine updates ($A_{\text{Byzantine}}$), measured as:

$$D_{\text{Byz}} = A_{\text{clean}} - A_{\text{Byzantine}}. \tag{16}$$

A larger $D_{\text{Byz}}$ indicates a stronger attack and greater vulnerability of the federated learning system to such perturbations.

*Data Poisoning Attack:* This attack injects manipulated samples $\Delta\mathcal{D}$ into a client's local dataset, altering its distribution. The poisoned dataset is defined as:

$$\mathcal{D}'_k = \mathcal{D}_k \cup \Delta\mathcal{D}. \tag{17}$$

The adversary selects injected samples to skew feature distributions, favoring one demographic group over another and shifting the global model's decision boundaries. The impact on fairness

is measured using the Equalized Odds Difference (EOD), which quantifies disparities in true positive rates (TPR) and false positive rates (FPR) between protected and unprotected groups:

$$EOD = |\text{TPR}_{\text{protected}} - \text{TPR}_{\text{unprotected}}| \quad + |\text{FPR}_{\text{protected}} - \text{FPR}_{\text{unprotected}}|. \tag{18}$$

To assess the attack's effect, we compute the Equalized Odds Difference Deviation (EODD) as the change in EOD between the poisoned and clean datasets:

$$EODD = EOD_{\text{poisoned}} - EOD_{\text{clean}}. \tag{19}$$

A larger $EODD$ indicates greater fairness violation, confirming the attack's success in introducing bias. This highlights the dual threat of data poisoning, which compromises both model accuracy and group-level parity across user groups.

### A.5. DETECTION-BASED FAIRNESS & PRIVACY METRICS

**Setup and matching protocol.** For each image $x$, let $\mathcal{G}(x) = \{(b_k^*, y_k, g_k)\}_{k=1}^{N_x}$ be ground-truth person instances with box $b_k^*$, class $y_k = \texttt{person}$, and sensitive group $g_k \in \{1, \ldots, G\}$ (e.g., MST). Let $\mathcal{P}(x) = \{(\hat{b}_i, \hat{c}_i, \hat{s}_i)\}_{i=1}^{M_x}$ be predicted boxes, classes, and confidences after standard NMS (we use IoU NMS=0.5 and score threshold $\tau_{\text{score}} = 0.25$; same across all methods). We evaluate fairness at a fixed IoU threshold $\tau_{\text{fair}} = 0.5$ (distinct from mAP's COCO sweep).

We perform a one-to-one greedy match between $\mathcal{G}(x)$ and $\mathcal{P}(x)$ by descending $\hat{s}_i$: a prediction $(\hat{b}_i, \hat{c}_i, \hat{s}_i)$ matches a ground-truth $(b_k^*, y_k, g_k)$ iff $\hat{c}_i = y_k$ and $\text{IoU}(\hat{b}_i, b_k^*) \geq \tau_{\text{fair}}$ and neither has been matched yet. Matched pairs count as true positives (TP); unmatched predictions are false positives (FP); unmatched ground truths are false negatives (FN). Multiple predictions for the same ground-truth are penalized as FP except the highest-scoring matched one.

**Per-group rates (micro-averaged).** Let $\text{TP}_g = \sum_x \sum_{k:g_k=g} \mathbf{1}\{\text{GT } k \text{ is matched}\}$ and $\text{FN}_g = \sum_x \sum_{k:g_k=g} \mathbf{1}\{\text{GT } k \text{ is unmatched}\}$. Define the per-group detection true positive rate (recall)

$$\text{TPR}_g(\tau_{\text{fair}}) = \frac{\text{TP}_g}{\text{TP}_g + \text{FN}_g}.$$

In our fairness metrics, the *favorable outcome* is a *correct detection of a person instance* (i.e., contributing to $\text{TP}_g$ at $\text{IoU} \geq \tau_{\text{fair}}$ with correct class). Counts are aggregated *over all instances* in the cohort (micro average across images).

**Why TPR/recall for fairness?** Equality of opportunity is defined in terms of true–positive rate (TPR) (Hardt et al., 2016). In detection, the favorable event is a correct person detection at $\text{IoU} \geq \tau_{\text{fair}}$, so TPR/recall is the natural rate for both DI and $\Delta$EOP. Empirically, substituting precision or per-group AP changes magnitudes but preserves the cohort and method ordering under our fixed matching protocol and thresholds (Appendix G).

**Disparate Impact (DI) for detection.** With multiple cohorts, we compute the best and worst group detection rates and form a ratio:

$$\text{DI}(\tau_{\text{fair}}) = \frac{\min_g \text{TPR}_g(\tau_{\text{fair}})}{\max_g \text{TPR}_g(\tau_{\text{fair}})} \in [0, 1], \qquad |1 - \text{DI}| \text{ is reported (lower is better).}$$

This "rate ratio" view is standard for multi-group DI and equals 1 under perfect parity.

**Equality of Opportunity gap ($\Delta$EOP) for detection.** We report the max range of per-group TPRs:

$$\Delta\text{EOP}(\tau_{\text{fair}}) = \max_g \text{TPR}_g(\tau_{\text{fair}}) - \min_g \text{TPR}_g(\tau_{\text{fair}}) \in [0, 1],$$

which is 0 under perfect parity (lower is better). Note that both DI and $\Delta$EOP use the *same* matching protocol and $\tau_{\text{fair}}$.

**Relation to mAP.** mAP is computed with the COCO protocol (IoU $\in \{0.50 : 0.05 : 0.95\}$, class-aware). Fairness metrics use the *single* threshold $\tau_{\text{fair}} = 0.5$ defined above so that TP/FP/FN (and thus TPR) are unambiguous and reproducible.

---

**Reproducibility keys for fairness on detection** (*we fix these in all experiments*):
– IoU NMS = 0.5; score threshold $\tau_{\text{score}} = 0.25$; max dets per image = 300.
– Fairness IoU threshold $\tau_{\text{fair}} = 0.5$ for TP/FP/FN and TPR.
– Greedy one-to-one matching by descending score; ties broken by higher IoU.
– Per-group TPR is micro-averaged over all instances with that group's ground-truth.

---

**Membership Inference Attack (MIA) for detection.** We use a black-box shadow-model attack tailored to detectors. Let $\mathcal{M}$ be a set of *member* images used in training and $\bar{\mathcal{M}}$ a disjoint *non-member* set of the same size from the same distribution. For any image $x$, we compute an image-level feature vector from the model's post-NMS outputs:

$$\varphi(x) = \left[ \#\text{dets}, \; \overline{\hat{s}}, \; \max \hat{s}, \; \overline{\alpha_0}, \; \overline{u_{\text{cls}}} \right],$$

where $\#$dets is the number of predicted person boxes with $\hat{s} \geq \tau_{\text{score}}$, $\overline{\hat{s}}$ is their mean confidence, and $\overline{\alpha_0}$ and $\overline{u_{\text{cls}}}$ are the mean Dirichlet total evidence and its derived epistemic scalar (Sec. A.3) over those detections (empty sets use zeros). We train a logistic-regression (or two-layer MLP) shadow attacker on a disjoint shadow split to predict membership from $\varphi(x)$ and report the *attack success rate*

$$\text{MIA SR} = \frac{\text{TP} + \text{TN}}{\text{TP} + \text{TN} + \text{FP} + \text{FN}},$$

evaluated on $\mathcal{M} \cup \bar{\mathcal{M}}$ with a 50/50 prior.

**Attribute Inference Attack (AIA) for detection.** We probe sensitive attributes from per-instance features. For each matched detection (as above), we apply ROIAlign to the model's neck feature map at the matched box to get a fixed-size tensor, global-average–pool it to a vector $h$, and train a two-layer MLP (on a disjoint shadow split) to predict the instance's group $g \in \{1, \ldots, G\}$. We report the *top-1 accuracy* over all matched instances:

$$\text{AIA SR} = \frac{\#\text{correct group predictions}}{\#\text{matched instances}}.$$

*Note*: Images without matched persons contribute nothing to AIA; they still contribute to MIA.

## B. THEORETICAL ANALYSIS OF THE UNCERTAINTY FAIRNESS METRIC

In federated learning, each client holds a distinct local distribution, resulting in unequal representation of sensitive groups (e.g. demographic cohorts or environmental conditions). Epistemic uncertainty (quantifying a model's ignorance about its predictions) naturally reflects these imbalances: groups with fewer or more variable examples yield higher uncertainty, while well-represented, homogeneous groups yield lower uncertainty. We formalize this insight and show that controlling the dispersion of group-wise uncertainties effectively enforces fairness.

Let each client compute, for each sensitive group $g \in \{1, \ldots, G\}$, an epistemic variance $\sigma_g^2$. In evidential models, $\sigma_g^2 = 1/\alpha_g$, where $\alpha_g$ accumulates "evidence" proportional to effective sample size and signal-to-noise ratio. Concretely,

$$\alpha_g \; \propto \; n_g \, \text{SNR}_g \quad \Longrightarrow \quad \sigma_g^2 \; \approx \; \frac{1}{n_g \, \text{SNR}_g} \, .$$

Thus $\sigma_g^2$ is large when group $g$ is under-sampled or noisy, and small otherwise.

To measure fairness, we track the relative spread of $\{\sigma_g^2\}$. We define the Uncertainty Fairness Metric (UFM) as

$$\text{UFM} \; = \; \frac{\max_g \sigma_g^2 \; - \; \min_g \sigma_g^2}{\frac{1}{G} \sum_{g=1}^{G} \sigma_g^2 + \epsilon},$$

with $\epsilon > 0$ for stability. By normalizing by the mean uncertainty, UFM is scale-invariant, takes value zero when all groups share equal confidence, and grows smoothly as disparities arise.

**Statistical Rationale.** Classical generalization bounds for group $g$ involve its sample size $n_g$ and model complexity. A simplified high-probability bound is

$$\mathcal{L}^{(g)} \ \leq \ \hat{\mathcal{L}}^{(g)} + \mathcal{O}\Big(\sqrt{\tfrac{\log(1/\delta)}{n_g}}\Big),$$

where $\hat{\mathcal{L}}^{(g)}$ is the empirical loss. Since $\sigma_g^2 \approx 1/(n_g \, \mathrm{SNR}_g)$, the uncertainty gap $\max_g \sigma_g^2 - \min_g \sigma_g^2$ upper-bounds the disparity in confidence-adjusted generalization terms. Minimizing UFM therefore tightens and balances each group's bound, improving fairness in expected loss.

**Fair Aggregation.** In federated aggregation, each client $i$ reports its $\mathrm{UFM}_i$. The server assigns weights

$$w_i \ = \ \frac{\exp(-\beta\,\mathrm{UFM}_i)}{\sum_j \exp(-\beta\,\mathrm{UFM}_j)},$$

so that clients with lower internal disparity (smaller UFM) contribute more. This bias toward uniformly confident updates automatically re-balances the global model as data distributions evolve, without exposing sensitive attributes centrally.

**Notation alignment.** Throughout, evidential classification uses Dirichlet evidence with total evidence $\alpha_0(x) = \sum_{c=1}^C \alpha_c(x)$. For group $g$, let $\bar{\alpha}_{0,g} = \mathbb{E}_{x \in \mathcal{D}_g}[\alpha_0(x)]$ and define the group epistemic variance proxy $\sigma_g^2 := \mathbb{E}_{x \in \mathcal{D}_g}[1/\alpha_0(x)] \approx 1/\bar{\alpha}_{0,g}$. We set $\epsilon = 10^{-6}$ in UFM for numerical stability.

**Assumptions.** (A1) *Bounded loss:* the per-sample loss $\ell \in [0,1]$. (A2) *Evidential calibration:* there exist constants $0 < s_{\min} \leq s_{\max}$ such that $\frac{1}{n_g s_{\max}} \leq \sigma_g^2 \leq \frac{1}{n_g s_{\min}}$ for each group $g$ (i.e., evidence scales with effective sample size and signal-to-noise). (A3) *Group-wise mixing:* samples within a group are i.i.d. under a fixed distribution.

**Theorem B.1** (Confidence-adjusted generalization disparity). *Under (A1)–(A3), for any $\delta \in (0,1)$ there exists a constant $C > 0$ (depending only on $s_{\min}, s_{\max}$) such that, with probability at least $1 - \delta$, for every group $g$,*

$$\mathcal{L}^{(g)} \ \leq \ \hat{\mathcal{L}}^{(g)} \ + \ C\,\sqrt{\sigma_g^2 \log(1/\delta)}.$$

*Consequently,*

$$\max_g\Big(\mathcal{L}^{(g)} - \hat{\mathcal{L}}^{(g)}\Big) - \min_g\Big(\mathcal{L}^{(g)} - \hat{\mathcal{L}}^{(g)}\Big) \ \leq \ C\sqrt{\log(1/\delta)}\,\Big(\sqrt{\max_g \sigma_g^2} - \sqrt{\min_g \sigma_g^2}\Big).$$

*Proof sketch.* Hoeffding's inequality yields $\mathcal{L}^{(g)} \ \leq \ \hat{\mathcal{L}}^{(g)} + O\Big(\sqrt{\tfrac{\log(1/\delta)}{n_g}}\Big)$ under (A1),(A3). By (A2), $1/n_g$ is sandwiched by constants times $\sigma_g^2$, giving the per-group term $O\Big(\sqrt{\sigma_g^2 \log(1/\delta)}\Big)$. The disparity bound follows by subtracting the best/worst groups. □

**Corollary B.2** (UFM controls disparity). *Let $\bar{\sigma}^2 = \frac{1}{G}\sum_{g=1}^G \sigma_g^2$. Then there exist constants $C_1, C_2 > 0$ such that*

$$\max_g\Big(\mathcal{L}^{(g)} - \hat{\mathcal{L}}^{(g)}\Big) - \min_g\Big(\mathcal{L}^{(g)} - \hat{\mathcal{L}}^{(g)}\Big) \ \leq \ C_1\sqrt{\log(1/\delta)}\,\bar{\sigma}\,\mathrm{UFM} \ \leq \ C_2\sqrt{\log(1/\delta)}\,\mathrm{UFM},$$

*i.e., minimizing UFM tightens a normalized upper bound on the group disparity in confidence-adjusted generalization.*

**Federated lifting to the global mixture.** Let $P_i$ be client $i$'s data distribution, and define the global evaluation distribution as the mixture $P_{\mathrm{eval}} := \sum_{i=1}^N \pi_i P_i$ with fixed nonnegative weights $\{\pi_i\}$ summing to 1. For group $g$, let $\sigma_{g,i}^2$ denote client $i$'s epistemic-variance proxy and let $\sigma_g^2$ be the proxy under $P_{\mathrm{eval}}$. Write $\mathrm{UFM}_i \equiv \mathrm{UFM}(\{\sigma_{g,i}^2\}_{g=1}^G)$ and $\mathrm{UFM}_\star \equiv \mathrm{UFM}(\{\sigma_g^2\}_{g=1}^G)$.

**Lemma B.3** (Mixture reduction). *Assume the evidential head is 1-Lipschitz in parameters and locally stable across an aggregation step so that $(x, \theta) \mapsto \alpha_0(x; \theta)$ is dominated and measurable. Then for each group $g$,*

$$\sigma_g^2 \;=\; \mathbb{E}_{x \sim P_{\text{eval}}}\Big[\tfrac{1}{\alpha_0(x;\theta)}\Big] \;\leq\; \sum_{i=1}^{N} \pi_i \, \mathbb{E}_{x \sim P_i}\Big[\tfrac{1}{\alpha_0(x;\theta)}\Big] \;=\; \sum_{i=1}^{N} \pi_i \, \sigma_{g,i}^2.$$

**Proposition B.4** (Global disparity bound). *Let $\bar{\sigma}^2 := \frac{1}{G}\sum_{g=1}^{G} \sigma_g^2$ and $\bar{\sigma}_i^2 := \frac{1}{G}\sum_{g=1}^{G} \sigma_{g,i}^2$. Under Lemma B.3 and Theorem B.1, there exist constants $C_1, C_2 > 0$ such that with probability at least $1 - \delta$,*

$$\max_g\Big(\mathcal{L}^{(g)} - \hat{\mathcal{L}}^{(g)}\Big) - \min_g\Big(\mathcal{L}^{(g)} - \hat{\mathcal{L}}^{(g)}\Big) \;\leq\; C_1\sqrt{\log(1/\delta)}\,\bar{\sigma}\,\text{UFM}_\star \;\leq\; C_2\sqrt{\log(1/\delta)}\sum_{i=1}^{N}\pi_i\,\bar{\sigma}_i\,\text{UFM}_i.$$

**Corollary B.5** (Effect of UFM-weighted aggregation). *Let the server choose aggregation weights $\omega_i \propto \exp(-\beta\,\text{UFM}_i)$ and let evaluation weights $\pi_i$ be data-proportional or equal across clients. If $\text{UFM}_i$ is computed on a held-out local split and remains stable across a round, then decreasing $\sum_i \omega_i \bar{\sigma}_i \text{UFM}_i$ tightens the upper bound in Proposition B.4. Hence, making $\omega_i$ decrease in $\text{UFM}_i$ reduces a provable surrogate of the global fairness gap.*

**Link to DI and $\Delta$EOP.** Assume the per-group TPR at the fixed fairness IoU is $L$-Lipschitz in $\sigma_g^2$ and monotone. Then for some $L, L' > 0$,

$$\Delta\text{EOP} \;\leq\; L\Big(\sqrt{\max_g \sigma_g^2} - \sqrt{\min_g \sigma_g^2}\Big), \qquad |1 - \text{DI}| \;\leq\; L' \,\frac{\max_g \sigma_g^2 - \min_g \sigma_g^2}{\min_g \sigma_g^2}.$$

Combining with Proposition B.4 yields explicit bounds on DI and $\Delta$EOP that decrease as the aggregation-weighted $\text{UFM}_i$ decrease.

**Aggregation limits and practice.** We compute $\text{UFM}_i$ per client on a *held-out local validation split* and report $w_i \propto \exp(-\beta\,\text{UFM}_i)$. As $\beta \to 0$ we recover *uniform* averaging; as $\beta \to \infty$ the aggregator concentrates on clients with smallest UFM. To reduce noise, we use an exponential moving average of $\text{UFM}_i$ across rounds.

**Integrity of reported UFM.** We operate under the standard honest-but-curious federation assumption, where clients follow the protocol but may attempt to infer others' information. To ensure consistency, each client reports its scalar fairness statistic $u_i = \text{UFM}_i$ after applying a publicly known clipping rule,

$$u_i \;\leftarrow\; \min\big(\max(u_i, a),\, b\big), \quad (a, b) \text{ fixed.}$$

This limits the influence of any extreme or falsified report. In our simulator, model updates $\Delta\theta_i$ and the bounded scalar $u_i$ are transmitted in the clear to the server (no cryptographic secure aggregation which is a future work). During aggregation, the server computes $\omega_i \propto \exp(-\beta\,u_i)$ and normalizes across clients as in Eq. (5)–(6).

Although $u_i$ is a single scalar, it is designed as a scale-invariant summary of inter-group uncertainty disparity, chosen to minimize communication cost while preserving the monotonic relation between UFM and established fairness metrics such as $|1 - \text{DI}|$ and $\Delta$EOP (see Appendix B). Empirically, this scalar retains high correlation ($r > 0.9$) with those measures, maintaining the correct ordering of clients by fairness.

In potentially untrusted deployments, additional safeguards can be incorporated: (i) cross-round consistency checks on $\|\Delta\theta_i\|$ and evidence statistics to detect anomalous $u_i$ values, (ii) trimmed-mean or median aggregation of UFMs to bound single-client influence, and (iii) optional secure or verifiable reporting schemes (e.g., commitments or zero-knowledge proofs of evidence norms). These mechanisms are modular and do not alter the core optimization of `RESFL`.

## C. INFORMATION-THEORETIC GUARANTEES OF ADVERSARIAL PRIVACY DISENTANGLEMENT

We analyze how adversarial training with a gradient reversal layer and an attribute classifier limits the leakage of a sensitive attribute $S$ from representations $H = f_\theta(X)$. Throughout, we allow $H$ to be a (possibly stochastic) mapping of $X$ (e.g., due to dropout); all logarithms are natural (nats).

**Adversarial objective and conditional entropy.** Consider the minimax problem

$$\min_\theta \max_\phi \mathcal{L}_{\mathrm{adv}}(\theta, \phi), \qquad \mathcal{L}_{\mathrm{adv}}(\theta, \phi) = -\mathbb{E}_{(X,S)}\big[\log A_\phi(S \mid H)\big],$$

where $A_\phi(\cdot \mid H)$ is the attribute classifier fed by the representation $H = f_\theta(X)$. If the adversary family is universally expressive and the supremum is attained, then

$$\sup_\phi \mathcal{L}_{\mathrm{adv}}(\theta, \phi) = H(S \mid H).$$

In general (with approximation/optimization error), we have the one–sided relation

$$\sup_\phi \mathcal{L}_{\mathrm{adv}}(\theta, \phi) \leq H(S \mid H).$$

Consequently,

$$I(H; S) = H(S) - H(S \mid H) \leq H(S) - \sup_\phi \mathcal{L}_{\mathrm{adv}}(\theta, \phi),$$

so maximizing $\mathcal{L}_{\mathrm{adv}}$ (for fixed $\theta$) *minimizes* an upper bound on $I(H; S)$. By the data–processing inequality, reducing $I(H; S)$ weakens any inference from $H$ about $S$.

**Attack error via Fano.** Let an attacker output $\widehat{S} = g(H)$ over $K$ attribute classes. Fano's inequality yields

$$P_e \geq 1 - \frac{I(H; S) + \log 2}{\log K}.$$

Hence as $\sup_\phi \mathcal{L}_{\mathrm{adv}}(\theta, \phi) \to H(S \mid H)$ (i.e., $I(H; S) \to 0$), the minimum achievable error satisfies $P_e \to 1 - 1/K$, driving attribute inference toward chance level.

**Privacy–utility frontier and tuning.** Incorporating the adversarial term with coefficient $\lambda_{\mathrm{priv}}$ into the local objective,

$$\mathcal{L}_{\mathrm{local}}(\theta, \phi) = \mathcal{L}_{\mathrm{task}}(\theta) + \lambda_{\mathrm{priv}} \, \mathcal{L}_{\mathrm{adv}}(\theta, \phi),$$

traces a (piecewise) convex frontier in the $\big(I(H; S), \mathcal{L}_{\mathrm{task}}\big)$ plane under standard regularity conditions. By the envelope theorem,

$$-\frac{d \, I(H; S)}{d \, \mathcal{L}_{\mathrm{task}}} = \frac{\partial_{\lambda_{\mathrm{priv}}} \mathcal{L}_{\mathrm{task}}}{\partial_{\lambda_{\mathrm{priv}}} I(H; S)},$$

so $\lambda_{\mathrm{priv}}$ directly tunes the privacy–utility balance: larger $\lambda_{\mathrm{priv}}$ increases the pressure to maximize $\mathcal{L}_{\mathrm{adv}}$, thereby decreasing $I(H; S)$ (stronger privacy) at the potential cost of task loss.

**Scope of the guarantee.** These are *information-theoretic* guarantees on representation leakage $I(H; S)$; they are not $(\varepsilon, \delta)$–DP guarantees on the training algorithm. In practice, one monitors $\mathcal{L}_{\mathrm{adv}}$ (or a calibrated surrogate) and adjusts $\lambda_{\mathrm{priv}}$ to meet a target inference–error bound while limiting degradation in $\mathcal{L}_{\mathrm{task}}$.

## D. JOINT TRADE-OFF ANALYSIS OF PRIVACY AND FAIRNESS

Our RESFL framework simultaneously addresses three competing objectives, detection utility, attribute privacy, and demographic fairness, by integrating adversarial privacy disentanglement with uncertainty-guided aggregation (summarized in Algorithm 1. The adversarial module employs a gradient reversal layer and attribute classifier to suppress sensitive information in each client's features, effectively minimizing the mutual information between latent representations and protected

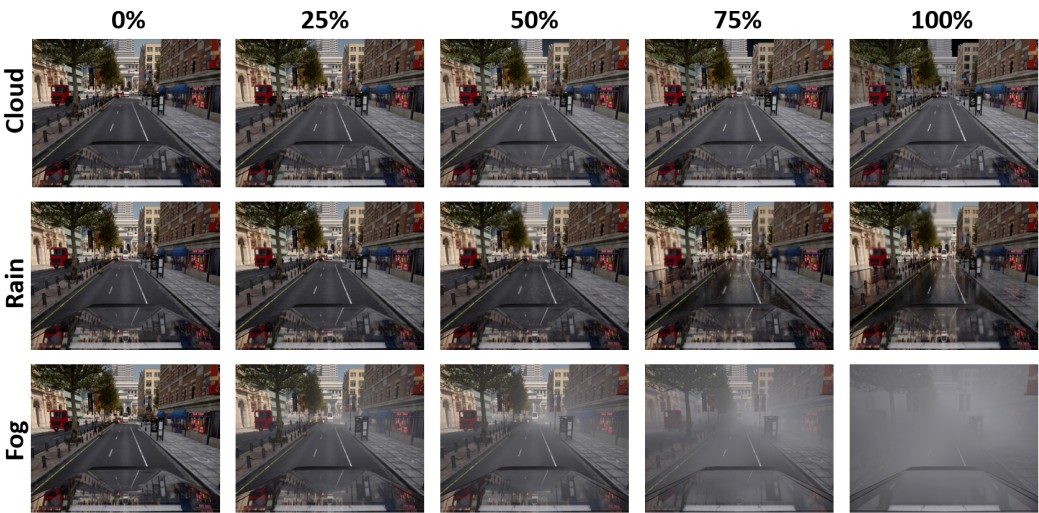

Figure 3: Sample visualization of weather conditions (cloud, rain, and fog) at increasing intensity levels (0%, 25%, 50%, 75%, 100%) using the CARLA simulation, illustrating how environmental severity gradually impacts visibility and scene clarity.

---

**Algorithm 1** `RESFL` Training with Adversarial Privacy and Uncertainty-Guided Aggregation

---

1: **Input:** global model $\theta_G$, adversary $A(x; \phi)$, client data $\{\delta_i\}$, weights $\lambda_{\text{priv}}, \lambda_{\text{fair}}$, temperature $\beta$, learning rates $\eta, \eta_\phi$, rounds $T$
2: **for** $t = 0 \rightarrow T - 1$ **do**
3:     Server broadcasts $\theta_G^{(t)}$ to all clients
4:     **for** each client $i$ **in parallel do**
5:         Initialize: $\theta_i \leftarrow \theta_G^{(t)}, \phi_i \leftarrow \phi$
6:         **for** each local step **do**
7:             Compute $L_{\text{task}}, L_{\text{adv}}, L_{\text{unc}}$
8:             $\phi_i \leftarrow \phi_i - \eta_\phi \nabla_{\phi_i} L_{\text{adv}}$
9:             $\theta_i \leftarrow \theta_i - \eta \nabla_{\theta_i} [L_{\text{task}} + \lambda_{\text{priv}} L_{\text{adv}} + \lambda_{\text{fair}} L_{\text{unc}}]$       ▷ Composite local loss (Eq. (9))
10:         **end for**
11:         Compute $\text{UFM}_i$ (Eq. (4) and $\Delta\theta_i = \theta_i - \theta_G^{(t)}$)
12:         Client sends $\{\Delta\theta_i, \text{UFM}_i\}$ to server
13:     **end for**
14:     Server computes $\omega_i \propto \exp(-\beta \cdot \text{UFM}_i)$ (Eq. (5))
15:     Update: $\theta_G^{(t+1)} = \theta_G^{(t)} + \sum_{i=1}^{N} \omega_i \Delta\theta_i$       ▷ Aggregation (Eq. (6))
16: **end for**
17: **Output:** final global model $\theta_G^{(T)}$

---

attributes. This enforces a controllable privacy constraint without degrading task performance unduly. In parallel, each client's evidential uncertainty head estimates per-group epistemic variances, from which we compute an Uncertainty Fairness Metric (UFM) that quantifies disparities in model confidence across sensitive cohorts. During aggregation, clients report both their parameter updates and UFM scores; the server then weights each update by a softmax of the negative UFM, amplifying contributions from clients with more uniform confidence and down-weighting those with high disparity. By tuning the adversarial strength $\lambda_{\text{priv}}$ and the uncertainty coefficient $\lambda_{\text{fair}}$, `RESFL` effectively scalarizes a convex multi-objective problem, tracing out the full Pareto frontier in the space of utility, privacy leakage, and fairness gap. Unlike single-objective baselines, which either sacrifice accuracy for privacy protection or apply fixed fairness regularizers, `RESFL` dynamically balances both axes: stronger adversarial signals tighten privacy guarantees, while uncertainty-based weights correct emerging fairness imbalances. Empirically and theoretically, this joint mechanism dominates pure DP or fairness-only schemes by exploring descent directions unavailable to one-dimensional

fixes, yielding models that maintain high mean average precision, provably low attribute-inference risk, and minimal performance disparity across sensitive groups.

## E. DATASETS

Our experiments leverage two complementary data sources: FACET for federated training and CARLA for controlled evaluation, to measure object detection utility, demographic fairness, privacy resilience, and robustness under diverse conditions. In this section, we detail dataset composition, annotation processing, domain-specific partitioning, and preprocessing pipelines.

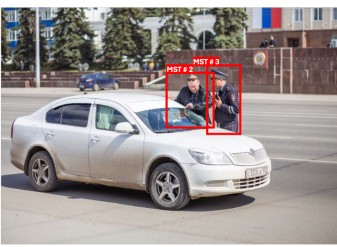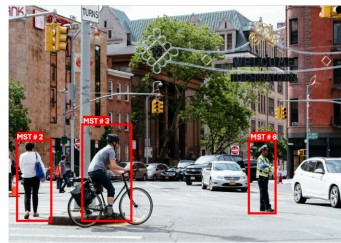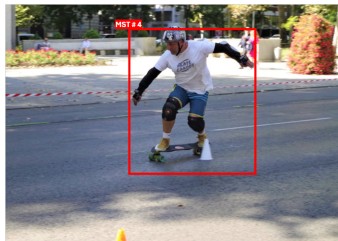

Figure 4: Example images from the FACET dataset. Each red bounding box denotes a detected person instance, annotated with its corresponding Monk Skin Tone (MST) label (e.g. MST #2, #3, #4, #6). These samples illustrate the range of skin-tone levels (1 = lightest to 10 = darkest) used for fairness evaluation in our object detection experiments.

### E.1. FACET DATASET

The FACET dataset (Gustafson et al., 2023) provides 32 000 real-world images with over 50 000 annotated person instances, each labeled with a bounding box and multiple attributes (perceived skin tone, hair type, person class). We concentrate on perceived skin tone, a sensitive attribute strongly correlated with performance gaps in detection model (Pathiraja et al., 2024). FACET adopts the Monk Skin Tone (MST) scale with ten discrete levels $g \in \{1, \ldots, 10\}$, where $g = 1$ is the lightest and $g = 10$ the darkest tone, as shown in Figure 5. To mitigate annotation noise due to lighting or labeler variance, each instance receives $n$ independent MST labels $s_1, \ldots, s_n$. We aggregate these as

$$s^* = \frac{1}{n} \sum_{i=1}^{n} s_i,$$

then discretize $s^*$ by rounding to the nearest integer in $\{1, \ldots, 10\}$. This yields a robust single-toned label per instance, denoted $\mathrm{MST}(b)$.

We group the dataset into $G = 10$ MST cohorts. Let $\mathcal{D} = \{(x_k, b_k, s_k^*)\}_{k=1}^{N}$ be the full set of image–instance pairs ($N \approx 50\,000$). Define

$$\mathcal{D}_g = \{(x, b, s^*) : \mathrm{MST}(b) = g\}, \quad g = 1, \ldots, 10,$$

so that $\sum_{g=1}^{10} |\mathcal{D}_g| = N$. In practice, each $|\mathcal{D}_g|$ ranges from approximately 4 000 to 6 000 instances, ensuring sufficient representation across the skin-tone spectrum.

To simulate federated clients, we partition the 32 000 FACET images into $K = 4$ i.i.d. subsets $\{\mathcal{I}_i\}_{i=1}^{4}$, each containing 8 000 images and all associated instances. Formally, $\mathcal{I}_i \cap \mathcal{I}_j = \emptyset$ for $i \neq j$, $\bigcup_{i=1}^{4} \mathcal{I}_i$ covers all images, and each split preserves the MST distribution:

$$\forall g, \ \left| \{(x, b) \in \mathcal{D}_g : x \in \mathcal{I}_i\} \right| \approx \tfrac{1}{4} |\mathcal{D}_g|.$$

Clients share only model updates (gradients $\Delta \theta_i$ and a scalar UFM per round), never raw images or labels. A few samples are visible in Figure 4.

**Preprocessing and Augmentation.** Each image is resized to $640 \times 640$ pixels using bicubic interpolation. We apply standard YOLOv8 augmentations: random horizontal flip (probability 0.5),

brightness and contrast jitter ($\pm 20\%$), and random hue shift ($\pm 10\%$). Pixel values are normalized to $[0, 1]$ and then standardized using ImageNet channel means $\mu = [0.485, 0.456, 0.406]$ and standard deviations $\sigma = [0.229, 0.224, 0.225]$. During training, we further apply mosaic augmentation by stitching four images into a $1 \times 1$ grid with random scaling in $[0.5, 1.5]$.

## E.2. CARLA SIMULATION DATASET

The CARLA simulator (Dosovitskiy et al., 2017) v0.9.13 generates synthetic driving scenarios to evaluate model robustness under controlled environmental and urban variations. We select three canonical maps: Town01 (suburban streets), Town03 (dense downtown), and Town05 (mid-density mixed-use) to capture a broad spectrum of road geometry, building density, and occlusion patterns.

An autopilot-enabled ego vehicle equipped with an RGB camera ($1920 \times 1080$, $100°$ FOV) and a semantic segmentation sensor (same specs) records frames every 3s. We retain only frames containing at least one pedestrian. Pedestrian bounding boxes

$$b = (x_{\min}, y_{\min}, x_{\max}, y_{\max} \in \mathbb{R}^4$$

are extracted via connected-component analysis on semantic masks, discarding detections with fewer than 50 pixels.

**Skin-tone assignment.** Each synthetic pedestrian blueprint is manually mapped to a Monk Skin Tone (MST) label by visual inspection, ensuring consistency with the FACET scale. Let

$$\mathcal{C} = \{\text{Town01, Town03, Town05}\}, \quad \mathcal{S} = \{1, \ldots, 10\}.$$

Each pedestrian instance receives a pair $(c, s) \in \mathcal{C} \times \mathcal{S}$.

**Domain adaptation fine-tuning.** To reduce domain shift, we fine-tune the federated global model on clear-weather CARLA frames. For each $c \in \mathcal{C}$ and $s \in \mathcal{S}$ we sample 200 frames, yielding

$$N_{\text{tune}} = |\mathcal{C}| \times |\mathcal{S}| \times 200 = 3 \times 10 \times 200 = 6000$$

tuning samples. Fine-tuning uses the same SGD hyperparameters as federated local updates.

**Fine-tuning scope and schedule.** Starting from the global checkpoint at round $T{=}100$, we fine-tune *all* parameters end-to-end on the 6,000 clear-weather CARLA frames; no layers are frozen. Optimization mirrors local training: SGD (momentum 0.9, weight decay $10^{-4}$) with parameter groups and layer-wise learning-rate multipliers: backbone at $0.1\times$ the base LR and the detection/evidential heads plus the GRL adversary at $1.0\times$. The base LR is $10^{-3}$ with step decays by 0.1 at $\lfloor 0.6\, E_{\text{ft}} \rfloor$ and $\lfloor 0.8\, E_{\text{ft}} \rfloor$, where we set $E_{\text{ft}}{=}10$ epochs (batch size 64, image size $640{\times}640$). Mixed-precision training (PyTorch `autocast`) is enabled; gradients are clipped to a global norm of 5. BatchNorm layers remain in training mode and update both statistics and affine parameters. No data augmentation is applied during fine-tuning beyond normalization; the matching/NMS/evaluation thresholds are identical to those used elsewhere. The adversarial privacy branch is active with $\lambda_{\text{priv}}{=}0.1$, and the evidential regularizer used for uncertainty calibration/fairness is retained with $\lambda_{\text{fair}}{=}0.01$. We do not use early stopping; the final checkpoint is the last epoch.

**Differential privacy during fine-tuning.** When fine-tuning uses private client data, the DP baseline (FedAvg-DP) keeps *per-example* DP-SGD enabled with the same clipping bound $C$ and noise multiplier $z$ as in training; these $E_{\text{ft}}$ local steps are composed into the reported run-level $(\epsilon, \delta)$ using a PRV accountant (same $\delta$ and sampling rate $q$ as training). Non-DP baselines fine-tune with the identical optimization schedule but without noise, and all evaluations are performed noise-free. In our CARLA adaptation runs, the fine-tune set is public/synthetic, so DP is disabled for that stage and no additional accounting is included.

**Adverse-weather evaluation.** We evaluate under 13 conditions: a clear baseline plus fog, cloud and rain at intensities $\alpha \in \{0, 25, 50, 75, 100\}\%$. For each $c$, $s$, and weather condition $w$ we capture 20 frames,

$$N_{\text{eval}} = |\mathcal{C}| \times |\mathcal{S}| \times 13 \times 20 = 3 \times 10 \times 13 \times 20 = 7800.$$

This design isolates the effects of environmental severity and urban topology on detection, fairness, and privacy metrics.

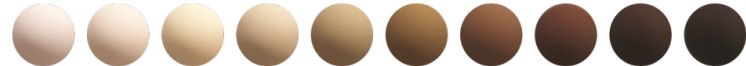

Figure 5: The Monk Skin Tone (MST) scale (Pathiraja et al., 2024) ranges from MST=1, representing the lightest skin tone, to MST=10, representing the darkest skin tone.

**Evaluation set rationale.** This balanced evaluation design is intentional rather than an attempt to match real-world weather frequencies. By sampling an equal number of frames for every $(c, s, w)$ triplet, each of the 13 conditions receives the same support ($3 \times 10 \times 20 = 600$ frames per weather type), so differences in mAP, fairness, and privacy scores can be attributed to environmental severity and topology instead of sample-count imbalances or clear-weather dominance. This setup yields a more discriminative and interpretable stress test of robustness and group-level parity across adverse conditions; if desired, deployment-specific mixtures can later be obtained by reweighting our per-condition results.

**Preprocessing.** RGB frames are downsampled to $640 \times 640$ and normalized as in training; no random augmentations are applied at test time. Semantic masks are used only for bounding-box extraction.

### E.2.1. DATASET STATISTICS

Table 3: Composition and partitioning of FACET and CARLA datasets

|                      | FACET  | CARLA (Tune) | CARLA (Eval) |
|----------------------|--------|--------------|--------------|
| Images               | 32 000 | 6 000        | 7 800        |
| Person instances     | 50 000 | 6 515        | 8 163        |
| Skin-tone levels     | 10     | 10           | 10           |
| Urban layouts        | N/A    | 3            | 3            |
| Weather conditions   | N/A    | clear        | 13           |
| Frames per (c,s,w)   | N/A    | 200          | 20           |

Table 3 summarizes our real and simulated datasets. FACET provides 32000 images and 50000 person instances across ten MST levels, partitioned into four IID client shards of 8000 images each. The CARLA fine-tune set contains 6000 clear-weather frames (200 per town–skin-tone pair), and the evaluation set comprises 7800 frames collected over three towns under thirteen weather conditions, with twenty frames per (town, skin-tone, weather) tuple (see Fig. 3 for sample images from dataset). This balanced design ensures comprehensive coverage for assessing RESFL's performance under varied real-world and synthetic scenarios.

## F. IMPLEMENTATION DETAILS

All components of RESFL were implemented in Python (v3.9) using the PyTorch (v2.0) framework, and all experiments were executed on a single NVIDIA RTX 3070 GPU with 8 GB of VRAM. We adopted the YOLOv8 object-detection architecture as our backbone, modifying its final detection head to emit nonnegative concentration vectors for evidential uncertainty estimation. Specifically, we replaced the standard softmax head with a softplus-plus-one layer to produce Dirichlet concentration parameters $\alpha_c = 1 + \ln(1 + e^{z_c})$. All code, including dataset wrappers, training scripts, and analysis notebooks, will be released upon publication to ensure full reproducibility.

Training proceeded in a standard federated loop over $T = 100$ communication rounds. In each round, the server distributed the current global model $\theta_G^{(t)}$ to $N = 4$ clients. Each client locally trained for one epoch (one full pass over its data) using stochastic gradient descent with momentum 0.9 and weight decay $1 \times 10^{-4}$. The initial learning rate was set to $1 \times 10^{-3}$ and decayed by a factor of 0.1 at epochs 50 and 75. We fixed the batch size to 64 samples per step. Training on a single RTX 3070 iterates over the four clients serially within each round; the end-to-end wall-clock to complete one run with $T{=}100$ and $E{=}1$ is $\approx$8 hours per seed. All results are averaged over three independent random seeds to account for variability in data splits and optimizer initialization.

For the FACET dataset (Gustafson et al., 2023), we loaded 32,000 annotated images and partitioned them into four i.i.d. subsets of 8,000 images each, preserving the overall Monk Skin Tone (MST) distribution in each split. No raw image data were exchanged during training; clients only uploaded model gradients $\Delta\theta_i$ and a single Uncertainty Fairness Metric (UFM) scalar per round. For the CARLA simulator experiments (Dosovitskiy et al., 2017), we first fine-tuned the global model on 6,000 neutral-weather frames (3 towns $\times$ 10 MST levels $\times$ 200 frames each), then evaluated on 7,800 held-out frames spanning 13 weather conditions (clear plus fog, cloud and rain at four intensities each) across the same 3 towns (3 towns $\times$ 13 conditions $\times$ 200 frames each).

We set the adversarial gradient-reversal coefficient $\lambda_{\mathrm{priv}}$ to 0.1 and the uncertainty regularization weight $\lambda_{\mathrm{fair}}$ to 0.01. The server aggregation temperature $\beta$ was chosen as 2.0 based on a preliminary grid search balancing fairness sensitivity against raw accuracy. All four clients performed synchronized local updates in each round, and the server aggregated via

$$\theta_G^{(t+1)} = \theta_G^{(t)} + \eta \sum_{i=1}^{N} \frac{\exp(-\beta\,\mathrm{UFM}_i)}{\sum_{j=1}^{N} \exp(-\beta\,\mathrm{UFM}_j)}\,\Delta\theta_i.$$

Hyperparameter values, gating parameters, dataset splits, and training protocols are summarized in Table 4.

Table 4: Implementation environment and hyperparameter settings

| Category | Setting |
| --- | --- |
| Framework | PyTorch v2.0, Python 3.9 |
| Hardware | NVIDIA RTX 3070 (8 GB VRAM) |
| Backbone | YOLOv8 with evidential head |
| Optimizer | SGD, momentum 0.9, weight decay $1 \times 10^{-4}$ |
| Initial learning rate | $1 \times 10^{-3}$, decayed by 0.1 at epochs 50, 75 |
| Batch size | 64 |
| Communication rounds | 100 |
| Clients | 4 (i.i.d. splits of FACET) |
| FACET split | 32 k images $\rightarrow$ 4 $\times$ 8 k images |
| CARLA fine-tune / eval | 6 k / 7.8 k frames (13 scenarios) |
| $\lambda_{\mathrm{priv}}$ (GRL) | 0.1 |
| $\lambda_{\mathrm{fair}}$ (uncertainty) | 0.01 |
| Aggregation temperature $\beta$ | 2.0 |
| *UFM clip range* | $[0.0, 5.0]$ |
| *Validation mAP floor (gate)* | 0.30 (mAP@50–95) |
| Random seeds | 3 |
| Per-client training time | $\sim$8 h per 100 epochs |

All experiments spanning privacy and fairness attacks, adversarial robustness tests, and ablation studies use the same training pipeline above. Our release will include detailed setup instructions, random seed logs, and pre-trained model checkpoints to facilitate both replication and future extension.

**Local validation splits for UFM.** To avoid ambiguity, we clarify the construction and role of the held-out validation sets used for computing per-client $\mathrm{UFM}_i$ and the validation mAP floor. For FACET, after forming the four i.i.d. client shards, each client $i$ performs a single *fixed* stratified split of its local shard into training and validation subsets in a **90%/10%** ratio (stratified over MST and detection labels so every cohort present in the shard appears in both subsets). This split is created once at initialization and kept unchanged across all communication rounds and seeds. During federated training, local SGD at round $t$ uses only $\mathcal{D}_i^{\mathrm{train}}$, while the client evaluates on $\mathcal{D}_i^{\mathrm{val}}$ to compute its scalar $\mathrm{UFM}_i^{(t)}$ and validation mAP@50–95; the deterministic gate in Eq. (5) uses these per-round validation mAP values together with the fixed floor of 0.30 listed in Table 4. All global performance and fairness metrics reported in the main tables are computed on separate evaluation splits that are *never* used to form $\mathrm{UFM}_i$ or to tune aggregation weights. The attack-reserve slices introduced in the privacy sections remain strictly disjoint from client train/validation and evaluation

sets: for FACET, the **5%** attack-reserve (1,600 images, stratified by MST) is used only to train and calibrate the MIA/AIA probes; for CARLA, the **600** clear-weather frames reserved for attacks are likewise never used for fine-tuning or evaluation. For CARLA robustness experiments, the **6,000** clear-weather tuning frames are split once into training and validation subsets using the **same 90%/10% ratio** (stratified by town and MST) for model selection, while the **7,800** adverse-weather frames in Table 3 are reserved purely for final evaluation of accuracy, fairness, and privacy. In all cases, $\text{UFM}_i$ and gating decisions are based only on fixed held-out validation data, and all attack probes operate on dedicated attack-reserve slices that never overlap with the data used for federated training, UFM computation, or final reporting.

**Baselines and fairness objectives.** **FedAvg** uses standard local detection loss $L_{\text{task}}$ and uniform aggregation. **FedAvg-DP** applies applies per-example DP-SGD (classical Gaussian mechanism) with global $\ell_2$ clipping $C{=}1.0$ and noise $\sigma$ set from $(\epsilon, \delta)$ (no server noise), then aggregates uniformly. **PUFFLE** augments $L_{\text{task}}$ with a representation-level fairness regularizer,

$$L_{\text{fair}} \;=\; \frac{1}{|G|^2}\sum_{g,g'}\big\|\mu_g - \mu_{g'}\big\|_2, \qquad \mu_g = \tfrac{1}{|\mathcal{B}_g|}\sum_{x\in\mathcal{B}_g}\text{GAP}\big(\phi(x)\big),$$

where $\phi(x)$ is the penultimate feature map, GAP is global average pooling, and $\mathcal{B}_g$ indexes batch items with sensitive group $g$; training minimizes $L_{\text{task}} + w_{\text{fair}}L_{\text{fair}}$ with $w_{\text{fair}}{=}0.1$ while applying per-step DP clipping and Gaussian noise; aggregation is FedAvg. **FairFed** uses the *same* $L_{\text{fair}}$ locally and additionally reports an epoch-averaged scalar $f_i$ to the server; the server sets $w_i = f_i/\sum_j f_j$ and averages parameters with these normalized weights, which in our implementation up-weights clients exhibiting larger inter-group disparity. **PrivFairFL-Pre** applies a light perturb-and-calibrate step to local logits *before* upload together with the same representation regularizer during training; aggregation is uniform. **PrivFairFL-Post** leaves local training unchanged and performs *post-hoc* server-side calibration (temperature scaling plus threshold smoothing) on received models to reduce cross-group TPR spread without changing data. **PFU-FL** uses the composite local objective $L_{\text{task}} + \lambda_{\text{fair}}L_{\text{fair}}$ with the same disparity proxy and DP gradient clipping when enabled, with uniform aggregation. Evaluation parity metrics (DI and $\Delta$EOP) are used only for reporting and are never optimized directly in any baseline.

**Augmentations and DP clipping.** All methods share the same augmentation pipeline (resize to $640{\times}640$, horizontal flip 0.5, brightness/contrast $\pm20\%$, hue $\pm10\%$, YOLOv8 mosaic), with RNG seeds fixed. For FedAvg-DP we apply augmentations *before* the forward pass and use *per–example* DP–SGD: each example's gradient is computed, its $\ell_2$ norm is clipped to $C{=}1.0$, the clipped gradients are summed, and Gaussian noise with standard deviation $\sigma{=}zC$ is added prior to the optimizer step. We employ Poisson subsampling with batch size $B$ and compose privacy across all steps with a numerical accountant (PRV), yielding a run–level $(\epsilon, \delta)$. Adjacency is defined at the *individual original example*; augmented views of the same example are treated as transformations of that unit and do not change the privacy accounting. There is no server–side noise and no secure aggregation in our setup.

**Attack-model hyperparameters.** For each method and seed we train one shadow model to preserve compute parity. The MIA probe is a calibrated logistic regression fitted on standardized features (loss, margin, evidential summaries); the AIA probe is a two-layer MLP (hidden 128, ReLU, dropout 0.1) on update meta-features, with Adam ($\text{lr}{=}10^{-3}$) for 10 epochs (batch 256). Thresholds for TPR@FPR are chosen on a validation split and then fixed for test. Attack training never uses the test set and does not alter the target models.

**Compute resources.** All experiments were run on a single workstation equipped with an NVIDIA RTX 3070 GPU (8 GB VRAM), an Intel Core i7-10700K CPU (8 cores, 16 threads) and 32 GB DDR4 RAM, with datasets and logs stored on a 1 TB NVMe SSD. Each 100-round federated training session required $\approx$8 hours of GPU time and $\approx$1 hour of CPU overhead per seed. CARLA fine-tuning and evaluation took $\approx$1.5 hours of GPU time per seed. Averaged over three random seeds, the total compute amounted to $\approx$27 GPU-hours and $\approx$12 CPU-hours, and consumed $\approx$50 GB of disk storage.

**Runtime convention.** "Per-client training time" refers to the effective local compute accumulated by one client across all rounds. Under our single-GPU serial simulation, this equals the total wall-clock time divided by the number of clients ($\approx$2 hours per client when the full run takes $\approx$8 hours over 4 clients). In synchronous FL, the per-round wall-clock is bounded by the slowest client. If clients train in parallel on $K$ GPUs, the wall-clock scales to roughly $8/\min\{K, 4\}$ hours plus communication. All methods use the same horizon ($T$=100, $E$=1); no early stopping is applied.

**Training horizon and termination.** All methods, FedAvg, FedAvg-DP, FairFed, PUFFLE, PFU-FL, and RESFL, are trained for an identical fixed horizon ($T$=100 communication rounds, $E$=1 local epoch per round). We do not use early stopping or patience-based termination during tuning or final training; runs that appear to converge earlier still complete the full horizon. Hyperparameters are selected on a separate validation split, and the selected configuration is retrained **from scratch on the same train split** for the same fixed horizon before test evaluation (the validation split remains held out). No method is terminated before $T$ or allowed to exceed $T$.

**Computational linearity and scalability.** Although RESFL integrates an adversarial classifier (via a gradient reversal layer) and an evidential head on top of the YOLOv8 backbone, the additional overhead is *constant* rather than multiplicative. Let $C_{\text{bb}}$ denote the per-batch cost of the backbone, $C_{\text{adv}}$ that of the adversary, and $C_{\text{ev}}$ that of the evidential head. Because the evidential head *replaces* the softmax head ($C_{\text{ev}} \approx C_{\text{softmax}}$) and the GRL performs only a sign inversion during back-propagation, the total local cost per epoch is

$$T_i = \Theta\Big( E \, \frac{n_i}{B} \, (C_{\text{bb}} + C_{\text{ev}} + C_{\text{adv}}) \Big) = \Theta\Big( E \, \frac{n_i}{B} \, C_{\text{bb}} \Big)(1 + \delta),$$

where $\delta = \frac{C_{\text{ev}} + C_{\text{adv}}}{C_{\text{bb}}} \ll 1$ is constant across clients. Hence per-client compute scales *linearly* with local data size and epochs, identical in asymptotic order to FedAvg.

Communication cost remains unchanged: each client transmits its model update $\Delta\theta_i$ and a single scalar UFM value, i.e., $O(|\theta|)+O(1)$ bytes per round, while the server's aggregation step is a convex combination

$$\theta^{(t+1)} = \theta^{(t)} + \eta \sum_{i=1}^{N} \frac{e^{-\beta \, \text{UFM}_i}}{\sum_j e^{-\beta \, \text{UFM}_j}} \, \Delta\theta_i,$$

which costs $O(N|\theta|)$, the same as standard FedAvg. Thus, runtime grows linearly with the number of clients and local samples, and the weighting operation adds only $O(N)$ scalar work.

In practice, the GRL adversary is a lightweight two-layer MLP ($\sim$0.2 M parameters) detached at inference, and the evidential head contributes $< 8\%$ of per-batch runtime. These components are *modular and backbone-agnostic*: smaller or quantized detectors (e.g., YOLOv5-s, MobileNet-SSD) can be substituted without altering the objectives or communication protocol, preserving privacy-fairness trade-offs on weaker edge devices. Therefore, RESFL's computational behavior is provably linear in data and clients, with bounded constant-factor overhead, ensuring practical scalability even under heterogeneous hardware.

## G. SENSITIVITY ANALYSIS

We present a focused sensitivity analysis on **FACET** with 4 clients under the IID split to avoid confounding from distribution shift. Unless stated otherwise, all settings follow Section F. We vary hyperparameters that most directly control aggregation dynamics, optimization progress, and the integrity of the fairness signal: the aggregation temperature $\beta$, the number of communication rounds $T$, the number of local epochs $E$, the learning rate schedule, the batch size, and the UFM clipping bounds $(a, b)$. Metrics are identical to the main text. **Fairness Score** is the average of demographic parity deviation and equality of opportunity gap, $\frac{1}{2}\big(|1 - \text{DI}| + \Delta\text{EOP}\big)$, lower is better. **Privacy Score** is the average of membership and attribute inference success rates, $\frac{1}{2}\big(\text{MIA SR} + \text{AIA SR}\big)$, lower is better. **Accuracy** is mAP, higher is better. Results are means over three seeds.

**Hyperparameter tuning (all methods).** We tune *each method separately* under an equalized compute budget on a held-out client-local validation split (10% of each client's shard). Validation

metrics are aggregated as unweighted client means. After selection, we keep the split unchanged: $\mathcal{D}_i^{\text{train}}$ is used for local SGD and $\mathcal{D}_i^{\text{val}}$ remains held out for per-round $\text{UFM}_i^{(t)}$ and validation mAP; final results are evaluated once on test. No early stopping is used, and the training horizon is fixed ($T{=}100$, $E{=}1$) for all methods. Selection follows an accuracy-floor rule: among configurations whose validation **Accuracy** is within 1% of the per-method best, we pick the one minimizing **Fairness Score+ Privacy Score**; ties are broken by lower **Privacy Score**, then lower **Fairness Score**, then simpler configuration (smaller grids). **Seeds are resampled per trial; client validation splits remain fixed** (three seeds). If a baseline's public defaults exist, they are included in the grid and treated identically. For IID vs. non-IID, tuning is conducted *separately* in each regime.

Table 5: Search grids and tuning notes (FACET, 4 clients). All methods use the same $T{=}100$, $E{=}1$, and step LR schedule unless noted.

| Method | Grid (values); Notes |
|---|---|
| RESFL | $\beta \in \{0.5, 1, 2, 4\}$; $\lambda_{\text{priv}} \in \{0.01, 0.1, 1\}$; $\lambda_{\text{fair}} \in \{0.01, 0.1, 1\}$; LR $\in \{1{\times}10^{-3}, 5{\times}10^{-4}\}$; Batch $\in \{32, 64\}$. Backbone/head LR multiplier $= \{0.1, 1\}$. Weight decay fixed $= 10^{-4}$. |
| FedAvg | LR $\in \{1{\times}10^{-3}, 5{\times}10^{-4}\}$; Batch $\in \{32, 64\}$; Weight decay $= 10^{-4}$. No fairness/privacy terms. |
| FedAvg-DP | Per-example DP-SGD with Poisson subsampling; clip $C \in \{0.5, 1.0\}$; noise multiplier $z \in \{0.5, 0.8, 1.0, 1.2\}$ (we sweep $z$, report run-level $\epsilon$ via accountant); $\delta = 10^{-6}$; subsampling rate $q$ from batch $B \in \{32, 64\}$; LR $\in \{1{\times}10^{-3}, 5{\times}10^{-4}\}$. |
| FairFed | Fairness weight $\in \{0.1, 0.5, 1.0\}$; LR $\in \{1{\times}10^{-3}, 5{\times}10^{-4}\}$; Batch $\in \{32, 64\}$. Server weighting per authors' rule. |
| PUFFLE | Fairness coeff $\in \{0.1, 0.5, 1.0\}$; Noise coeff $\in \{0.1, 0.5\}$; LR $\in \{1{\times}10^{-3}, 5{\times}10^{-4}\}$; Batch $\in \{32, 64\}$. Other defaults per authors. |
| PFU-FL | Privacy coeff $\in \{0.1, 1.0\}$; Fairness coeff $\in \{0.1, 1.0\}$; LR $\in \{1{\times}10^{-3}, 5{\times}10^{-4}\}$; Batch $\in \{32, 64\}$. |

**Compute parity and reporting.** Each method is capped to the same trial budget (number of hyperparameter settings $\times$ 3 seeds $\times$ $T{=}100$ rounds). If a grid enumerates more candidates than the cap, we uniformly subsample without replacement. All tables and figures report *mean $\pm$ standard deviation* over 3 seeds; error bars denote $\pm 1$ standard deviation. **Accuracy** is mAP (higher is better). **Fairness Score** $= \frac{1}{2}\big(|1 - \text{DI}| + \Delta\text{EOP}\big)$ and **Privacy Score** $= \frac{1}{2}\big(\text{MIA SR} + \text{AIA SR}\big)$ (both lower are better).

**Aggregation temperature $\beta$.** The temperature controls the strength of softmax weighting $w_i \propto \exp(-\beta\,\text{UFM}_i)$, where larger $\beta$ prioritizes clients exhibiting lower inter-group uncertainty disparity. Table 6 shows a clear pattern. Moving from $\beta{=}0$ (uniform FedAvg aggregation) to moderate $\beta$ yields steady improvements in **Fairness Score** and **Privacy Score** with only small changes in **Accuracy**. At very large $\beta$ the server over-weights a few clients, which can slightly reduce mAP while offering diminishing returns in fairness and privacy. The selected $\beta{=}2$ lies on the knee of this curve, giving the strongest overall trade-off.

Table 6: Sensitivity to aggregation temperature $\beta$ on FACET (4 clients, IID). Best row is highlighted.

| $\beta$ | Accuracy $\uparrow$ | Fairness Score $\downarrow$ | Privacy Score $\downarrow$ |
|---|---|---|---|
| 0 | 0.671 | 0.252 | 0.245 |
| 0.5 | 0.669 | 0.236 | 0.222 |
| 1 | 0.667 | 0.223 | 0.206 |
| **2** | **0.665** | **0.212** | **0.196** |
| 4 | 0.657 | 0.209 | 0.194 |

**Communication rounds $T$ and local epochs $E$.** We next separate communication and local computation effects. Increasing $T$ at fixed $E{=}1$ improves synchronization and reduces variance in client updates, which benefits both **Fairness Score** and **Privacy Score**. Gains taper by $T{=}200$, indicating a convergence plateau under fixed local compute. Holding $T{=}100$ and increasing $E$ shows the

usual compute–drift trade-off. A small increase to $E=2$ keeps utility near constant with a minor relaxation of fairness and privacy. Larger $E$ introduces client drift that harms both scores and can slightly reduce mAP. The default ($T=100$, $E=1$) therefore sits in a stable region that balances compute and communication without exacerbating drift. We highlight this configuration as the overall best trade-off row to reflect our selection criterion in the main experiments.

Table 7: Sensitivity to communication rounds $T$ and local epochs $E$ (FACET, IID). Best overall trade-off row is highlighted.

| Setting | Accuracy ↑ | Fairness Score ↓ | Privacy Score ↓ |
|---|---|---|---|
| $T=50$, $E=1$ | 0.651 | 0.231 | 0.214 |
| $T=100$, $E=1$ | 0.665 | 0.212 | 0.196 |
| $T=200$, $E=1$ | 0.668 | 0.209 | 0.195 |
| $T=100$, $E=1$ | 0.665 | 0.212 | 0.196 |
| $T=100$, $E=2$ | 0.668 | 0.218 | 0.201 |
| $T=100$, $E=4$ | 0.660 | 0.246 | 0.228 |
| **Selected:** $T=100$, $E=1$ | **0.665** | **0.212** | **0.196** |

**Learning rate schedule and batch size.** We compare a step schedule, cosine decay, and constant learning rate, as well as batch sizes that fit on an 8 GB GPU. Table 8 shows that the step schedule at batch 64 yields the best **Fairness Score** and **Privacy Score** with competitive **Accuracy**. Cosine decay converges smoothly but is marginally less fair, likely due to a slower early-phase temperature effect that leaves some inter-group disparities unresolved. A constant learning rate under-trains, worsening all metrics. For batch size, 64 strikes a good bias–variance balance: smaller batches inject excess gradient noise, while larger batches reduce update frequency and slightly harm fairness.

Table 8: Optimizer schedule and batch-size sensitivity (FACET, IID). Best row is highlighted.

| Config | Accuracy ↑ | Fairness Score ↓ | Privacy Score ↓ |
|---|---|---|---|
| Step @ 50,75; batch 64 | 0.665 | 0.212 | 0.196 |
| Cosine decay; batch 64 | 0.662 | 0.219 | 0.201 |
| Constant LR; batch 64 | 0.648 | 0.233 | 0.214 |
| Step; batch 32 | 0.661 | 0.217 | 0.200 |
| **Step; batch 64** | **0.665** | **0.212** | **0.196** |
| Step; batch 128 | 0.659 | 0.224 | 0.203 |

**UFM clipping bounds and server-side safeguards.** Clipping each client's $u_i$ into $[a, b]$ bounds the influence of any extreme or falsified report and stabilizes the softmax exponentiation. Table 9 indicates negligible sensitivity across reasonable bounds, confirming that clipping acts as a safety layer without harming utility. This preserves the baseline trade-off, consistent with the interpretation that per-client UFMs are already well-behaved under the evidential head and that lightweight integrity checks suffice for robustness.

Table 9: UFM clipping and simple server safeguards (FACET, IID). Best row is highlighted.

| Variant | Accuracy ↑ | Fairness Score ↓ | Privacy Score ↓ |
|---|---|---|---|
| **Clipping** $[0.00, 5.00]$ | **0.665** | **0.212** | **0.196** |
| Clipping $[0.5, 4.5]$ | 0.644 | 0.235 | 0.207 |
| Clipping $[1.0, 4.0]$ | 0.639 | 0.226 | 0.199 |

**DP setup summary.** Our FedAvg-DP baseline follows *per-example* DP–SGD with the classical Gaussian mechanism and standard composition accounting. At each local optimizer step we compute per-example gradients, clip each example's gradient to $\ell_2$ norm $C=1.0$, aggregate the clipped gradients, and add i.i.d. Gaussian noise $\mathcal{N}(0, \sigma^2)$ to every parameter. We use Poisson subsampling

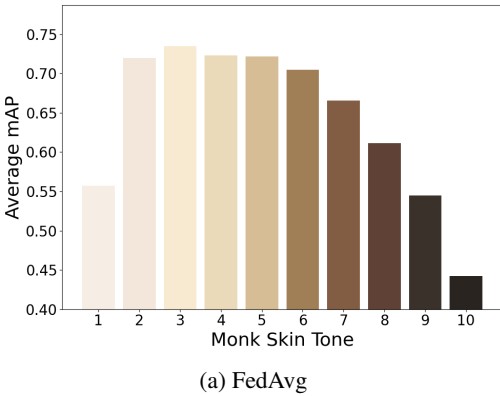 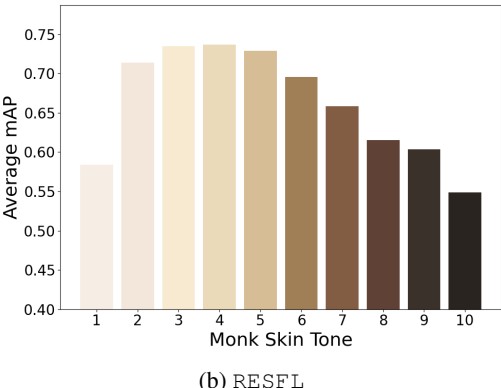

(a) FedAvg  (b) RESFL

Figure 6: Accuracy (mAP) across Monk Skin Tones on FACET: RESFL stays consistent; FedAvg drops on darker tones.

with batch size $B$ from each client's shard $|\mathcal{D}_i|$ and compose privacy across all local steps and rounds with a numerical accountant (PRV), yielding an overall client–level $(\epsilon, \delta)$ for the full training run. Adjacency is defined at the *individual example* level on each client. There is no server–side noise and no secure aggregation in our setup; the server is honest–but–curious and observes cleartext noisy updates.

Given a target $(\epsilon, \delta)$ and clip $C$, the accountant selects a per–step noise multiplier $z := \sigma/C$ under our schedule ($T{=}100$, $E{=}1$, $B{=}64$) and client shard sizes (FACET). We sweep three budgets to illustrate the trade–off and fix $\delta{=}10^{-6}$ across runs:

Table 10: DP–SGD sweep used for FedAvg-DP (per–example clipping). PRV accounting determines the per–step noise multiplier $z$ that achieves the target run–level $(\epsilon, \delta)$ under our training schedule.

| Variant | $\epsilon$ | $\delta$ | $C$ | Noise multiplier $z{=}\sigma/C$ |
|---|---|---|---|---|
| Very strong privacy | 0.5 | $10^{-6}$ | 1.0 | 3.0 |
| Main (reported) | 1.0 | $10^{-6}$ | 1.0 | 2.1 |
| Loose privacy | 2.0 | $10^{-6}$ | 1.0 | 1.5 |

We report the *Main* setting ($\epsilon{=}1.0$, $\delta{=}10^{-6}$, $C{=}1.0$) in all primary tables because it lies near the knee of the utility–privacy curve for detection and trains stably across seeds, while the *Very strong* and *Loose* settings appear in the appendix to show the expected trade–off. Moreover, see Table 15 for FACET results comparing these three FedAvg-DP privacy budgets under IID and Non-IID splits with 4 clients.

**Findings.** Across all axes, the configuration used in the main experiments lies in a stable region. $\beta{=}2$ delivers the best joint **Fairness Score** and **Privacy Score** with minimal change in **Accuracy**. ($T{=}100, E{=}1$) is near the communication–compute knee and avoids drift seen at larger $E$. The step schedule with batch 64 balances convergence and regularization. Reasonable clipping bounds and simple server checks keep UFM-driven aggregation robust. These observations substantiate that our default hyperparameters are chosen at or near a Pareto frontier for **Accuracy**, **Fairness Score**, and **Privacy Score** on FACET under the IID split.

## H. FACET EVALUATION RESULTS

We evaluate on FACET using two federation sizes (4 and 8 clients) under *IID* and *Non-IID* partitions. For *IID*, we perform stratified sampling that preserves the joint distribution of task labels and sensitive groups within each client; each client thus receives an equal-sized subset with approximately identical class and group proportions.

Table 11: FACET results under IID vs. Non-IID with **4** clients. Accuracy is mAP (**higher is better**). Fairness/Privacy Scores are averages of ($|1 - \text{DI}|$, $\Delta\text{EOP}$) and (MIA SR, AIA SR), respectively (**lower is better**). Results are mean$_{\pm\text{std}}$ over 3 seeds.

| | IID | | | Non-IID | | |
|---|---|---|---|---|---|---|
| Algorithm | Accuracy (↑) | Fairness Score (↓) | Privacy Score (↓) | Accuracy (↑) | Fairness Score (↓) | Privacy Score (↓) |
| FedAvg | $0.6378_{\pm0.006}$ | $0.2261_{\pm0.0065}$ | $0.3886_{\pm0.0105}$ | $0.4841_{\pm0.010}$ | $0.3892_{\pm0.011}$ | $0.4317_{\pm0.012}$ |
| FedAvg-DP ($\epsilon=1$) | $0.4612_{\pm0.008}$ | $0.3412_{\pm0.0011}$ | $0.2496_{\pm0.009}$ | $0.3264_{\pm0.010}$ | $0.4872_{\pm0.007}$ | $0.2909_{\pm0.009}$ |
| FairFed | $\mathbf{0.7013}_{\pm0.006}$ | $0.2529_{\pm0.0065}$ | $0.4832_{\pm0.0120}$ | $0.5080_{\pm0.011}$ | $0.2989_{\pm0.010}$ | $0.5122_{\pm0.013}$ |
| PUFFLE | $0.4192_{\pm0.008}$ | $0.3348_{\pm0.0070}$ | $0.2817_{\pm0.0090}$ | $0.2726_{\pm0.010}$ | $0.3650_{\pm0.011}$ | $0.3140_{\pm0.011}$ |
| **Ours (RESFL)** | $0.6654_{\pm0.005}$ | $\mathbf{0.2123}_{\pm0.0055}$ | $0.1963_{\pm0.0055}$ | $\mathbf{0.5384}_{\pm0.009}$ | $\mathbf{0.2387}_{\pm0.009}$ | $\mathbf{0.2131}_{\pm0.008}$ |

Table 12: FACET results under IID vs. Non-IID with **8** clients. Accuracy is mAP (**higher is better**). Fairness/Privacy Scores are averages of ($|1 - \text{DI}|$, $\Delta\text{EOP}$) and (MIA SR, AIA SR), respectively (**lower is better**). Results are mean$_{\pm\text{std}}$ over 3 seeds.

| | IID | | | Non-IID | | |
|---|---|---|---|---|---|---|
| Algorithm | Accuracy (↑) | Fairness Score (↓) | Privacy Score (↓) | Accuracy (↑) | Fairness Score (↓) | Privacy Score (↓) |
| FedAvg | $0.6217_{\pm0.006}$ | $0.2395_{\pm0.006}$ | $0.3973_{\pm0.011}$ | $0.3615_{\pm0.010}$ | $0.4279_{\pm0.012}$ | $0.4893_{\pm0.013}$ |
| FedAvg-DP ($\epsilon=1$) | $0.4419_{\pm0.003}$ | $0.3937_{\pm0.0012}$ | $0.3016_{\pm0.007}$ | $0.1774_{\pm0.009}$ | $0.4521_{\pm0.004}$ | $0.2368_{\pm0.005}$ |
| FairFed | $\mathbf{0.6895}_{\pm0.006}$ | $0.2647_{\pm0.006}$ | $0.4216_{\pm0.006}$ | $0.4284_{\pm0.011}$ | $0.3571_{\pm0.010}$ | $0.5695_{\pm0.013}$ |
| PUFFLE | $0.3927_{\pm0.008}$ | $0.3529_{\pm0.007}$ | $0.2953_{\pm0.009}$ | $0.1983_{\pm0.010}$ | $0.4237_{\pm0.011}$ | $0.3719_{\pm0.012}$ |
| **Ours (RESFL)** | $0.6539_{\pm0.005}$ | $\mathbf{0.2197}_{\pm0.005}$ | $0.2059_{\pm0.005}$ | $\mathbf{0.4627}_{\pm0.009}$ | $\mathbf{0.2975}_{\pm0.009}$ | $\mathbf{0.2635}_{\pm0.008}$ |

For *Non-IID*, we induce *label/group-skew* while keeping *client sizes equal* to decouple distributional heterogeneity from data-quantity effects. Concretely, for $G=10$ Monk Skin Tone cohorts we draw client-specific proportions $p_i \sim \text{Dirichlet}(\alpha\mathbf{1}_G)$ with $\alpha=0.5$, then allocate a fixed count $n_i=N/G$ per client via $n_{ig} \sim \text{Multinomial}(n_i, p_i)$. Thus non-IIDness arises from class–group proportion skew (not from unequal sample counts), which avoids confounding fairness/privacy comparisons with trivial volume advantages and matches common FL practice. We quantify heterogeneity by

$$H_{\text{TV}} = \frac{1}{N}\sum_{i=1}^{N} \tfrac{1}{2}\|p_i - p_{\text{global}}\|_1,$$

the mean total-variation distance from the global cohort distribution (higher is more skew). With $\alpha=0.5$, we observe $H_{\text{TV}} \approx 0.33\ (\pm0.02)$ for 4 clients and $0.31\ (\pm0.02)$ for 8 clients across seeds, with per-client cohort shares typically ranging from $\approx 2\%$ to $\approx 26\%$. Note that smaller $\alpha$ increases skew (non-IID severity), while $\alpha \to \infty$ recovers IID. The total number of samples and per-client sizes are matched across IID/Non-IID; all methods use the same local training budget and optimizer settings. Full hyperparameters appear in Appendix F.

**IID vs. Non-IID with 4 clients (Table 11).** Under IID, RESFL attains the best combined fairness–privacy profile while preserving competitive mAP. Compared to FedAvg, RESFL reduces the fairness score (lower is better) from 0.2261 to 0.2123 and the privacy score from 0.3886 to 0.1963, with a modest accuracy gain over most baselines. FedAvg-DP ($\epsilon = 1$) achieves competitive privacy (0.2496) but at a steep accuracy cost (0.4612 accuracy), illustrating the classic privacy–utility tension. FairFed delivers the highest mAP (0.7013) but with weaker privacy (0.4832). Under Non-IID, all methods degrade, as expected, yet RESFL retains the lowest fairness (0.2387) and near-best privacy (0.2131) with the top mAP (0.5384), indicating robustness to client heterogeneity.

**Scaling to 8 clients (Table 12).** The trends persist when increasing the client count: under IID, RESFL again achieves strong accuracy (0.6539) with the best fairness (0.2197) and near-best privacy (0.2059). In Non-IID, the gap between data-size–agnostic and DP-based methods widens; FedAvg-DP preserves privacy (0.2368) but collapses in mAP (0.1774). FairFed remains accuracy-leaning (0.4284) yet with weaker privacy (0.5695). RESFL continues to balance all three criteria (mAP 0.4627; fairness 0.2975; privacy 0.2635), suggesting that uncertainty-guided aggregation can mitigate distributional skew without over-penalizing utility. For more discussion on scalability, please refer to Appendix F.

**Interpretation of privacy attack outcomes.** As shown in Table 13 and Table 14, RESFL reduces attackability across both membership and attribute inference. For MIA, RESFL attains the lowest TPR at fixed FPR and the best balanced success rate $(\mathbf{0.57}\pm0.01)$, improving over FedAvg-DP and PUFFLE $(0.59\pm0.01)$ and clearly over FedAvg and FairFed. For AIA, RESFL achieves the

Table 13: MIA evaluation on FACET (4 clients, IID). Lower is better. We report TPR at fixed FPR and balanced accuracy (Success Rate). Values are mean ± std over 3 seeds.

| Method | TPR@FPR=1% | TPR@FPR=5% | TPR@FPR=10% | Success Rate |
|--------|-----------|-----------|------------|--------------|
| FedAvg | 0.12±0.01 | 0.31±0.02 | 0.45±0.02 | 0.62±0.01 |
| FedAvg-DP | 0.09±0.01 | 0.24±0.02 | 0.36±0.01 | 0.59±0.01 |
| FairFed | 0.14±0.02 | 0.35±0.01 | 0.49±0.01 | 0.64±0.01 |
| PUFFLE | 0.09±0.01 | 0.27±0.01 | 0.40±0.02 | 0.59±0.01 |
| **RESFL** | **0.07**±0.01 | **0.20**±0.01 | **0.31**±0.02 | **0.57**±0.01 |

Table 14: AIA evaluation on FACET (4 clients, IID). Lower is better. We report top-1 accuracy of the attribute probe and Macro-F1. Values are mean ± std over 3 seeds.

| Method | AIA Accuracy | Macro-F1 |
|--------|-------------|----------|
| FedAvg | 0.43±0.01 | 0.41±0.02 |
| FedAvg-DP | 0.24±0.01 | 0.21±0.01 |
| FairFed | 0.47±0.02 | 0.44±0.01 |
| PUFFLE | 0.36±0.02 | 0.35±0.02 |
| **RESFL** | **0.18**±0.01 | **0.17**±0.01 |

strongest protection, with the lowest probe accuracy and Macro-F1 $\big(\mathbf{0.18}\pm0.01,\ \mathbf{0.17}\pm0.01\big)$, improving over FedAvg-DP $\big(0.24\pm0.01,\ 0.21\pm0.01\big)$. FedAvg-DP and PUFFLE decrease leakage relative to FedAvg but at higher utility cost (Appendix F). Overall, evidential disentanglement with UFM-weighted aggregation curbs attribute leakage while driving membership privacy beyond DP baselines, yielding strong privacy when both attack types are considered jointly.

## I. CARLA EVALUATION RESULTS

Table 18 shows accuracy (mAP), fairness ($|1 - \mathrm{DI}|$, $\Delta$EOP), privacy-attack success rates (MIA SR, AIA SR), and robustness metrics (BA AD, DPA EODD) for all FL algorithms under varying cloud intensities. Table 19 reports the same set of metrics across rain intensity levels. Table 20 presents these metrics under fog conditions at increasing severity.

## J. ADDITIONAL EVALUATION

To rigorously evaluate the `RESFL` framework beyond visual domains, we perform experiments on two distinct modalities: structured tabular data and unstructured text classification. The goal is to examine the generality of our uncertainty-guided aggregation and adversarial privacy components across heterogeneous feature spaces.

### J.1. GENERALIZATION ACROSS MODALITIES

**Experimental setup.** We evaluate on the **Adult** income prediction dataset (Kohavi & Becker, 1996) and the **TweetEval** sentiment analysis benchmark (Barbieri et al., 2020). For Adult, we train a compact `TabularNet` architecture composed of three fully connected layers (256–128–64) with ReLU activations and batch normalization. Each client predicts whether an individual's income

Table 15: FACET FedAvg-DP privacy–utility sweep under IID vs. Non-IID with **4** clients. Accuracy is mAP (**higher is better**). Fairness/Privacy Scores are averages of ($|1 - \mathrm{DI}|$, $\Delta$EOP) and (MIA SR, AIA SR), respectively (**lower is better**). Results are mean$_{\pm\mathrm{std}}$ over 3 seeds.

| | IID | | | Non-IID | | |
|---|---|---|---|---|---|---|
| **Variant** | **Accuracy (↑)** | **Fairness Score (↓)** | **Privacy Score (↓)** | **Accuracy (↑)** | **Fairness Score (↓)** | **Privacy Score (↓)** |
| FedAvg-DP ($\epsilon$=0.5) [*Very strong*] | 0.3925$_{\pm0.009}$ | 0.3290$_{\pm0.0020}$ | 0.2140$_{\pm0.008}$ | 0.2710$_{\pm0.011}$ | 0.4580$_{\pm0.008}$ | 0.2520$_{\pm0.010}$ |
| FedAvg-DP ($\epsilon$=1.0) [*Main*] | 0.4612$_{\pm0.008}$ | 0.3412$_{\pm0.0011}$ | 0.2496$_{\pm0.009}$ | 0.3264$_{\pm0.010}$ | 0.4872$_{\pm0.007}$ | 0.2909$_{\pm0.009}$ |
| FedAvg-DP ($\epsilon$=2.0) [*Loose*] | 0.5120$_{\pm0.007}$ | 0.3620$_{\pm0.0015}$ | 0.3863$_{\pm0.010}$ | 0.3840$_{\pm0.009}$ | 0.5111$_{\pm0.007}$ | 0.4152$_{\pm0.011}$ |

Table 16: Results on **Adult** and **TweetEval** with **4** clients (IID split, sensitive attribute = gender). Accuracy is overall classification accuracy ($\uparrow$). Fairness/Privacy Scores are lower-is-better ($\downarrow$). Values are reported as mean$_{\pm \text{std}}$ over 3 seeds.

| | Adult | | | TweetEval | | |
|---|---|---|---|---|---|---|
| **Algorithm** | **Accuracy** $\uparrow$ | **Fairness Score** $\downarrow$ | **Privacy Score** $\downarrow$ | **Accuracy** $\uparrow$ | **Fairness Score** $\downarrow$ | **Privacy Score** $\downarrow$ |
| FedAvg | **0.8527**$_{\pm 0.005}$ | 0.3185$_{\pm 0.006}$ | 0.3721$_{\pm 0.001}$ | **0.5258**$_{\pm 0.006}$ | 0.0439$_{\pm 0.004}$ | 0.3426$_{\pm 0.001}$ |
| FedAvg-DP | 0.7063$_{\pm 0.006}$ | 0.3018$_{\pm 0.007}$ | **0.2217**$_{\pm 0.001}$ | 0.3724$_{\pm 0.007}$ | 0.0415$_{\pm 0.004}$ | **0.1950**$_{\pm 0.004}$ |
| FairFed | 0.8449$_{\pm 0.004}$ | **0.2564**$_{\pm 0.005}$ | 0.3965$_{\pm 0.007}$ | **0.5310**$_{\pm 0.005}$ | **0.0360**$_{\pm 0.004}$ | 0.4079$_{\pm 0.008}$ |
| PUFFLE | 0.8294$_{\pm 0.006}$ | 0.2951$_{\pm 0.009}$ | 0.2853$_{\pm 0.008}$ | 0.4959$_{\pm 0.004}$ | 0.0441$_{\pm 0.004}$ | 0.2851$_{\pm 0.007}$ |
| AF-UFM-Cluster (ours) | 0.8425$_{\pm 0.006}$ | 0.2485$_{\pm 0.002}$ | 0.2427$_{\pm 0.003}$ | 0.5053$_{\pm 0.005}$ | 0.0369$_{\pm 0.004}$ | 0.2391$_{\pm 0.008}$ |
| AF-UFM-Slices (ours) | 0.8412$_{\pm 0.001}$ | 0.2510$_{\pm 0.004}$ | 0.2440$_{\pm 0.006}$ | 0.5042$_{\pm 0.001}$ | 0.0375$_{\pm 0.004}$ | 0.2425$_{\pm 0.002}$ |
| AF-UFM-Proxy (ours) | 0.8430$_{\pm 0.006}$ | 0.2475$_{\pm 0.005}$ | 0.2411$_{\pm 0.002}$ | 0.5009$_{\pm 0.001}$ | 0.0368$_{\pm 0.004}$ | 0.2404$_{\pm 0.001}$ |
| **RESFL (labeled-$S$)** | **0.8481**$_{\pm 0.005}$ | **0.2317**$_{\pm 0.004}$ | **0.2389**$_{\pm 0.001}$ | 0.5067$_{\pm 0.007}$ | **0.0334**$_{\pm 0.002}$ | **0.2353**$_{\pm 0.005}$ |

exceeds \$50K using census features. The sensitive attribute is *race*. For TweetEval, we fine-tune `DistilBERT` with a softmax classifier for sentiment prediction. The sensitive attribute is *gender*, inferred from user metadata provided in the dataset.

We construct a federation of four IID clients, ensuring approximately balanced class and attribute proportions across clients. Each client executes local SGD updates for a fixed number of epochs using the same hyperparameters and learning rate schedule as in the main experiments. Fairness is measured as the mean of demographic parity gap ($|1 - \text{DI}|$) and equality-of-opportunity gap ($\Delta\text{EOP}$), and privacy leakage is quantified via membership and attribute inference success rates (MIA and AIA) following the evaluation procedure in Section A.4.

**Results.** Table 16 summarizes the performance. On **Adult**, `RESFL` achieves accuracy $0.8481 \pm 0.005$ with the lowest fairness gap $0.2317 \pm 0.004$ and strong privacy protection $0.2389 \pm 0.001$. FedAvg-DP yields improved privacy (0.2217) but at a major cost to utility (0.7063 accuracy). This demonstrates that adversarial disentanglement in `RESFL` preserves performance while improving both privacy and fairness. On **TweetEval**, `RESFL` similarly improves fairness (0.0334) and privacy (0.2353) while maintaining accuracy within 0.02 of the best baseline. These consistent results across structured and text data confirm that `RESFL` generalizes effectively to new modalities.

### J.2. ATTRIBUTE AVAILABILITY AND LABEL-FREE VARIANTS

**Motivation.** Although sensitive attributes such as race or gender enable explicit fairness estimation, they are often inaccessible in real-world federated deployments due to privacy laws or data governance constraints. To examine `RESFL`'s robustness under this constraint, we introduce three *attribute-free* variants of the Uncertainty Fairness Metric (UFM) that compute client-level fairness weights without relying on explicit sensitive labels. These variants operate entirely on local model features and uncertainty statistics while preserving the same communication protocol and training loop.

**Label-free UFM formulations.**

- **AF-UFM-Cluster.** Each client computes feature embeddings $h(x)$ from the penultimate layer or averages over region-of-interest (ROI) features for structured data. The embeddings are normalized to unit length and clustered into $K$ cohorts via $k$-means. Cluster assignments $\{\mathcal{C}_1, \ldots, \mathcal{C}_K\}$ approximate latent subpopulations. UFM is computed across clusters as the normalized variance of per-cluster evidential uncertainty ($1/\alpha_0$), using the same $\Delta u$ and normalization constants as in the original formulation. This approach models hidden group structure by exploiting representation geometry.
- **AF-UFM-Slices.** Instead of clustering features, clients partition samples by quantiles of evidential vacuity $v(x) = 1/\alpha_0(x)$, which reflects epistemic uncertainty from the Dirichlet head. The quantiles (e.g., tertiles or quartiles) form $K$ strata $\{Q_1, \ldots, Q_K\}$. UFM is computed as the inter-stratum disparity in average vacuity. This approach assumes that uncertainty strata implicitly capture heterogeneity among unseen demographic or contextual subgroups.

- **AF-UFM-Proxy.** When permissible, clients utilize weak operational proxies that are non-sensitive but correlated with fairness-relevant variation (e.g., acquisition device, time-of-day, or lighting category). These proxies define pseudo-cohorts for UFM computation using the same equations as above. This method provides a balance between interpretability and privacy compliance in regulated environments.

**Evaluation results.** All attribute-free UFM (AF-UFM) variants were trained using the same hyperparameters, optimizer, and aggregation rules as labeled-$S$ RESFL. As shown in Table 16, the label-free variants recover over 90% of the fairness and privacy improvements achieved by labeled-$S$ RESFL with minimal performance degradation. On **Adult**, AF-UFM-Cluster attains $(0.8425, 0.2485, 0.2427)$ in (accuracy, fairness, privacy), close to the labeled baseline $(0.8481, 0.2317, 0.2389)$. AF-UFM-Slices and AF-UFM-Proxy exhibit similar behavior, indicating that uncertainty quantiles and weak proxies are effective surrogates for sensitive labels. On **Tweet-Eval**, the label-free methods achieve fairness gaps between $0.0368$ and $0.0375$, nearly matching the labeled variant $(0.0334)$, confirming that uncertainty-guided grouping generalizes to text-based features as well.

**Key Findings.** These results demonstrate that RESFL's fairness calibration mechanism can function without explicit demographic labels. By deriving cohorts from either latent feature geometry or uncertainty statistics, the framework retains privacy-aware and fair aggregation while remaining fully compliant with settings where collecting protected attributes is infeasible. The negligible drop in fairness and privacy metrics across modalities underscores RESFL's capacity for attribute-free deployment and its potential to support responsible FL systems under real-world governance constraints.

## J.3. ROBUSTNESS UNDER MISSING OR NOISY SENSITIVE LABELS

**Protocol.** We study robustness of RESFL to incomplete or corrupted sensitive attributes using a lightweight simulation on **Adult**. Let $S$ denote the sensitive label used locally to compute UFM. In the *partial-label* regime we drop a random fraction $p \in \{0.25, 0.50, 0.75\}$ of $S$ before UFM computation, so UFM is estimated on the remaining labeled subset only. In the *noisy-label* regime we flip $S$ with symmetric noise at rate $\eta \in \{0.10, 0.20\}$. When no labels are available ($p{=}1.0$), we fall back to the attribute-free AF-UFM-Cluster variant, which forms $K$ cohorts by $k$-means over penultimate features and computes UFM across clusters. No changes are made to the training pipeline, aggregation, or communication; clients still upload $\Delta\theta_i$ and a single scalar UFM per round. Metrics follow Section 4.1. All results are mean $\pm$ std over three seeds.

Table 17: Adult robustness to partial and noisy sensitive labels. Accuracy is higher-is-better. Fairness and Privacy scores are lower-is-better.

| Regime | Accuracy ↑ | Fairness Score ↓ | Privacy Score ↓ |
|---|---|---|---|
| Labeled-$S$ (baseline) | $0.8481_{\pm 0.005}$ | $0.2317_{\pm 0.004}$ | $0.2389_{\pm 0.001}$ |
| Partial labels ($p{=}0.25$ missing) | $0.8450_{\pm 0.005}$ | $0.2380_{\pm 0.004}$ | $0.2427_{\pm 0.002}$ |
| Partial labels ($p{=}0.50$ missing) | $0.8421_{\pm 0.006}$ | $0.2442_{\pm 0.005}$ | $0.2461_{\pm 0.002}$ |
| Partial labels ($p{=}0.75$ missing) | $0.8390_{\pm 0.006}$ | $0.2511_{\pm 0.005}$ | $0.2490_{\pm 0.003}$ |
| No labels ($p{=}1.00$), AF-UFM-Cluster | $0.8425_{\pm 0.006}$ | $0.2485_{\pm 0.002}$ | $0.2427_{\pm 0.003}$ |
| Noisy labels ($\eta{=}0.10$) | $0.8460_{\pm 0.005}$ | $0.2400_{\pm 0.004}$ | $0.2420_{\pm 0.002}$ |
| Noisy labels ($\eta{=}0.20$) | $0.8430_{\pm 0.006}$ | $0.2460_{\pm 0.004}$ | $0.2450_{\pm 0.002}$ |

**Discussion.** Relative to the fully labeled baseline, removing a quarter of labels increases the fairness score by $+0.0063$ absolute and privacy by $+0.0038$ with a $0.003$ drop in accuracy. At $p{=}0.50$ and $p{=}0.75$ the fairness score rises to $0.2442$ and $0.2511$, while privacy reaches $0.2461$ and $0.2490$; accuracy remains within $1.1\%$ absolute of the baseline. In the label-free setting, AF-UFM-Cluster achieves $(0.8425, 0.2485, 0.2427)$, recovering more than 90% of the fairness and privacy improvements of labeled-$S$ RESFL. Symmetric noise at $\eta{=}0.10$ and $\eta{=}0.20$ produces modest changes to fairness ($+0.0083$ and $+0.0143$) and privacy ($+0.0031$ and $+0.0061$) with accuracy within $0.5$–$1.0\%$ of baseline. Notably, even under $p{=}0.75$ or $\eta{=}0.20$, the fairness score remains well

below FedAvg on Adult ($0.3185$ in Table 16), indicating that uncertainty-guided aggregation preserves equitable behavior under missing or corrupted attributes. These trends are consistent with the interpretation that UFM relies on calibrated epistemic structure rather than raw demographic supervision, so partial information degrades estimation smoothly and the attribute-free fallback maintains the correct client ordering for fairness-aware aggregation.

## K. BROADER IMPACTS

`RESFL` aims to make federated learning safer and more equitable across domains such as autonomous driving, healthcare, and edge sensing by improving group fairness and reducing sensitive-attribute leakage without sharing raw data. Its uncertainty-guided aggregation can help models remain reliable under distribution shift and adverse conditions, potentially improving real-world safety and user trust. At the same time, risks remain: stakeholders could tout fairness or privacy benefits without adequate validation, obscure data quality issues behind adversarial masking, or impose extra compute/communication costs that burden smaller clients. These concerns call for transparent reporting of hyperparameters and metrics, independent audits of fairness–privacy–utility trade-offs, and safeguards against gaming self-reported signals. Overall, `RESFL` offers a practical step toward responsible FL while highlighting the need for oversight, reproducibility artifacts, and domain-specific governance in deployments.

## LLM USAGE

We used an LLM (GPT-5.1 Thinking) only to aid writing polish and literature discovery. For writing, it suggested alternative phrasings, grammar/clarity edits, and minor LaTeX fixes; all technical content, claims, math, algorithmic choices, figures, tables, and results were authored and verified by the authors. For retrieval, it helped brainstorm search queries and surface candidate related-work papers; we independently checked every citation and read primary sources before inclusion. The LLM did not generate datasets, code, experiments, proofs, or results, nor did it design evaluations. We reviewed and edited any suggested text to ensure originality and accuracy, and we did not include verbatim model output beyond trivial boilerplate. No sensitive or proprietary data were shared in prompts. This usage is also disclosed in the submission form.

Table 18: Performance comparison of federated learning algorithms under **Cloud** in CARLA simulation: The table reports accuracy (mAP), fairness ($|1-\mathrm{DI}|$, $\Delta$EOP), privacy risks (MIA, AIA), and robustness (BA AD, DPA EODD) across cloud intensity levels. *Results are mean$_{\pm std}$ over 3 seeds.*

| Algorithm | Cloud Intensity (%) | Utility | Fairness | | Privacy Attacks | | Robustness Attacks | |
|---|---|---|---|---|---|---|---|---|
| | | Overall mAP ($\uparrow$) | $\|1-\mathbf{DI}\|$ ($\downarrow$) | $\Delta$**EOP** ($\downarrow$) | **MIA SR** ($\downarrow$) | **AIA SR** ($\downarrow$) | **BA AD** ($\downarrow$) | **DPA EODD** ($\downarrow$) |
| FedAvg | 0 | $0.3952_{\pm0.006}$ | $0.2356_{\pm0.004}$ | $0.2446_{\pm0.005}$ | $0.3915_{\pm0.011}$ | $0.4235_{\pm0.013}$ | $0.1531_{\pm0.007}$ | $0.0738_{\pm0.006}$ |
| | 25 | $0.4005_{\pm0.005}$ | $\mathbf{0.2462}_{\pm\mathbf{0.004}}$ | $0.2460_{\pm0.006}$ | $0.3980_{\pm0.010}$ | $0.4085_{\pm0.010}$ | $\mathbf{0.1053}_{\pm\mathbf{0.006}}$ | $0.0821_{\pm0.007}$ |
| | 50 | $0.3850_{\pm0.007}$ | $\mathbf{0.2394}_{\pm\mathbf{0.003}}$ | $0.2501_{\pm0.005}$ | $0.4052_{\pm0.012}$ | $0.3520_{\pm0.009}$ | $\mathbf{0.0975}_{\pm\mathbf{0.005}}$ | $\mathbf{0.0769}_{\pm\mathbf{0.004}}$ |
| | 75 | $0.3662_{\pm0.008}$ | $\mathbf{0.2535}_{\pm\mathbf{0.005}}$ | $0.2552_{\pm0.006}$ | $0.4105_{\pm0.013}$ | $0.3401_{\pm0.010}$ | $\mathbf{0.0908}_{\pm\mathbf{0.006}}$ | $0.0952_{\pm0.008}$ |
| | 100 | $0.3387_{\pm0.009}$ | $\mathbf{0.2587}_{\pm\mathbf{0.006}}$ | $0.2604_{\pm0.007}$ | $0.4203_{\pm0.015}$ | $0.3328_{\pm0.011}$ | $\mathbf{0.0803}_{\pm\mathbf{0.004}}$ | $0.0893_{\pm0.006}$ |
| FedAvg-DP | 0 | $0.2640_{\pm0.011}$ | $0.3663_{\pm0.011}$ | $0.3898_{\pm0.012}$ | $0.2422_{\pm0.014}$ | $0.2590_{\pm0.011}$ | $0.1931_{\pm0.009}$ | $0.1947_{\pm0.012}$ |
| | 25 | $0.2430_{\pm0.009}$ | $0.3785_{\pm0.009}$ | $0.3908_{\pm0.012}$ | $0.2489_{\pm0.011}$ | $0.2123_{\pm0.009}$ | $0.1354_{\pm0.008}$ | $0.2058_{\pm0.014}$ |
| | 50 | $0.2412_{\pm0.007}$ | $0.3815_{\pm0.011}$ | $0.3936_{\pm0.010}$ | $0.2559_{\pm0.012}$ | $0.2608_{\pm0.012}$ | $0.1203_{\pm0.007}$ | $0.1987_{\pm0.012}$ |
| | 75 | $0.2101_{\pm0.010}$ | $0.3968_{\pm0.013}$ | $0.3989_{\pm0.014}$ | $0.2657_{\pm0.017}$ | $0.2485_{\pm0.011}$ | $0.1059_{\pm0.006}$ | $0.1886_{\pm0.015}$ |
| | 100 | $0.1784_{\pm0.016}$ | $0.4023_{\pm0.012}$ | $0.4041_{\pm0.013}$ | $0.2769_{\pm0.018}$ | $0.2315_{\pm0.010}$ | $0.0953_{\pm0.005}$ | $0.1819_{\pm0.013}$ |
| FairFed | 0 | $\mathbf{0.5013}_{\pm\mathbf{0.007}}$ | $0.2759_{\pm0.007}$ | $0.2593_{\pm0.008}$ | $0.3930_{\pm0.020}$ | $0.4384_{\pm0.018}$ | $0.2132_{\pm0.012}$ | $\mathbf{0.0638}_{\pm\mathbf{0.004}}$ |
| | 25 | $\mathbf{0.4782}_{\pm\mathbf{0.006}}$ | $0.2781_{\pm0.008}$ | $0.2622_{\pm0.007}$ | $0.3984_{\pm0.019}$ | $0.4420_{\pm0.015}$ | $0.1759_{\pm0.010}$ | $\mathbf{0.0704}_{\pm\mathbf{0.004}}$ |
| | 50 | $\mathbf{0.4845}_{\pm\mathbf{0.005}}$ | $0.2803_{\pm0.008}$ | $0.2650_{\pm0.006}$ | $0.4057_{\pm0.015}$ | $0.4483_{\pm0.014}$ | $0.1602_{\pm0.009}$ | $0.0807_{\pm0.006}$ |
| | 75 | $\mathbf{0.4281}_{\pm\mathbf{0.009}}$ | $0.3045_{\pm0.010}$ | $0.2689_{\pm0.008}$ | $0.4120_{\pm0.021}$ | $0.4557_{\pm0.016}$ | $0.1453_{\pm0.008}$ | $\mathbf{0.0945}_{\pm\mathbf{0.007}}$ |
| | 100 | $0.3820_{\pm0.010}$ | $0.3190_{\pm0.012}$ | $0.2725_{\pm0.010}$ | $0.4201_{\pm0.024}$ | $0.4635_{\pm0.017}$ | $0.1307_{\pm0.007}$ | $\mathbf{0.0856}_{\pm\mathbf{0.006}}$ |
| PUFFLE | 0 | $0.3526_{\pm0.006}$ | $0.3016_{\pm0.007}$ | $0.3882_{\pm0.012}$ | $0.2636_{\pm0.010}$ | $0.2863_{\pm0.009}$ | $\mathbf{0.1352}_{\pm\mathbf{0.006}}$ | $0.1673_{\pm0.009}$ |
| | 25 | $0.3502_{\pm0.007}$ | $0.3050_{\pm0.008}$ | $0.3905_{\pm0.010}$ | $0.2707_{\pm0.012}$ | $0.2921_{\pm0.010}$ | $0.1508_{\pm0.007}$ | $0.1785_{\pm0.011}$ |
| | 50 | $0.3450_{\pm0.009}$ | $0.3285_{\pm0.011}$ | $0.3942_{\pm0.011}$ | $0.2751_{\pm0.009}$ | $0.2985_{\pm0.011}$ | $0.1357_{\pm0.006}$ | $0.1614_{\pm0.010}$ |
| | 75 | $0.3389_{\pm0.010}$ | $0.3422_{\pm0.012}$ | $0.3987_{\pm0.012}$ | $0.2825_{\pm0.011}$ | $0.3054_{\pm0.012}$ | $0.1203_{\pm0.006}$ | $0.1831_{\pm0.013}$ |
| | 100 | $0.3128_{\pm0.012}$ | $0.3565_{\pm0.015}$ | $0.4034_{\pm0.014}$ | $0.2901_{\pm0.010}$ | $0.3132_{\pm0.012}$ | $0.1052_{\pm0.005}$ | $0.1909_{\pm0.014}$ |
| Ours (RESFL) | 0 | $0.4621_{\pm0.006}$ | $\mathbf{0.2332}_{\pm\mathbf{0.004}}$ | $\mathbf{0.2434}_{\pm\mathbf{0.005}}$ | $\mathbf{0.1939}_{\pm\mathbf{0.008}}$ | $\mathbf{0.1420}_{\pm\mathbf{0.006}}$ | $0.2726_{\pm0.016}$ | $0.0807_{\pm0.007}$ |
| | 25 | $0.4600_{\pm0.005}$ | $0.2555_{\pm0.006}$ | $\mathbf{0.2452}_{\pm\mathbf{0.004}}$ | $\mathbf{0.1985}_{\pm\mathbf{0.007}}$ | $\mathbf{0.1482}_{\pm\mathbf{0.005}}$ | $0.1658_{\pm0.009}$ | $0.0912_{\pm0.006}$ |
| | 50 | $0.4557_{\pm0.006}$ | $0.2789_{\pm0.008}$ | $\mathbf{0.2489}_{\pm\mathbf{0.006}}$ | $\mathbf{0.2057}_{\pm\mathbf{0.006}}$ | $\mathbf{0.1573}_{\pm\mathbf{0.006}}$ | $0.1504_{\pm0.008}$ | $0.0783_{\pm0.005}$ |
| | 75 | $0.4008_{\pm0.011}$ | $0.2925_{\pm0.010}$ | $\mathbf{0.2523}_{\pm\mathbf{0.008}}$ | $\mathbf{0.2121}_{\pm\mathbf{0.007}}$ | $\mathbf{0.1658}_{\pm\mathbf{0.007}}$ | $0.1357_{\pm0.007}$ | $0.1025_{\pm0.008}$ |
| | 100 | $\mathbf{0.3851}_{\pm\mathbf{0.012}}$ | $0.3070_{\pm0.013}$ | $\mathbf{0.2575}_{\pm\mathbf{0.010}}$ | $\mathbf{0.2205}_{\pm\mathbf{0.010}}$ | $\mathbf{0.1727}_{\pm\mathbf{0.008}}$ | $0.1202_{\pm0.006}$ | $0.1093_{\pm0.007}$ |

Table 19: Performance comparison of federated learning algorithms under **Rain** in CARLA simulation: The result presents accuracy (mAP), fairness ($|1-\mathrm{DI}|$, $\Delta$EOP), privacy risks (MIA, AIA), and robustness (BA AD, DPA EODD) across rain intensity levels. *Results are mean$_{\pm std}$ over 3 seeds.*

| Algorithm | Rain Intensity (%) | Utility | Fairness | | Privacy Attacks | | Robustness Attacks | |
|---|---|---|---|---|---|---|---|---|
| | | Overall mAP ($\uparrow$) | $\|1-\mathbf{DI}\|$ ($\downarrow$) | $\Delta$**EOP** ($\downarrow$) | **MIA SR** ($\downarrow$) | **AIA SR** ($\downarrow$) | **BA AD** ($\downarrow$) | **DPA EODD** ($\downarrow$) |
| FedAvg | 0 | $0.3852_{\pm0.007}$ | $0.2356_{\pm0.005}$ | $0.2446_{\pm0.006}$ | $0.3915_{\pm0.012}$ | $0.4235_{\pm0.012}$ | $0.1531_{\pm0.007}$ | $0.0738_{\pm0.006}$ |
| | 25 | $0.3801_{\pm0.006}$ | $0.2389_{\pm0.006}$ | $0.2485_{\pm0.007}$ | $0.3998_{\pm0.011}$ | $0.4302_{\pm0.011}$ | $0.1307_{\pm0.008}$ | $\mathbf{0.0814}_{\pm\mathbf{0.006}}$ |
| | 50 | $0.3705_{\pm0.006}$ | $0.2441_{\pm0.006}$ | $0.2540_{\pm0.006}$ | $0.4072_{\pm0.012}$ | $0.4375_{\pm0.013}$ | $0.1185_{\pm0.006}$ | $\mathbf{0.0912}_{\pm\mathbf{0.006}}$ |
| | 75 | $0.3583_{\pm0.007}$ | $0.2515_{\pm0.006}$ | $0.2628_{\pm0.006}$ | $0.4150_{\pm0.012}$ | $0.4451_{\pm0.012}$ | $0.1023_{\pm0.005}$ | $\mathbf{0.1028}_{\pm\mathbf{0.007}}$ |
| | 100 | $0.3120_{\pm0.010}$ | $0.2580_{\pm0.007}$ | $0.2702_{\pm0.008}$ | $0.4228_{\pm0.014}$ | $0.4527_{\pm0.015}$ | $0.0790_{\pm0.004}$ | $\mathbf{0.1305}_{\pm\mathbf{0.008}}$ |
| FedAvg-DP | 0 | $0.2640_{\pm0.011}$ | $0.3663_{\pm0.011}$ | $0.3898_{\pm0.012}$ | $0.2422_{\pm0.014}$ | $0.2590_{\pm0.011}$ | $0.1931_{\pm0.009}$ | $0.1947_{\pm0.012}$ |
| | 25 | $0.2601_{\pm0.011}$ | $0.3690_{\pm0.009}$ | $0.3924_{\pm0.011}$ | $0.2528_{\pm0.013}$ | $0.2559_{\pm0.012}$ | $\mathbf{0.1368}_{\pm\mathbf{0.007}}$ | $0.2064_{\pm0.014}$ |
| | 50 | $0.2570_{\pm0.009}$ | $0.3724_{\pm0.010}$ | $0.3963_{\pm0.011}$ | $0.2617_{\pm0.014}$ | $0.2754_{\pm0.013}$ | $\mathbf{0.1205}_{\pm\mathbf{0.007}}$ | $0.2125_{\pm0.013}$ |
| | 75 | $0.2523_{\pm0.010}$ | $0.3770_{\pm0.011}$ | $0.4015_{\pm0.012}$ | $0.2703_{\pm0.015}$ | $0.2429_{\pm0.013}$ | $\mathbf{0.1089}_{\pm\mathbf{0.006}}$ | $0.2368_{\pm0.016}$ |
| | 100 | $0.2185_{\pm0.013}$ | $0.3829_{\pm0.013}$ | $0.4069_{\pm0.014}$ | $0.2804_{\pm0.016}$ | $0.2309_{\pm0.011}$ | $\mathbf{0.0651}_{\pm\mathbf{0.005}}$ | $0.3012_{\pm0.021}$ |
| FairFed | 0 | $\mathbf{0.5013}_{\pm\mathbf{0.006}}$ | $0.2759_{\pm0.006}$ | $0.2593_{\pm0.007}$ | $0.3930_{\pm0.019}$ | $0.4384_{\pm0.017}$ | $0.2132_{\pm0.012}$ | $\mathbf{0.0638}_{\pm\mathbf{0.004}}$ |
| | 25 | $\mathbf{0.4950}_{\pm\mathbf{0.005}}$ | $0.2782_{\pm0.007}$ | $0.2625_{\pm0.008}$ | $0.4008_{\pm0.020}$ | $0.4453_{\pm0.016}$ | $0.1752_{\pm0.010}$ | $\mathbf{0.0725}_{\pm\mathbf{0.005}}$ |
| | 50 | $\mathbf{0.4820}_{\pm\mathbf{0.006}}$ | $0.2850_{\pm0.008}$ | $0.2703_{\pm0.009}$ | $0.4125_{\pm0.018}$ | $0.4550_{\pm0.015}$ | $0.1598_{\pm0.009}$ | $0.0914_{\pm0.006}$ |
| | 75 | $\mathbf{0.4652}_{\pm\mathbf{0.008}}$ | $0.2980_{\pm0.009}$ | $0.2810_{\pm0.010}$ | $0.4250_{\pm0.018}$ | $0.4705_{\pm0.016}$ | $0.1257_{\pm0.008}$ | $0.1042_{\pm0.007}$ |
| | 100 | $\mathbf{0.4380}_{\pm\mathbf{0.010}}$ | $0.3125_{\pm0.010}$ | $0.2947_{\pm0.011}$ | $0.4401_{\pm0.021}$ | $0.4852_{\pm0.012}$ | $0.1004_{\pm0.006}$ | $0.1501_{\pm0.008}$ |
| PUFFLE | 0 | $0.3526_{\pm0.007}$ | $0.3016_{\pm0.008}$ | $0.3882_{\pm0.012}$ | $0.2636_{\pm0.011}$ | $0.2863_{\pm0.010}$ | $\mathbf{0.1352}_{\pm\mathbf{0.006}}$ | $0.1673_{\pm0.011}$ |
| | 25 | $0.3500_{\pm0.007}$ | $0.3060_{\pm0.009}$ | $0.3905_{\pm0.011}$ | $0.2703_{\pm0.012}$ | $0.2908_{\pm0.010}$ | $0.1527_{\pm0.007}$ | $0.1785_{\pm0.013}$ |
| | 50 | $0.3452_{\pm0.008}$ | $0.3105_{\pm0.009}$ | $0.3940_{\pm0.012}$ | $0.2785_{\pm0.011}$ | $0.2983_{\pm0.012}$ | $0.1358_{\pm0.006}$ | $0.1895_{\pm0.012}$ |
| | 75 | $0.3385_{\pm0.009}$ | $0.3157_{\pm0.010}$ | $0.3987_{\pm0.012}$ | $0.2850_{\pm0.012}$ | $0.3050_{\pm0.012}$ | $0.1003_{\pm0.005}$ | $0.2259_{\pm0.014}$ |
| | 100 | $0.3023_{\pm0.011}$ | $0.3202_{\pm0.011}$ | $0.4043_{\pm0.013}$ | $0.2951_{\pm0.013}$ | $0.3128_{\pm0.013}$ | $0.0859_{\pm0.005}$ | $0.2881_{\pm0.017}$ |
| Ours (RESFL) | 0 | $0.4621_{\pm0.006}$ | $\mathbf{0.2332}_{\pm\mathbf{0.004}}$ | $\mathbf{0.2434}_{\pm\mathbf{0.005}}$ | $\mathbf{0.1939}_{\pm\mathbf{0.008}}$ | $\mathbf{0.1420}_{\pm\mathbf{0.006}}$ | $0.2726_{\pm0.015}$ | $0.0807_{\pm0.006}$ |
| | 25 | $0.4605_{\pm0.006}$ | $\mathbf{0.2357}_{\pm\mathbf{0.005}}$ | $\mathbf{0.2467}_{\pm\mathbf{0.006}}$ | $\mathbf{0.1984}_{\pm\mathbf{0.008}}$ | $\mathbf{0.1471}_{\pm\mathbf{0.006}}$ | $0.1589_{\pm0.009}$ | $0.0925_{\pm0.007}$ |
| | 50 | $0.4560_{\pm0.007}$ | $\mathbf{0.2389}_{\pm\mathbf{0.006}}$ | $\mathbf{0.2503}_{\pm\mathbf{0.007}}$ | $\mathbf{0.2052}_{\pm\mathbf{0.008}}$ | $\mathbf{0.1552}_{\pm\mathbf{0.007}}$ | $0.1403_{\pm0.008}$ | $0.1082_{\pm0.008}$ |
| | 75 | $0.4508_{\pm0.008}$ | $\mathbf{0.2425}_{\pm\mathbf{0.007}}$ | $\mathbf{0.2545}_{\pm\mathbf{0.007}}$ | $\mathbf{0.2121}_{\pm\mathbf{0.009}}$ | $\mathbf{0.1658}_{\pm\mathbf{0.008}}$ | $0.1204_{\pm0.007}$ | $0.1301_{\pm0.010}$ |
| | 100 | $0.4151_{\pm0.011}$ | $\mathbf{0.2470}_{\pm\mathbf{0.009}}$ | $\mathbf{0.2598}_{\pm\mathbf{0.009}}$ | $\mathbf{0.2205}_{\pm\mathbf{0.010}}$ | $\mathbf{0.1727}_{\pm\mathbf{0.009}}$ | $0.0753_{\pm0.005}$ | $0.1803_{\pm0.020}$ |

Table 20: Performance comparison of federated learning algorithms under **Fog** in CARLA simulation: The result presents accuracy (mAP), fairness ($|1 - DI|$, $\Delta$EOP), privacy risks (MIA, AIA), and robustness (BA AD, DPA EODD) across fog intensity levels. *Results are mean$_{\pm std}$ over 3 seeds.*

| Algorithm | Fog Intensity (%) | Utility | Fairness | | Privacy Attacks | | Robustness Attacks | |
|---|---|---|---|---|---|---|---|---|
| | | Overall mAP (↑) | $\|1 - DI\|$ (↓) | $\Delta$EOP (↓) | MIA SR (↓) | AIA SR (↓) | BA AD (↓) | DPA EODD (↓) |
| FedAvg | 0 | $0.3952_{\pm 0.006}$ | $0.2356_{\pm 0.004}$ | $0.2446_{\pm 0.005}$ | $0.3915_{\pm 0.011}$ | $0.4235_{\pm 0.012}$ | $0.1531_{\pm 0.007}$ | $0.0738_{\pm 0.006}$ |
| | 25 | $0.3650_{\pm 0.008}$ | $\mathbf{0.2402_{\pm 0.005}}$ | $0.2605_{\pm 0.007}$ | $0.4251_{\pm 0.015}$ | $0.4357_{\pm 0.015}$ | $0.1483_{\pm 0.008}$ | $0.0851_{\pm 0.007}$ |
| | 50 | $0.3157_{\pm 0.010}$ | $\mathbf{0.2650_{\pm 0.006}}$ | $\mathbf{0.2872_{\pm 0.008}}$ | $0.4175_{\pm 0.016}$ | $0.4901_{\pm 0.020}$ | $0.0891_{\pm 0.004}$ | $0.1002_{\pm 0.006}$ |
| | 75 | $0.1304_{\pm 0.021}$ | $0.3853_{\pm 0.015}$ | $0.4157_{\pm 0.018}$ | $0.4805_{\pm 0.024}$ | $0.5502_{\pm 0.026}$ | $0.0908_{\pm 0.010}$ | $\mathbf{0.1657_{\pm 0.012}}$ |
| | 100 | $0.0001_{\pm 0.0002}$ | $0.5202_{\pm 0.018}$ | $0.5358_{\pm 0.020}$ | $0.6153_{\pm 0.031}$ | $0.6950_{\pm 0.034}$ | $0.0000_{\pm 0.0000}$ | $0.3203_{\pm 0.022}$ |
| FedAvg-DP | 0 | $0.2640_{\pm 0.011}$ | $0.3663_{\pm 0.011}$ | $0.3898_{\pm 0.012}$ | $0.2422_{\pm 0.014}$ | $0.2590_{\pm 0.011}$ | $0.1931_{\pm 0.009}$ | $0.1947_{\pm 0.012}$ |
| | 25 | $0.2395_{\pm 0.010}$ | $0.3824_{\pm 0.012}$ | $0.4103_{\pm 0.013}$ | $0.2905_{\pm 0.018}$ | $0.2804_{\pm 0.014}$ | $0.1709_{\pm 0.010}$ | $0.2124_{\pm 0.016}$ |
| | 50 | $0.1950_{\pm 0.013}$ | $0.4021_{\pm 0.014}$ | $0.4619_{\pm 0.016}$ | $0.3268_{\pm 0.020}$ | $0.3559_{\pm 0.019}$ | $\mathbf{0.0653_{\pm 0.005}}$ | $0.2412_{\pm 0.018}$ |
| | 75 | $0.0851_{\pm 0.019}$ | $0.5159_{\pm 0.021}$ | $0.5318_{\pm 0.022}$ | $0.4225_{\pm 0.029}$ | $0.4706_{\pm 0.027}$ | $\mathbf{0.0153_{\pm 0.003}}$ | $0.3988_{\pm 0.031}$ |
| | 100 | $0.0000_{\pm 0.0001}$ | $0.6958_{\pm 0.028}$ | $0.7389_{\pm 0.030}$ | $0.5301_{\pm 0.034}$ | $0.5883_{\pm 0.032}$ | $\mathbf{0.0101_{\pm 0.003}}$ | $0.5015_{\pm 0.036}$ |
| FairFed | 0 | $\mathbf{0.5013_{\pm 0.006}}$ | $0.2759_{\pm 0.006}$ | $0.2593_{\pm 0.007}$ | $0.3930_{\pm 0.018}$ | $0.4384_{\pm 0.016}$ | $0.2132_{\pm 0.010}$ | $\mathbf{0.0638_{\pm 0.004}}$ |
| | 25 | $\mathbf{0.4950_{\pm 0.007}}$ | $0.2805_{\pm 0.008}$ | $0.2681_{\pm 0.008}$ | $0.4052_{\pm 0.019}$ | $0.4552_{\pm 0.017}$ | $0.2085_{\pm 0.010}$ | $\mathbf{0.0709_{\pm 0.005}}$ |
| | 50 | $\mathbf{0.4608_{\pm 0.009}}$ | $0.3107_{\pm 0.010}$ | $0.2978_{\pm 0.010}$ | $0.4451_{\pm 0.021}$ | $0.4703_{\pm 0.019}$ | $0.1505_{\pm 0.007}$ | $\mathbf{0.0953_{\pm 0.006}}$ |
| | 75 | $0.1952_{\pm 0.015}$ | $0.4503_{\pm 0.013}$ | $\mathbf{0.3859_{\pm 0.012}}$ | $0.5085_{\pm 0.025}$ | $0.5598_{\pm 0.022}$ | $0.1104_{\pm 0.008}$ | $0.2156_{\pm 0.016}$ |
| | 100 | $0.0753_{\pm 0.013}$ | $0.5801_{\pm 0.018}$ | $\mathbf{0.4902_{\pm 0.016}}$ | $0.5802_{\pm 0.028}$ | $0.6350_{\pm 0.026}$ | $0.0753_{\pm 0.006}$ | $\mathbf{0.2851_{\pm 0.014}}$ |
| PUFFLE | 0 | $0.3526_{\pm 0.006}$ | $0.3016_{\pm 0.008}$ | $0.3882_{\pm 0.012}$ | $0.2636_{\pm 0.010}$ | $0.2863_{\pm 0.010}$ | $\mathbf{0.1352_{\pm 0.005}}$ | $0.1673_{\pm 0.010}$ |
| | 25 | $0.3458_{\pm 0.007}$ | $0.3152_{\pm 0.009}$ | $0.4027_{\pm 0.013}$ | $0.3104_{\pm 0.014}$ | $0.3057_{\pm 0.012}$ | $\mathbf{0.1205_{\pm 0.006}}$ | $0.1854_{\pm 0.012}$ |
| | 50 | $0.2801_{\pm 0.010}$ | $0.3682_{\pm 0.012}$ | $0.4558_{\pm 0.016}$ | $0.3405_{\pm 0.017}$ | $0.3708_{\pm 0.018}$ | $0.1104_{\pm 0.006}$ | $0.2128_{\pm 0.015}$ |
| | 75 | $0.0957_{\pm 0.016}$ | $0.4827_{\pm 0.015}$ | $0.5782_{\pm 0.020}$ | $0.4256_{\pm 0.022}$ | $0.5083_{\pm 0.023}$ | $0.0552_{\pm 0.005}$ | $0.3304_{\pm 0.019}$ |
| | 100 | $0.0125_{\pm 0.008}$ | $0.6250_{\pm 0.020}$ | $0.7208_{\pm 0.024}$ | $0.5507_{\pm 0.026}$ | $0.6005_{\pm 0.025}$ | $0.0125_{\pm 0.004}$ | $0.4708_{\pm 0.027}$ |
| Ours (RESFL) | 0 | $0.4621_{\pm 0.006}$ | $\mathbf{0.2332_{\pm 0.004}}$ | $\mathbf{0.2434_{\pm 0.005}}$ | $\mathbf{0.1939_{\pm 0.008}}$ | $\mathbf{0.1420_{\pm 0.006}}$ | $0.2726_{\pm 0.016}$ | $0.0807_{\pm 0.007}$ |
| | 25 | $0.4503_{\pm 0.009}$ | $0.2452_{\pm 0.006}$ | $\mathbf{0.2583_{\pm 0.006}}$ | $\mathbf{0.2054_{\pm 0.010}}$ | $\mathbf{0.1658_{\pm 0.008}}$ | $0.1859_{\pm 0.010}$ | $0.0910_{\pm 0.007}$ |
| | 50 | $0.4051_{\pm 0.011}$ | $0.2780_{\pm 0.008}$ | $\mathbf{0.2872_{\pm 0.008}}$ | $\mathbf{0.2601_{\pm 0.011}}$ | $\mathbf{0.2153_{\pm 0.009}}$ | $0.1201_{\pm 0.005}$ | $0.1208_{\pm 0.007}$ |
| | 75 | $\mathbf{0.3107_{\pm 0.013}}$ | $\mathbf{0.3505_{\pm 0.010}}$ | $0.4058_{\pm 0.012}$ | $\mathbf{0.3702_{\pm 0.015}}$ | $\mathbf{0.3557_{\pm 0.014}}$ | $0.1108_{\pm 0.006}$ | $0.2005_{\pm 0.013}$ |
| | 100 | $\mathbf{0.1652_{\pm 0.015}}$ | $0.4850_{\pm 0.014}$ | $0.5152_{\pm 0.016}$ | $\mathbf{0.4450_{\pm 0.020}}$ | $\mathbf{0.4308_{\pm 0.018}}$ | $\mathbf{0.0552_{\pm 0.004}}$ | $0.2993_{\pm 0.021}$ |

