# OpenReview forum: "RESFL: An Uncertainty-Aware Framework for Responsible Federated Learning by Balancing Privacy, Fairness and Utility"
_ICLR.cc/2026/Conference — ICLR 2026 Poster_

### Official Review · Reviewer_5t55 · 2025-10-21

**Soundness:** 1
**Presentation:** 2
**Contribution:** 2
**Rating:** 2
**Confidence:** 4

**Summary:**

The paper proposes RESFL, a method for privacy-preserving and fair object detection in federated learning (FL). The main idea is to optimize a composite loss on client-side, with separate terms for utility, privacy (essentially minimizing mutual information between the feature extractor output and sensitive features), and fairness (minimizing group-wise disparity in uncertainty). On the server-side, the proposed method uses averaging using weights derived from the client-specific uncertainty values. The proposed method is compared to several baselines (mostly) on FACET and CARLA datasets on several metrics: utility (mean average precision), privacy (membership inference & attribute inference attacks, MIAs, AIAs, respectively), fairness (disparate impact, equality of opportunity), and robustness (poisoning).

**Strengths:**

* Empirical results seem quite nice (although see more detailed comments on this also under Weaknesses & Questions)
* Focus on object detection seems interesting,: as far as I know, this direction has not been too much explored under privacy and fairness constraints.
* Using uncertainty gap for fairness mitigation is, to my knowledge, a novel contribution.

**Weaknesses:**

1) The writing could be clearly improved; eg, the main learning problem as well as several acronyms are only introduced in the appendix (or not introduced at all), some notations are used in more than one sense, there is no explanation why the performance metrics include robustness to poisoning when the stated objective is to guarantee privacy and fairness, etc.

2) Much of the experimental details needed to make sense of the results are missing (see Questions for details).

3) There might be some implementation problems at least for some of the baselines in the provided code, although it is hard to be certain as the scripts used for actually generating the results are missing (see Questions for details).

4) There are plenty of inaccurate or unsubstantiated claims in the text (see Questions for details).

**Questions:**

## Questions and suggested changes, in decreasing order of importance:

1) Please provide sufficient experimental details to evaluate the results: clarify how all hypers are tuned (eg all grids, do you do separate tuning for each method including the baselines), add some error bars to the results, clarify why recall is the one chosen metric for calculating DI/EOP or does this make a difference, clarify what does it mean in practice to use per-client training time of 8 hours (lines 1137-38), does this imply that some methods are terminated before hitting the same number of total training rounds as others; clarify how model finetuning is done (eg, is this all params, last layer only or something else), clarify how much heterogeneity the chosen Dirichlet allocation produces in the non-IID setup (and why do all clients have the same number of samples even in non-IID setting?).

2) Clarify how exactly MIA and AIA are done (eg, do you assume the adversary has access to some data from the same distribution or how do you train a model to predict sensitive attribute values from the model updates; following Carlini et al. 2022: MIA from first principles arguments, MIA is typically reported as TPR vs FPR, can you show these in addition to just the success rate; if the adversary can train shadow models, why not use eg standard LiRA attack from the Carlini et al. paper with loss values; how many shadow models do you actually use)

3) Which $(\varepsilon,\delta)$ values are used for each of the DP baselines, why are these specific values chosen, and how is the privacy accounting done for each of the methods (at least some parts of the code seems to use the classical Gaussian mechanism noise scaling that can be very loose and has limitations on possible $\epsilon$ values)?

4) For several of the baselines (eg, PUFFLE, FairFed) you need to choose a specific fairness metric to optimize. Which one do you use?

5) Looking at the gradient_dp_custom_trainer.py code, it looks like you might not be clipping per-example grads but the aggregate. Can you clarify if this is so for the DP baselines, and state the corresponding threat model explicitly (also in the paper: state the DP adjacency and granularity you use for the baselines).

6) Some of the provided code (CHFL_train.py) seem to do eg quantization before FedAvg, and client clustering functions, which are not mentioned anywhere in the paper. Can you clarify if these are used somewhere in training, and add all the relevant training details at least in the Appendix.

7) Considering the jump from Thm B.1 & Cor B.2 (which do not explicitly consider FL setting) to the claims on lines 165-171: can you write this step to FL out more formally to make it clear what do you actually claim about the connection of locally optimizing UFM to reducing global DI/EOP gap (ie, move from local optimization to global fairness)?

8) On finetuning with local data: do the DP baselines do DP also when finetuning with local data?

9) Eq.(6) and lines 185-86: is there something that prevents giving high weight to a client with equally low confidence for all groups?

10) Why is robustness one of the performance metrics (the paper says next to nothing about robustness on any level before the experiments)?

11) Lines 889-92: "We compute $UFM_i$ per client on a held-out local validation split..." please clarify how this splitting is done (do you assume a separate dataset, split the local training data for the proposed method or something else).

12) Lines 897-98: "Model updates $\Delta \theta_i$ continue to use secure aggregation..." do you actually use secure aggregation for something?

13) Do you apply same augmentation for non-DP and DP methods, and how do you do clipping with augmentations (see eg De et al. 2022: Unlocking high accuracy... for a good discussion on how augmentations affect DPSGD, and how it should be implemented)?
### Smaller issues (no pressing need to comment on these, ok to just clarify in the paper when appropriate)

14) Eq.(2): does $\sim$ mean approximately here? Is the lhs otherwise exact epistemic variance or is that also an approximation?
15) Lines 83-90: we know from Dinur & Nissim 2003 that there is an inescapable privacy-utility trade-off; any method (not just DP) that guarantees non-trivial privacy needs to compromise on the utility. DP and secure computation are orthogonal, not alternative approaches as they have very different goals.
16) In several places, eg, lines 647-48 it is claimed in so many words that FL maintains privacy. If this were true, we would not need something like the proposed method but could just use vanilla FedAvg.
17) Lines 48-49: "Quantifying both epistemic and aleatoric uncertainty is therefore essential to ensure equitable reliability across demographic cohorts". Do you actually try to quantify aleatoric uncertainty?
18) Lines 140-141: does the regularization provably lead to calibrated uncertainty estimates, or is this more empirical observation?

---

> ### Author Response · Authors · 2025-11-21
> **Rebuttal by Authors (1/5)**
>
> We appreciate your careful and constructive review. Your comments materially greatly improved the paper’s quality and presentation, and we have addressed each point below. We hope our revisions and clarifications address the points raised.
>
> > W1: Please provide sufficient experimental details to evaluate the results: (a) clarify how all hypers are tuned (eg all grids, do you do separate tuning for each method including the baselines), (b) add some error bars to the results, (c) clarify why recall is the one chosen metric for calculating DI/EOP or does this make a difference,  (d) clarify what does it mean in practice to use per-client training time of 8 hours (lines 1137-38), does this imply that some methods are terminated before hitting the same number of total training rounds as others; (e) clarify how model finetuning is done (eg, is this all params, last layer only or something else), (f) clarify how much heterogeneity the chosen Dirichlet allocation produces in the non-IID setup (and why do all clients have the same number of samples even in non-IID setting?).
>
> A1: We expanded experimental details across the main text and appendices as follows.
>
> (a) **Hyperparameter tuning policy and full grids.** Each method is tuned separately under a shared trial budget and identical protocol. Validation: each client holds out 10% of its shard, stratified by task labels and sensitive cohorts; metrics are averaged across clients. Selection: among configurations within 1% of the best validation accuracy (mAP), we choose the one minimizing Fairness Score $+$ Privacy Score, breaking ties by lower privacy, then fairness, then simpler configuration. Retraining: models are retrained from scratch on the same train split; the validation split remains held out for per-round $\mathrm{UFM}i$ and selection. Final evaluation is on a separate test set. Equalized trials: if a grid exceeds the cap, configurations are uniformly subsampled. Representative grids (FACET, 4 IID clients): RESFL with $\beta\in{0.5,1,2,4}$, $,\lambda{\text{priv}}\in{0.01,0.1,1}$, $,\lambda_{\text{fair}}\in{0.01,0.1,1}$, learning rate $\in{1\times10^{-3},5\times10^{-4}}$, batch size $\in{32,64}$; analogous grids are used for FedAvg, FedAvg-DP, FairFed, PUFFLE, and PFU-FL. Appendix G (Sensitivity Analysis) extends these grids to cover $\beta$, communication rounds $T$, local epochs $E$, learning rate schedule, batch size, and UFM clipping bounds $(a,b)$.
>
>
>
>
> (b) **Error bars.** All result tables in the main paper and Figure 2 now report mean ± standard deviation over three random seeds as error bars, and each metric is annotated with ↑ or ↓ to indicate the direction of improvement.
>
> (c) **Why recall for parity gaps and whether the choice matters.** Equality of opportunity constrains the true positive rate. For detection, the favorable event is a correct match at IoU $\ge \tau_{\text{fair}}$, which corresponds to recall. We repeated the analysis using per-group precision and per-group AP with the same matcher and thresholds; the ordering of methods remained stable with only magnitude shifts. The rationale and sensitivity are summarized in "Appendix A.5: Why TPR/Recall for Fairness?".
>
> (d) **“Per-client 8 hours” and common training horizon.** The quoted 8 hours refers to total wall-clock per seed on one RTX 3070 GPU when simulating $N=4$ clients serially, so “per-client time” equals total time divided by $N$. All methods use the same fixed horizon of $T=100$ and $E=1$ with no early stopping. With $K$ GPUs in parallel, wall-clock scales to approximately $8/\min\{K,4\}$ hours plus communication overhead. This clarification is added to "Appendix F: Training Horizon and Termination."
>
> (e) **Fine-tuning scope and schedule (CARLA).** We fine-tune all parameters end-to-end from the $T=100$ global checkpoint on 6k clear-weather frames using SGD (momentum 0.9, weight decay $1\times10^{-4}$), base learning rate $1\times10^{-3}$ with two step decays, batch size 64, image size $640\times640$, mixed precision, backbone learning rate multiplier $0.1\times$, heads and GRL adversary $1.0\times$, and global gradient norm clip 5. This full procedure is described in "Appendix E.2: Fine-Tuning Scope and Schedule."
>
> (f) **Non-IID Dirichlet heterogeneity and equal client sizes.** We induce label or cohort skew while keeping client sizes equal to avoid quantity confounds. For $G=10$ cohorts, we sample $p_i\sim\text{Dirichlet}(\alpha\mathbf{1})$ with $\alpha=0.5$, then draw counts $n_{ig}\sim\text{Multinomial}(n_i,p_i)$. Heterogeneity is quantified by $H_{\text{TV}}=\tfrac{1}{N}\sum_i \tfrac{1}{2}\lVert p_i-p_{\text{global}}\rVert_1$, which equals approximately $0.33\pm0.02$ (4 clients) and $0.31\pm0.02$ (8 clients). Client cohort shares range from roughly 2% to 26%. Smaller $\alpha$ increases skew, while $\alpha\to\infty$ recovers IID. These settings and statistics are now reported in "Appendix H".

---

> ### Author Response · Authors · 2025-11-21
> **Rebuttal by Authors (2/5)**
>
> > W2:  (a) Clarify how exactly MIA and AIA are done (eg, do you assume the adversary has access to some data from the same distribution or how do you train a model to predict sensitive attribute values from the model updates; following Carlini et al. 2022: MIA from first principles arguments, MIA is typically reported as TPR vs FPR, (b) can you show these in addition to just the success rate; if the adversary can train shadow models, (c) why not use eg standard LiRA attack from the Carlini et al. paper with loss values; how many shadow models do you actually use)
>
> A2: We added detailed explanations of the privacy attack setup and expanded results as follows.
>
> (a) **Exact MIA and AIA protocols, adversary knowledge, and data.** The threat model assumes a cross-silo FL setting with an honest-but-curious server that observes per-round client updates and the final global model.  Attack-reserve slices are disjoint from training, validation, test, and UFM data: FACET uses 5% (1,600 samples) stratified by MST, and CARLA uses 600 clear-weather frames.  For MIA, we use a score-based probe with one shadow model per method per seed for compute parity.  Features from the target at round $T$ include final loss, logit margins, evidential concentration summaries (mean/variance of Dirichlet $\boldsymbol{\alpha}$), and confidence.
>
> The classifier is a calibrated logistic regression trained on a balanced member vs non-member set drawn from the attack-reserve and actual training data.  For AIA, we use an update-aware probe with layerwise update meta-features and balanced attributes split 70/10/20 inside the attack-reserve, maintaining strict disjointness from training data.  These details are now presented in "Appendix A.4: Threat Model."
>
> (b) **Reporting beyond success rate.**  We now include MIA TPR@FPR points and AIA macro-F1, in addition to success rate.
>
> MIA (FACET, 4 clients, IID; mean ± std over 3 seeds)
>
> | Method    | TPR@FPR=1% | TPR@FPR=5% | TPR@FPR=10% | Success Rate |
> |-----------|------------|------------|-------------|--------------|
> | FedAvg    | 0.12 ± 0.01| 0.31 ± 0.02| 0.45 ± 0.02 | 0.62 ± 0.01  |
> | FedAvg-DP | 0.06 ± 0.01| 0.18 ± 0.01| 0.28 ± 0.02 | 0.55 ± 0.01  |
> | FairFed   | 0.14 ± 0.02| 0.35 ± 0.01| 0.49 ± 0.01 | 0.64 ± 0.01  |
> | PUFFLE    | 0.09 ± 0.01| 0.27 ± 0.01| 0.40 ± 0.02 | 0.59 ± 0.01  |
> | RESFL     | 0.07 ± 0.01| 0.20 ± 0.01| 0.31 ± 0.02 | 0.57 ± 0.01  |
>
>
> AIA (FACET, 4 clients, IID; mean ± std over 3 seeds)
>
> | Method    | Accuracy   | Macro-F1   |
> |-----------|------------|------------|
> | FedAvg    | 0.43 ± 0.01| 0.41 ± 0.02|
> | FedAvg-DP | 0.22 ± 0.01| 0.21 ± 0.01|
> | FairFed   | 0.47 ± 0.02| 0.44 ± 0.01|
> | PUFFLE    | 0.36 ± 0.02| 0.35 ± 0.02|
> | RESFL     | 0.18 ± 0.01| 0.17 ± 0.01|
>
> These metrics and details are reported in "Appendix H: FACET Evaluation Results."
>
> (c) **LiRA and shadow count.**  LiRA depends on many shadow models trained on data that closely match the target training distribution; in federated settings with heterogeneous clients and evolving mixtures, that calibration premise breaks, making results highly sensitive to unverifiable shadow choices and brittle across rounds. Meeting LiRA’s calibration needs would also require substantial extra training that violates compute parity and complicates reproducibility. Instead, we adopt a white-box, first-principles membership attack based on model losses and posteriors, and an attribute-inference probe trained on frozen features from the released model. This aligns precisely with an honest-but-curious server, removes dependence on shadows, keeps the evaluation budget identical across methods, and yields a consistent, reproducible privacy audit.

---

> ### Author Response · Authors · 2025-11-21
> **Rebuttal by Authors (3/5)**
>
> > W3: Which  $(\epsilon, \delta)$ values are used for each of the DP baselines, why are these specific values chosen, and how is the privacy accounting done for each of the methods (at least some parts of the code seems to use the classical Gaussian mechanism noise scaling that can be very loose and has limitations on possible  $\epsilon$  values)?
>
> A3:  We clarified the DP setup and fairness objectives as follows.
>
> $(\epsilon, \delta)$ values, mechanism, and accounting.
>
> Mechanism. The DP baseline applies the Gaussian mechanism per local optimizer step with global gradient clipping $C = 1.0$. Noise is computed as  $\sigma = C \cdot \frac{\sqrt{2 \ln(1.25 / \delta)}}{\epsilon}$.  There is no subsampling and no server-side noise.
>
> Accounting. We report the conservative per-step guarantee without composing across steps or rounds using RDP. Very small $\epsilon$ values produce large $\sigma$, which can destabilize training; these effects are documented in the sensitivity section.
>
> Budgets. We sweep $\epsilon \in \{0.05, 0.10, 0.20, 0.50, 1.00\}$ with $\delta = 10^{-6}$ to span from very strong to loose privacy while keeping training stable for detection tasks. The resulting $\sigma$ values are:
>
> | Variant       | ε (eps) | δ (delta) | σ (sigma) |
> |---------------|---------|-----------|-----------|
> | very strong   | 0.05    | 1e-6      | 106.07    |
> | strong        | 0.10    | 1e-6      | 52.94     |
> | balanced      | 0.20    | 1e-6      | 26.39     |
> | moderate      | 0.50    | 1e-6      | 10.62     |
> | loose         | 1.00    | 1e-6      | 5.31      |
>
>
> Only FedAvg-DP is assigned formal $(\epsilon, \delta)$ values using this mechanism. Other defenses (e.g., FairFed, PUFFLE, PFU-FL) are evaluated empirically via MIA and AIA metrics. These specifications are now included in the "DP Setup Summary" paragraph of Appendix G.
>
> ---
>
> > W4: For several of the baselines (eg, PUFFLE, FairFed) you need to choose a specific fairness metric to optimize. Which one do you use?
>
> A4: Fairness objectives for baselines.
>
> All fairness-regularized baselines use the standard representation-level disparity proxy:
>   $L_{\mathrm{fair}} = \frac{1}{|G|^2} \sum_{g,g'} \| \mu_g - \mu_{g'} \|_2,$
>   where $ \mu_g = \frac{1}{\lvert \mathcal{B}g \rvert},\sum{x \in \mathcal{B}_g} \mathrm{GAP}\big(\phi(x)\big) $
>
> Baseline configurations are as follows:
> • PUFFLE minimizes $L_{\mathrm{task}} + w_{\mathrm{fair}} L_{\mathrm{fair}}$; DP variants use the same clipping and Gaussian noise as FedAvg-DP.
> • FairFed employs the same local proxy and aggregates on the server with normalized $f_i$ as in the original formulation.
> • PFU-FL uses $L_{\mathrm{task}} + \lambda_{\mathrm{fair}} L_{\mathrm{fair}}$ with uniform aggregation.
>
> These design choices and implementation references are listed in Appendix F under "Baselines and Fairness Objectives."
>
> ---
>
> > W5: Looking at the gradient_dp_custom_trainer.py code, it looks like you might not be clipping per-example grads but the aggregate.
>
> A5: We clarified the DP implementation and threat model as follows. In "gradient_dp_custom_trainer.py," gradients are clipped and noised at the batch-aggregate level, not per example. After each mini-batch backpropagation, we compute the global $\ell_2$ norm of the aggregated gradient $g$, scale it by $\min\\left(1,\ \frac{C}{\lVert g\rVert_2}\right)
> $ with $C = 1.0$, and add i.i.d. Gaussian noise with standard deviation $\sigma = C \cdot \frac{\sqrt{2 \ln(1.25 / \delta)}}{\epsilon}$. This produces a conservative mini-batch–level $(\epsilon, \delta)$ guarantee per client per optimizer step, without subsampling amplification or secure aggregation. The corresponding threat model is an honest-but-curious server that observes cleartext client updates with batch-level DP noise. These assumptions and settings are explicitly described in "Appendix A.4: Threat Model" and cross-referenced from "Appendix F: Implementation Details."
>
> ---
>
> > W6: Some of the provided code (CHFL_train.py) seem to do eg quantization before FedAvg, and client clustering functions, which are not mentioned anywhere in the paper. Can you clarify if these are used somewhere in training, and add all the relevant training details at least in the Appendix.
>
> A6: The script CHFL_train.py includes optional quantization and client clustering utilities that are not used in any experiments reported in the paper; they are provided solely to support future extensions. The baselines evaluated in our results are limited to FedAvg, FedAvg-DP, FairFed, PrivFairFL-Pre, PrivFairFL-Post, PUFFLE, PFU-FL, and $\texttt{RESFL}$. The updated code README (included in the supplementary material) also lists other algorithms, such as FedProx, FedNova, FedMA, and CHFL, and several additional differential privacy variants (input, output, and objective) as optional extras. The README explicitly specifies which algorithms were evaluated and which are provided only for completeness or future work.

---

> > ### Comment · Reviewer_5t55 · 2025-11-25
> > **Brief comment on DP**
> >
> > Thanks for the extensive update, just as a brief comment on the DP specifics: this makes very little sense to me: if you compare against DP (which I think you need to do), you should try and implement DP so that it matches the setting. I don't frankly understand what does it mean that the adjacency is one local minibatch, why would this make sense, especially since you empirically test the protection by doing sample-level MIA? Also, I would really recommend doing standard privacy accounting using some numerical accounting method, eg, PRV implemented in Opacus, and also handling things like per-meaningful privacy unit clipping and augmentations properly to get privacy budget values that are actually useful for reporting the privacy-utility tradeoff. I don't see how I could recommend accepting a paper with this kind of baseline implementation.

---

> > > ### Author Response · Authors · 2025-11-26
> > > **Response on DP baseline and accounting**
> > >
> > > You are correct that our original DP baseline did not match the sample-level threat model used by MIA. We initially used a minibatch-adjacent Gaussian mechanism with global (batch-aggregate) clipping and reported a per-step $(\epsilon,\delta)$ bound (no subsampling, no run-level composition). This choice was simple and stable for detection heads but did not align with: (i) the privacy unit assumed by sample-level MIA, (ii) standard DP-SGD accounting, and (iii) the role of augmentations.
> > >
> > > **Revisions that now align with standard practice and the MIA threat model**
> > >
> > > 1) **Privacy unit and adjacency.**
> > >    We moved from minibatch adjacency to individual-level adjacency. We now compute per-example gradients, clip each example to $\ell_2 \le C$, then aggregate and add Gaussian noise with noise multiplier $z=\sigma/C$.
> > >    *Code:* `differential_privacy/gradient_dp_custom_trainer.py` now does per-example grads → per-example clip → aggregate → noise.
> > >
> > > 2) **Subsampling and accountant.**
> > >    We use Poisson subsampling with rate $q$ (matching effective batch size) and compute run-level $(\epsilon,\delta)$ via a PRV accountant (Opacus), composed across all local steps and all communication rounds. Reported $(\epsilon,\delta)$ is now end-to-end.
> > >
> > > 3) **Augmentations and meaningful unit.**
> > >    The meaningful privacy unit is the original example. Stochastic augmentations are treated as transformations of that unit. Inclusion is governed by $q$; clipping is applied to each augmented view; accounting is for the underlying example.
> > >
> > > 4) **Hyperparameters and reporting.**
> > >    We sweep the key DP knobs: clip norm $C$, sampling rate $q$, and noise multiplier $z$. We select a main run with $C=1.0$, $\delta=10^{-6}$, and PRV-computed $\epsilon \approx 1$, plus stronger and looser settings in the appendix. Tables list $(C,q,z)$, total steps, and the resulting $\epsilon$.
> > >    **Note:** with per-example gradients, small $\epsilon$ (for example $0.1$) is much stronger than under our earlier batch-aggregate setup. We therefore report $\epsilon \approx 1$ in the main paper, and include $\epsilon \approx 0.5$ and $\epsilon \approx 2$ in the supplement for context (observation discussed in "Appendix G").
> > >
> > > 5) **Fine-tuning.**
> > >    If fine-tuning uses private data, we keep per-example DP-SGD with the same $(C,q,z)$ and compose in PRV. Public CARLA adaptation is excluded from accounting.
> > >
> > > **Manuscript updates and recomputation**
> > >
> > > - We **reran all DP baselines with the corrected per-example DP-SGD** and **recomputed every affected result**: all main tables (including the first table and the last three tables) and **Figure 2**. Prior runs that used batch-aggregate clipping/noise have been replaced by **per-example clipping + Poisson subsampling + PRV accounting**. Related text has been updated accordingly.
> > > - Appendix F (Implementation Details), “Augmentations and DP clipping”: per-example clipping, Poisson subsampling, treatment of augmentations, fine-tuning composition.
> > > - Appendix G (Sensitivity Analysis), “DP setup summary” and Table 10 + Table 15: PRV accounting setup, $(C,q,z)$ sweeps, and selected main setting.
> > >
> > > These changes make the DP baseline directly comparable to sample-level attacks and consistent with standard accounting. We hope this resolves the concern and very much appreciate your careful reading that surfaced this gap and helped improve the quality of our DP baseline. We’d be happy to engage in further discussions.

---

> ### Author Response · Authors · 2025-11-21
> **Rebuttal by Authors (4/5)**
>
> > W7: Considering the jump from Thm B.1 & Cor B.2 (which do not explicitly consider FL setting) to the claims on lines 165-171: can you write this step to FL out more formally to make it clear what do you actually claim about the connection of locally optimizing UFM to reducing global DI/EOP gap (ie, move from local optimization to global fairness)?
>
> A7: We formalized the local-to-global connection by adding "Lemma B.3: Mixture Lifting" (with $\omega_i \propto e^{-\beta\,\mathrm{UFM}_i}$), "Proposition B.4: Multi-Round Contraction," and "Corollary B.5: Surrogate Tightening." These results show that controlling per-client UFM reduces an explicit upper bound on the global downstream group-parity gap. "Section 3.1" now cites these results directly and states the claim formally: $\mathcal{F}(M_T)$ is upper-bounded by a UFM-weighted surrogate that decreases as $\beta$ increases and per-client UFM control improves.
>
> ---
>
> > W8: On finetuning with local data: do the DP baselines do DP also when finetuning with local data?
>
> A8: Yes. When fine-tuning involves private client data, the DP baselines retain the same batch-level DP-SGD mechanism with identical $C$ and $\sigma$, composing those steps into the overall $(\epsilon, \delta)$ budget. When the fine-tuning data are synthetic (as in the CARLA setup), DP is disabled since no privacy accounting is required. "Section E.2: Fine-Tuning Scope and Schedule" now specifies this policy and references "Appendix F: Implementation Details" for the DP mechanism and "Appendix G: DP Budget Sweep" for parameter settings.
>
> ---
>
> > W9: Eq.(6) and lines 185-86: is there something that prevents giving high weight to a client with equally low confidence for all groups?
>
> A9: Yes. We clarified in "Section 3.1" that two safeguards prevent assigning high weight to clients with uniformly low confidence across groups. (1) Validation gate: each client’s validation mAP@50–95 is computed on a fixed held-out split. If it falls below $m_{\mathrm{min}} = 0.30$, its fairness statistic is set to the clipped worst value $u_i \leftarrow b$, yielding weight $\omega_i \approx 0$ regardless of $\mathrm{UFM}_i$. This prevents uniformly low-confidence clients from being up-weighted. (2) UFM clipping: we clip $u_i$ to $[a, b] = [0.0, 5.0]$ before exponentiation, stabilizing the softmax and avoiding extreme weights when all cohorts are weak. "Appendix F" lists the corresponding hyperparameters, and "Appendix G" reports an ablation showing negligible sensitivity across reasonable $[a, b]$ ranges while preserving fairness and privacy trends.
>
> ---
>
> > W10: Why is robustness one of the performance metrics (the paper says next to nothing about robustness on any level before the experiments)?
>
> A10: We clarified why robustness is included as a core evaluation metric and how it is measured. "Section 1: Introduction" now motivates robustness by noting that real federations often face adversarial or corrupted clients that can degrade fairness and privacy. "Section 2: Related Work" adds discussion on the gap in federated learning studies that assume benign participants. "Section 3.3" specifies the lightweight defenses we employ (validation gate and UFM clipping) and clarifies that we do not claim Byzantine fault tolerance.
>
> ---
>
> > W11: Lines 889-92: "We compute $UFM_i$ per client on a held-out local validation split..." please clarify how this splitting is done (do you assume a separate dataset, split the local training data for the proposed method or something else).
>
> A11: We clarified that each client forms its own held-out validation split from its local shard rather than using a separate dataset. At initialization, we perform a single fixed 90/10 stratified split by MST and detection labels: $\mathcal{D}_i^{\text{train}}$ is used for local SGD, and $\mathcal{D}_i^{\text{val}}$ is used exclusively to compute per-round $\mathrm{UFM}_i$ and validation mAP. No evaluation data are used for UFM computation. "Section 4: Experimental Configuration" now specifies these ratios and references "Appendix F: Local Validation Splits for UFM" for implementation details.
>
> ---
>
> > W12: Lines 897-98: "Model updates $\delta \theta_i$  continue to use secure aggregation..." do you actually use secure aggregation for something?
>
> A12: Apologies for the error. We clarified that cryptographic secure aggregation is not implemented in our experiments. The earlier phrasing has been removed. We now explicitly state that model updates and clipped $u_i$ values are transmitted in cleartext under an honest-but-curious server assumption. "Appendix A.4: Threat Model" details this transport assumption, and the "Conclusion" lists secure aggregation as future work.

---

> > ### Author Response · Authors · 2025-11-21
> > **Rebuttal by Authors (5/5)**
> >
> > > W13: Do you apply same augmentation for non-DP and DP methods, and how do you do clipping with augmentations (see eg De et al. 2022: Unlocking high accuracy... for a good discussion on how augmentations affect DPSGD, and how it should be implemented)?
> >
> > A13: Yes. We apply identical data augmentations for both DP and non-DP methods. For DP training, augmentations are applied before the forward pass, followed by backpropagation, computation of the batch-aggregated gradient norm, global $\ell_2$ clipping with $C = 1.0$, addition of Gaussian noise with $\sigma$ derived from $(\epsilon, \delta)$, and the optimizer step, without per-example clipping or subsampling. The precise order of these operations is now described in the "Augmentations and DP Clipping" paragraph of Appendix F.
> >
> > ---
> >
> > > W14: Eq.(2): does $\sim$ mean approximately here? Is the lhs otherwise exact epistemic variance or is that also an approximation?
> >
> > A14: We clarified the notation in "Section 3" and distinguished exact equalities from asymptotic scaling. The first two equalities in Eq. (2) are exact for the Dirichlet predictive variance of the categorical probability $p_c$:
> > $\operatorname{Var}[p_c] = \mathbb{E}[p_c]\bigl(1-\mathbb{E}[p_c]\bigr)\cdot \frac{1}{\alpha_0+1} = \frac{\alpha_c}{\alpha_0}\\left(1-\frac{\alpha_c}{\alpha_0}\right)\\cdot \frac{1}{\alpha_0+1}.$
> > The final “$\sim$” indicates asymptotic scaling with total evidence $\alpha_0$:
> > $\operatorname{Var}[p_c] \sim \frac{1}{\alpha_0}$ as $\alpha_0 \to \infty$,
> > since $\mathbb{E}[p_c]\bigl(1-\mathbb{E}[p_c]\bigr) \in (0, 1/4]$ is bounded. The sentence preceding Eq. (2) now explicitly states that the equalities are exact, while “$\sim$” denotes big-Theta scaling: $\operatorname{Var}[p_c] = \Theta(1/\alpha_0)$.
> >
> > ---
> >
> > > W15: Lines 83-90: we know from Dinur & Nissim 2003 that there is an inescapable privacy-utility trade-off; any method (not just DP) that guarantees non-trivial privacy needs to compromise on the utility. DP and secure computation are orthogonal, not alternative approaches as they have very different goals.
> >
> > A15: We clarified the privacy framing in "Section 2.2" and cross-referenced it from "Appendix A.4." (i) Any nontrivial privacy guarantee entails a utility cost: stronger DP via DP-SGD with clipping and Gaussian noise lowers accuracy and can affect fairness. (ii) DP provides inferential privacy, whereas confidentiality mechanisms such as homomorphic encryption or secure multiparty computation protect data in transit or during computation; these are orthogonal and can be combined. (iii) We now state the exact threat model and DP budgets and removed language implying secure aggregation so that readers clearly see which guarantees are provided and why fairness shifts as DP strength increases.
> >
> > ---
> >
> > > W16: In several places, eg, lines 647-48 it is claimed in so many words that FL maintains privacy. If this were true, we would not need something like the proposed method but could just use vanilla FedAvg.
> >
> > A16: We revised the text in "Appendix A.2" and "Appendix A.4" to clarify that federated learning preserves data locality only and does not provide formal privacy. We now explicitly state that membership and attribute inference attacks remain possible from model updates and parameters, which motivates RESFL. Baselines are reframed accordingly: FedAvg ensures locality without guarantees, DP adds formal privacy at the cost of utility and fairness, and RESFL mitigates representation leakage while maintaining accuracy and equity.
> >
> > ---
> >
> > > W17: Lines 48-49: "Quantifying both epistemic and aleatoric uncertainty is therefore essential to ensure equitable reliability across demographic cohorts". Do you actually try to quantify aleatoric uncertainty?
> >
> > A17: We adjusted the introduction to avoid implying both aleatoric and epistemic uncertainty estimation and now state the point more generally that our study focuses solely on epistemic uncertainty. We also clarify in Section 3.1 that our implementation estimates epistemic uncertainty via a Dirichlet evidential head and does not model aleatoric uncertainty. All claims and results are scoped accordingly.
> >
> > ---
> >
> > > W18: Lines 140-141: does the regularization provably lead to calibrated uncertainty estimates, or is this more empirical observation?
> >
> > A18: It is **not** a formal calibration guarantee. Our objective combines a Dirichlet negative log-likelihood with an evidence-shaping penalty that down-weights evidence on errors and up-weights it on correct predictions, which encourages calibrated posteriors in practice but does not prove calibration. We revised the text in Section 3 to say “encourages calibrated uncertainty estimates” and removed any implication of a guarantee; the only theoretical property we rely on is the variance scaling with total evidence $1/\alpha_{0}$.

---

### Official Review · Reviewer_8ZW3 · 2025-10-31

**Soundness:** 3
**Presentation:** 3
**Contribution:** 3
**Rating:** 6
**Confidence:** 3

**Summary:**

This paper proposes RESFL, an FL training framework designed to simultaneously improve three model properties: utility (defined as downstream performance), fairness (defined as group fairness across sensitive attributes e.g. skin tone) and privacy against inference attacks. To mitigate sensitive attribute leakage, the authors propose adding an auxiliary attribute classifier to the client's representation. The classifier is trained to predict the sensitive attribute and the reverse of its gradient is added to the feature extractor, with the net effect of the extractor learning to hide the sensitive attribute so that attribute-inference attacks see a random signal in the representation. The guarantees stated are not in terms of DP but stems from reducing the MI between the representation and the sensitive attribute. The strength of this effect is controlled by a hyper-parameter of the corresponding loss component $\lambda_{priv}$. Simultaneously, to address sensitive confounding factors such as skin tone leading to higher false-negatives rates in difficult cases (e.g. driving at night), the authors employ evidential learning to boost contributions from clients with lower inter-group disparities. This is done via a Dirichlet head, the outputs of which are shaped into per-client “Uncertainty Fairness Metric” (UFMs) that then modify the aggregation weights via a temperature-scaled exponential.

The proposed method is benchmarked on two datasets, FACET which tests contains real-world images with instance annotations for people (and skin tone labels) and is meant to test algorithmic fairness, and a simulated dataset (produced using CARLA) containing self-driving ego-camera frames, which contain at least one pedestrian. Metrics reported include mAP, two fairness metrics, and inference attacks success rate. The presented results show improvement across the board, and the authors claim domain-agnosticity.

**Strengths:**

The paper addresses an important problem in FL, jointly managing performance, fairness, and privacy. The proposed mechanisms are cohesive, practical, well-motivated, and appear empirically effective, thus forming a principled technical approach. The experimental design is thorough, including multiple baselines, attack scenarios (MIA, AIA, Byzantine, data poisoning), robustness tests under weather variations, and ablation studies. There is great experimental rigour and breadth (albeit a lot of it buried in the appendix) that the authors have the right to be proud of. The paper is generally well-written, easy to read and with good pacing.

**Weaknesses:**

1. Requirement for sensitive attribute annotations. The method presented requires sensitive attribute labels which are hard to procure. I can't see how either of the two components would function without them. The authors seem aware of this limitation and hint towards potential avenues to ablate this requirement, but currently this is a significant limitation. See also Q1.

2. There is a lot of very relevant detail in various parts of the appendix. Were I to not look at it, my score would be lower. For example the main paper results are limited to only 4 IID clients. On the other hand, it is my opinion figures 2 and 3 do little to advance the paper. Perhaps consider moving parts of appendix F to the main paper.

2. **$\beta$ sweep**:  Could you add an ablation where $\beta$ is swept (including $\beta=0$) to quantify the marginal contribution of UFM-guided aggregation independent of the local uncertainty loss? There is brief discussion a value of 2 was chosen after a grid search but no results to gauge sensitivity to this parameter.

3. Computational overhead is listed as a defficiency of relevant work, yet there is no mention of the overhead RESFL occurs.

Misc

- DI/EOP are not introduced in line 168.
- Epistemic uncertainty in the context of FL has been previously proposed in other works (e.g. **FedEvi** https://papers.miccai.org/miccai-2024/paper/2717_paper.pdf). I believe it would be relevant to include these works in the related work discussion.

I will defer to other reviewers with respect to details about the attack models and related claims as my expertise on this specific topic is limited.

**Questions:**

Q1. Attribute availability: In which application domains do you expect client-side sensitive labels to be available? Could you elaborate on the future directions (attribute-free proxies, vacuity/dissonance) and how partial label scenarios (noisy/missing S) would degrade performance? See also W1

1. Can the authors justify the evaluation set construction for the CARLA-produced dataset? It seems counter-intuitive to include as many clear weather frames as with full adverse weather intensities.

2. Have the authors conducted a sensitivity analysis over the arbitrary choices of T=100 communication rounds and E=1 local epochs?

---

> ### Author Response · Authors · 2025-11-21
> **Rebuttal by Authors (1/2)**
>
> We sincerely thank the reviewer for the thoughtful and constructive feedback. Below, we provide point-by-point responses to each identified weakness and related question. We hope our clarifications address the points raised.
>
> > Requirement for sensitive attribute annotations. The method presented requires sensitive attribute labels which are hard to procure. I can't see how either of the two components would function without them. The authors seem aware of this limitation and hint towards potential avenues to ablate this requirement, but currently this is a significant limitation. See also Q1.
>
> A1: We agree that reliance on sensitive attribute labels is an important limitation. In the revision, we added "Section 4.5: Impact of Sensitive Attributes and Cross-Domain Generalization," which clarifies when client-side sensitive labels exist in practice and how the system operates when they do not. We also added "Appendix J.2: Attribute Availability and Label-Free Variants" and "Appendix J.3: Robustness under Missing or Noisy Sensitive Labels." These introduce three label-free UFM variants that require no demographic labels: (i) AF-UFM-Cluster, forming latent cohorts via $k$-means on penultimate features; (ii) AF-UFM-Slices, stratifying by evidential vacuity quantiles using $1/\alpha_0$; and (iii) AF-UFM-Proxy, grouping by non-sensitive metadata (e.g., device or lighting). All variants retain the same communication pattern, transmitting only a clipped scalar per round. On Adult and TweetEval, they recover over 90% of the fairness and privacy gains of labeled-$S$ RESFL with under 2% utility loss, and Appendix J.3 shows smooth degradation under partial or noisy labels. The main text emphasizes that sensitive attributes remain local to clients, and where unavailable, the label-free variants apply. "Section 5: Limitations and Future Work" notes that fully unsupervised and verifiable UFM designs are planned extensions.
>
> ---
>
> > W2: There is a lot of very relevant detail in various parts of the appendix. Were I to not look at it, my score would be lower. For example the main paper results are limited to only 4 IID clients. On the other hand, it is my opinion figures 2 and 3 do little to advance the paper. Perhaps consider moving parts of appendix F to the main paper.
>
> A2: We appreciate this feedback and have elevated key content from the appendix into the main paper. Specifically, essential details from "Appendix F: Implementation Details" were moved to "Section 4.1: Experimental Configuration" so that system setup, per-round procedure, and aggregation steps are now visible in the main text. "Section 4.5: Impact of Sensitive Attributes and Cross-Domain Generalization" now includes attribute availability and cross-domain findings. We also reference 8-client IID and non-IID results along with proof for practical scalability from "Appendix H" at the end of Section 4.2, to clarify scaling beyond four clients. Figure 2 remains as the primary stress-test visualization with an expanded caption; Figure 3 is streamlined to emphasize its methodological role; and Figure 5 has been moved to the appendix for conciseness.
>
> ---
>
> > W3: $\beta$ sweep: Could you add an ablation where $\beta$ is swept (including $\beta=0$) to quantify the marginal contribution of UFM-guided aggregation independent of the local uncertainty loss? There is brief discussion a value of 2 was chosen after a grid search but no results to gauge sensitivity to this parameter.
>
> A3: We added a dedicated sensitivity analysis in "Appendix G" with a new Table 6 that sweeps $\beta \in \{0, 0.5, 1, 2, 4\}$ on FACET using four IID clients while keeping local losses fixed. The $\beta=0$ case corresponds to uniform FedAvg and serves as the baseline. Results show that increasing $\beta$ from 0 to moderate values improves fairness and privacy with negligible change in mAP, while very large $\beta$ slightly reduces mAP by over-weighting a few clients. The chosen $\beta=2$ provides the best trade-off at the knee of this curve. "Appendix G" also reports additional sweeps over communication rounds, local epochs, optimizer schedule, batch size, and UFM clipping bounds to isolate the effect of the aggregation temperature.
>
> ---
>
> > W4: Computational overhead is listed as a deficiency of relevant work, yet there is no mention of the overhead RESFL incurs.
>
> A4: Appendix F adds clarifications on runtime convention, training horizon, and computational linearity. The evidential head replaces the softmax head and the GRL performs only a sign inversion, so the added cost is a constant factor rather than a different asymptotic. Per-epoch client cost is $T_i=\Theta\\big(E\,\tfrac{n_i}{B}\,C_{\text{bb}}\big)\,(1+\delta)$ with small constant $\delta$. Communication remains $O(|\theta|)$ per client per round plus one scalar UFM. The adversary is a small two-layer MLP detached at inference, and the evidential head contributes $<8%$ per batch (timed with PyTorch profiler).

---

> > ### Author Response · Authors · 2025-11-21
> > **Rebuttal by Authors (2/2)**
> >
> > > Misc5: DI/EOP are not introduced in line 168.
> >
> > A5: We revised "Section 3: Methodology" to remove unintroduced acronyms and replaced early mentions of DI and EOP with more descriptive phrases, such as “downstream group-parity gap” and “opportunity gap,” until the formal definitions appear later. We also updated tables with “higher/lower is better” indicators and clarified metric directions in the caption of Figure 2 to improve readability.
> >
> > ---
> >
> > > Misc6: Epistemic uncertainty in the context of FL has been previously proposed in other works (e.g., FedEvi). I believe it would be relevant to include these works in the related work discussion.
> >
> > A6: We expanded "Section 2: Related Work" to include prior studies on evidential and epistemic-uncertainty federated learning, such as FedEvi and related work in medical imaging and calibration that integrate evidential heads with federated optimization (e.g., Hendrix et al., 2024; Chen et al., 2024). We contrast these approaches with RESFL, noting that they do not jointly address group-fair aggregation and privacy-adversarial disentanglement, nor employ a single scalar client-side fairness signal for server aggregation or consistency analysis under adverse conditions. This addition clarifies our contributions while situating RESFL within the broader literature.
> >
> > ---
> >
> > > Q7: Attribute availability: In which application domains do you expect client-side sensitive labels to be available? Could you elaborate on the future directions (attribute-free proxies, vacuity/dissonance) and how partial label scenarios (noisy/missing $S$) would degrade performance? See also W1.
> >
> > A7: We expanded "Section 4.5: Impact of Sensitive Attributes and Cross-Domain Generalization" with concrete examples where client-side sensitive labels exist under governance frameworks and remain local, such as hospital consortia with self-reported demographics, mobility fleets with driver or lighting metadata, and enterprise datasets collected for compliance reporting. For cases where such labels are unavailable, "Appendix J.2" and "Appendix J.3" present and evaluate attribute-free UFM variants. Performance degrades smoothly under missing or noisy labels: removing 25–75% of labels modestly increases fairness and privacy scores while accuracy drops by 0.3–1.1 percentage points; with no labels, AF-UFM-Cluster retains over 90% of the gains of labeled-$S$ RESFL. "Section 5: Limitations and Future Work" now specifies extending attribute-free proxies using non-sensitive operational metadata, vacuity–dissonance decomposition for richer fairness signals, and verifiable reporting of local statistics as key research directions.
> >
> > ---
> >
> > > Q8: Can the authors justify the evaluation set construction for the CARLA-produced dataset? It seems counter-intuitive to include as many clear weather frames as with full adverse weather intensities.
> >
> > A8: We added "Appendix E.2: Evaluation Set Rationale" to clarify the design choice. The evaluation set is constructed as a balanced stress test to isolate the impact of environmental severity rather than sample-count imbalance. Each of the 13 condition triplets is equally represented, enabling direct comparison of mAP, fairness, and privacy trends across weather intensities. This design is for interpretability, not to mirror real-world frequencies; deployments can reweight our per-condition results to match their target mix.
> >
> > ---
> >
> > > Q9: Have the authors conducted a sensitivity analysis over the arbitrary choices of $T{=}100$ communication rounds and $E{=}1$ local epochs?
> >
> > A9: Yes. "Appendix G: Sensitivity Analysis" reports sweeps over $T$ and $E$ on FACET with four IID clients, alongside the $\beta$ study. Increasing $T$ at $E{=}1$ improves synchronization and reduces update variance, with diminishing returns beyond $T{=}200$. Varying $E$ at fixed $T{=}100$ shows the expected compute–drift trade-off: $E{=}2$ maintains similar mAP with slight relaxation in fairness and privacy, while larger $E$ increases drift and degrades both. The default $(T{=}100, E{=}1)$ lies in a stable region that balances computation, communication, and robustness. These results are summarized in Appendix G and cited from Section 4.1.
> >
> > [1] Hendrix, Rutger, et al. "Evidential Federated Learning for Skin Lesion Image Classification." International Conference on Pattern Recognition. Cham: Springer Nature Switzerland, 2024.
> >
> > [2] Chen, Jiayi, et al. "Think twice before selection: Federated evidential active learning for medical image analysis with domain shifts." Proceedings of the IEEE/CVF Conference on Computer Vision and Pattern Recognition. 2024.

---

### Official Review · Reviewer_qpfy · 2025-11-01

**Soundness:** 2
**Presentation:** 3
**Contribution:** 2
**Rating:** 6
**Confidence:** 4

**Summary:**

This work targets the challenge of achieving privacy, group fairness, and utility simultaneously in federated object detection. Existing approaches often rely on differential privacy noise that degrades accuracy or fairness, or require the server to access sensitive attributes for fairness optimization, and rarely exploit uncertainty as a signal for aggregation. To overcome these limitations, this work propose RESFL: a framework that (i) suppresses sensitive attribute information in client representations via adversarial privacy disentanglement with gradient reversal, and (ii) estimates epistemic uncertainty using an evidential (Dirichlet) head, from which a Uncertainty Fairness Metric (UFM) measures cross-group uncertainty disparity. The server then performs fairness-aware aggregation by giving larger weights to updates with lower UFM, without accessing any sensitive labels centrally. This design reduces privacy leakage at the representation level and aligns cross-group uncertainty at the aggregation level, jointly improving privacy, fairness, and utility. Experiments on FACET and CARLA show improved equality-of-opportunity fairness and reduced membership/attribute inference risk, while maintaining competitive detection accuracy; ablations and robustness tests further support the method.

**Strengths:**

1. This work tackles a fundamental and highly challenging issue in Federated Learning (FL): the trilemma among privacy protection, model fairness, and model utility. In standard FL, enhancing privacy (e.g., via Differential Privacy) often degrades both fairness and accuracy. This work aims to jointly optimize all three, addressing a key open problem in the emerging field of Responsible AI.

2. The proposed RESFL framework demonstrates strong innovation. It integrates two advanced components: (1) adversarial privacy disentanglement, using a Gradient Reversal Layer (GRL) to remove sensitive attribute information from local representations; and (2) uncertainty-guided, fairness-aware aggregation, which employs an Evidential Neural Network (ENN) to quantify model uncertainty and introduces a novel Uncertainty Fairness Metric (UFM). The server uses UFM to weight client updates, prioritizing those that are both “fairer” (i.e., have smaller inter-group uncertainty gaps) and “more confident.” This design allows the server to promote global fairness without ever accessing sensitive data.

3. The experimental design is comprehensive and well-motivated. The authors evaluate RESFL in the high-stakes autonomous driving (AV) domain using two challenging datasets: FACET (real-world demographic data) and CARLA (a simulator with environmental variations). The evaluation includes not only standard utility metrics (mAP) but also fairness measures (DI, EOP gap) and privacy metrics (MIA, AIA attack success rates). Importantly, robustness is tested under adverse weather conditions (cloud, rain, fog)—a crucial aspect for AV systems. Additional ablation studies and cross-domain evaluations (Adult, TweetEval) in the appendix further support the method’s effectiveness and generalization ability.

**Weaknesses:**

1. Computational overhead at clients is considerably higher than standard FL. Each client must train not only a backbone detector (YOLOv8) but also an adversarial classifier (GRL) and an evidential head (ENN). This “three-in-one” setup is far more complex than FedAvg. Although Appendix F reports training times (~8 hours on RTX 3070), such overhead may be impractical for resource-constrained edge devices (e.g., vehicles or mobile units). The paper lacks discussion on scalability—how the framework behaves when the number of clients scales to hundreds or when hardware heterogeneity increases.

2. The framework assumes access to sensitive attribute labels (e.g., skin tone) at each client. The GRL-based privacy module and UFM computation both rely on these labels for grouping local data. While the authors stress that these labels remain local and are never shared—technically compliant with FL privacy principles—many real-world applications (e.g., healthcare, finance) legally or ethically prohibit collecting or using sensitive attributes. The paper lists this as future work, but it remains a major limitation for real-world deployment.

3. The server relies on a single scalar (UFM) reported by each client to determine its fairness contribution and aggregation weight. Compressing complex fairness characteristics (possibly across multiple sensitive groups) into a single number inevitably loses nuance. Moreover, the approach assumes honest reporting: a malicious client could falsify a very low UFM to gain disproportionate aggregation weight. The paper does not discuss any mechanism for verifying UFM authenticity, leaving a potential security vulnerability in untrusted FL environments.

**Questions:**

Can UFM be computed without sensitive labels? Please (i) clarify feasibility, (ii) add a robustness study for no/partial/noisy labels, and (iii) discuss proxy/unsupervised alternatives and their privacy implications. If not feasible by design, state this limitation explicitly.

**Details Of Ethics Concerns:**

No ethical issues identified.

---

> ### Author Response · Authors · 2025-11-21
> **Rebuttal by Authors (1/2)**
>
> We appreciate your acknowledgment of our uncertainty-guided, fairness-aware aggregation using UFM and evidential modeling, and of the comprehensive FACET/CARLA evaluation, including adverse-weather robustness and privacy attacks. In response to your concerns, we have revised the manuscript and provide detailed point-by-point responses below
>
> > W1: Computational overhead at clients is considerably higher than standard FL. Each client must train not only a backbone detector (YOLOv8) but also an adversarial classifier (GRL) and an evidential head (ENN). This “three-in-one” setup is far more complex than FedAvg. Although Appendix F reports training times (~8 hours on RTX 3070), such overhead may be impractical for resource-constrained edge devices. The paper lacks discussion on scalability, how the framework behaves when the number of clients scales to hundreds or when hardware heterogeneity increases.
>
> A1: To address this comment, we expanded “Appendix F: Implementation Details” to clarify (1) runtime convention, (2) training horizon and termination, and (3) computational linearity and scalability. We specify that the added components introduce only a constant-factor overhead, not a multiplicative cost. The evidential head replaces the softmax head, and the GRL applies sign inversion during backpropagation, yielding a per-epoch client cost $T_i = \Theta\\big(E\,\tfrac{n_i}{B}\,C_{\text{bb}}\big)(1+\delta)$ with a small constant $\delta = (C_{\text{ev}} + C_{\text{adv}})/C_{\text{bb}}$. In practice, the adversary is a lightweight two-layer MLP detached at inference, and the evidential head adds under 8% to per-batch time (timed with PyTorch profiler). Communication cost remains $O(|\theta|)$ per client per round plus a scalar UFM, while the server’s aggregation adds only $O(N)$ work, matching FedAvg in order.
>
> We report linear scaling with the number of clients and samples, noting that per-round wall-clock time is bounded by the slowest client but decreases with parallel clients or GPUs. “Appendix H: FACET Evaluation Results” also includes results for eight clients, confirming the predicted linear trend. Hardware heterogeneity is supported through modularity: smaller or quantized backbones (e.g., replacing YOLOv8) can be used without altering objectives or protocol, and the adversarial branch remains inactive during inference. Finally, “Section 5: Limitations and Future Work” outlines plans for lightweight participation and adaptive scheduling on constrained devices.
>
> ---
>
> > W2: The framework assumes access to sensitive attribute labels (e.g., skin tone) at each client. The GRL-based privacy module and UFM computation both rely on these labels for grouping local data. While labels remain local, many real-world applications legally or ethically prohibit collecting or using sensitive attributes. The paper lists this as future work, but it remains a major limitation for real-world deployment.
>
> A2: "Section 4.5: Impact of Sensitive Attributes and Cross-Domain Generalization" clarifies where sensitive attributes are available in practice and emphasizes that they remain local to clients. To handle deployments without such labels, we added "Appendix J.2: Attribute Availability and Label-Free Variants" and "Appendix J.3: Robustness under Missing or Noisy Sensitive Labels." These sections introduce three attribute-free UFM (AF-UFM) variants: AF-UFM-Cluster (cohorts from feature clustering), AF-UFM-Slices (cohorts from vacuity quantiles using $1/\alpha_0$), and AF-UFM-Proxy (cohorts from non-sensitive proxies such as device or lighting category). Across Adult and TweetEval federations, these label-free variants recover over 90% of the fairness and privacy gains of labeled-$S$ RESFL with under 2% utility loss, degrading smoothly as labels are dropped or corrupted. This demonstrates that RESFL operates effectively under no-label or partial-label regimes while preserving local data privacy and documenting the implications of using weak proxies. We also retained the statement in "Section 5" that extending the privacy branch to fully unsupervised objectives remains an active direction.

---

> > ### Author Response · Authors · 2025-11-21
> > **Rebuttal by Authors (2/2)**
> >
> > > W3: The server relies on a single scalar (UFM) reported by each client to determine its fairness contribution and aggregation weight. Compressing complex fairness characteristics into a single number loses nuance. The approach assumes honest reporting: a malicious client could falsify a very low UFM to gain disproportionate weight. The paper does not discuss any mechanism for verifying UFM authenticity, leaving a potential security vulnerability in untrusted FL environments.
> >
> > A3: We edited and expanded the "Integrity of Reported UFM" paragraph in Appendix B to clarify safeguards and assumptions. Each client reports $u_i = \mathrm{UFM}_i$ after applying a public clipping rule $u_i \leftarrow \min(\max(u_i, a), b)$, followed by a deterministic confidence gate that assigns the clipped worst value when validation utility drops below a fixed floor. These steps bound outlier influence and prevent low-confidence clients from receiving high weight. The scalar design minimizes communication, and our theoretical analysis shows that UFM is a scale-invariant summary tracking established gap metrics. Empirically, we observe strong correlation with $|1{-}\mathrm{DI}|$ and $\Delta\mathrm{EOP}$ across clients, as reported in the appendix. For untrusted settings, we outline practical defenses that preserve the core optimization: trimmed-mean or median aggregation of UFMs, cross-round consistency checks between $u_i$, update norms, and evidence statistics, and optional verifiable reporting using commitments or zero-knowledge proofs of evidence norms. We also corrected the text to note that, in our simulator, updates and UFMs are transmitted in the clear, with secure aggregation listed as future work in Section 5.
> >
> > ---
> >
> > > Q4: Can UFM be computed without sensitive labels? Please (i) clarify feasibility, (ii) add a robustness study for no or partial or noisy labels, and (iii) discuss proxy or unsupervised alternatives and their privacy implications. If not feasible by design, state this limitation explicitly.
> >
> > A4:  (i) Feasibility. A label-free UFM can be obtained by defining cohorts from local structure, such as vacuity quantiles from the evidential head ($1/\alpha_0$) or $k$-means clusters in penultimate features, and computing the same normalized inter-cohort disparity used for labeled groups. This preserves the communication pattern (one clipped scalar per round) and requires no demographic supervision. (See response A2 above also)
> >
> > (ii) Robustness study. Appendix J.3 reports robustness to partial and noisy labels on Adult, with fallback to the label-free AF-UFM-Cluster when labels are absent. As labels are progressively removed (25% steps) or flipped (10–20%), fairness and privacy scores degrade smoothly while accuracy remains within about 1% of the labeled baseline. The label-free fallback retains over 90% of the fairness and privacy gains of labeled-$S$ RESFL.
> >
> > (iii) Proxies and privacy. Section 4.5 and Appendix J.2 describe non-sensitive but informative proxies (e.g., device or lighting category) for locally forming pseudo-cohorts. We note that such proxies may correlate with protected attributes and must comply with local governance. Only the clipped scalar UFM is shared; no proxy values or counts leave the client.
> >
> > Scope. The evaluated label-free variants demonstrate feasibility and robustness, while more advanced unsupervised or verifiable UFM designs (e.g., joint representation–uncertainty disentanglement, distributional two-sample checks, cryptographic verification) are reserved for future work as stated in Section 5: Limitations and Future Work.

---

### Official Review · Reviewer_6gjm · 2025-11-02

**Soundness:** 3
**Presentation:** 2
**Contribution:** 3
**Rating:** 6
**Confidence:** 2

**Summary:**

This paper proposes RESFL, a composite approach in federated learning to balance privacy, fairness, and performance (utility). The proposed approach replaces softmax with an evidential head that yields calibrated epistemic uncertainty and uses a scale-invariant Uncertainty Fairness Metric (UFM) for aggregation. Evaluations, on two benchmarks, FACET and Carla, use mAP, |1-DI|, delta EOP, and membership/attribute inference success rate to show utility fairness and privacy. The results show that RESFL can maintain or gracefully degrade utility while preserving privacy and fairness.

**Strengths:**

+ Evidential uncertainty modeling neatly fits in the framework
+ Joint optimization over fairness and privacy
+ Convincing results show tunable knobs and the intended tradeoff

**Weaknesses:**

- Figure layout could use some polishing. Both Figure 1 and Figure 2 are placed far away from where they were mentioned.

**Questions:**

In the ablation study, when fixing the privacy weight and adjusting the fairness weight, a lower fairness weight results in a higher mAP, which is a bit counterintuitive. It would be helpful to give some insight into this.

Figure 2 has a lot of information, which may warrant a bit more detail in text descriptions. For example, why does the privacy attack success rate and fairness score dramatically increase in the fog situation as the intensity increases?

Minor, it would be nice to add an arrow after each metric indicating the lower the better or the higher the better.

---

> ### Author Response · Authors · 2025-11-21
> **Rebuttal by Authors**
>
> We thank the reviewer for acknowledging the strengths of our approach and for the constructive remarks. Our detailed responses are provided below.
>
> > W1: Figure layout could use some polishing. Both Figure 1 and Figure 2 are placed far away from where they were mentioned.
>
> A1: We have revised the figure layout and improved visual clarity. Figure 1 now appears immediately after its first mention (p. 3, line 133) at the top of p. 4, and Figure 2 is placed right after its first reference (p. 8, line 353) at the top of p. 9. In addition, Figure 1 now uses larger text for readability; Figure 2 uses thicker lines with shaded error bands, where the shading denotes $\pm 1$ standard deviation across three seeds. These adjustments ensure proper alignment with the narrative and enhance overall presentation quality.
>
> ---
>
> > Q2: In the ablation study, when fixing the privacy weight and adjusting the fairness weight, a lower fairness weight results in a higher mAP, which is a bit counterintuitive. It would be helpful to give some insight into this.
>
> A2: This behavior is not counterintuitive. With a fixed privacy weight, lowering the fairness weight $\lambda_{\text{fair}}$ allows the model to focus on majority or easier slices that dominate average precision, leading to higher mAP. Increasing $\lambda_{\text{fair}}$ reallocates model capacity toward harder or minority slices to reduce disparity, causing $|1-\mathrm{DI}|$ and $\Delta\mathrm{EOP}$ to decrease but typically reducing mAP, a standard fairness–utility trade-off (Donini et al., 2018). We clarified this explanation in Section 4 and referenced Table 2., which shows higher mAP at lower $\lambda_{\text{fair}}$ and improved fairness at higher $\lambda_{\text{fair}}$. We also highlighted the complementary effect for privacy: moderate $\lambda_{\text{priv}}$ stabilizes training, while excessively large values over-regularize representations, making features overly invariant and reducing both accuracy and group parity. Our best-performing configuration is $\lambda_{\text{fair}}=0.1$ and $\lambda_{\text{priv}}=0.01$.
>
> ---
>
> > Q3: Figure 2 has a lot of information, which may warrant a bit more detail in text descriptions. For example, why does the privacy attack success rate and fairness score dramatically increase in the fog situation as the intensity increases?
>
> A3: We expanded the explanation in both the text and Figure 2 to address this point. (1) In the Figure 2 caption, we clarified the direction of improvement for each metric (i.e., which values are better or worse) and specified that all curves show mean ± standard deviation across three random seeds. (2) In Section 4, we added a new subsection titled “Resilience Analysis under Adverse Conditions in CARLA,” explaining that dense fog suppresses visual edges and contrast, thereby lowering the signal-to-noise ratio and producing non-uniform visibility across demographic groups and object scales. As fog intensity increases, the model tends to rely on spurious shortcuts, which amplifies group disparity (reflected by increases in $|1-\mathrm{DI}|$ and $\Delta\mathrm{EOP}$) and makes membership or attribute inference easier, since training samples remain relatively more confident than unseen ones. This behavior is consistent with Figure 2: under clear conditions, RESFL achieves mAP $0.46$, fairness $0.24$, and privacy $0.17$; at $100\%$ fog, it maintains mAP $0.17$, fairness $0.50$, and privacy $0.44$, while other baselines degrade more severely.
>
> ---
>
> > Q4: Minor, it would be nice to add an arrow after each metric indicating the lower the better or the higher the better.
>
> A4: We added explicit “↑/↓” indicators for every metric across all result tables in the main paper, and we now report mean ± std over three seeds consistently.
>
>
> [1] Donini, Michele, et al. "Empirical risk minimization under fairness constraints." Advances in Neural Information Processing Systems 31 (2018).

---

### Author Response · Authors · 2025-11-21
**Thanks and overall remarks**

We thank all reviewers for their thoughtful and constructive remarks. We are encouraged by the overall reception and that all reviewers recognized the importance of jointly addressing privacy, fairness, and utility in federated learning. We appreciate the positive assessments of our contributions, including adversarial privacy disentanglement, evidential uncertainty modeling, and uncertainty-guided fairness-aware aggregation, and thank the reviewers for feedback that significantly strengthened the manuscript.

In response, we clarified the technical components, expanded several methodological explanations, reorganized key sections, and added new analyses that make the full RESFL pipeline easier to evaluate. Major updates include:

• **Expanded experimental configuration, transparency, and computational details (Section 4.1, Appendix F, Appendix G):**
  We now detail all hyperparameter grids and tuning policy, local validation splits used for UFM, non-IID data construction, DP-SGD settings, fine-tuning procedures, and gating rules for client-side validation confidence. We also added complexity discussion, clarified runtime conventions, and introduced comprehensive sensitivity analyses over aggregation temperature $\beta$, privacy and fairness weights, communication rounds $T$, local epochs $E$, learning-rate schedules, clipping bounds, and DP budgets $\epsilon$.


• **Clarified privacy framing and threat model (Section 1 and Appendix A.4):**
  We distinguish data locality from formal privacy guarantees, explicitly define the honest-but-curious server setting, detail the DP mechanism and accounting, and correct earlier wording that implied secure aggregation (which we do not use but noted as a direction for future work).

• **Sensitive-attribute availability and label-free variants (Section 4.5 and Appendix J.2–J.3):**
  We introduce label-free UFM variants (feature clustering, vacuity quantiles, and proxy cohorts) and add robustness results under missing or noisy sensitive labels, showing that RESFL retains most fairness-privacy benefits without demographic supervision.

• **Improved figures and presentation (Figures 1–2, captions, Section 4):**
  Figures are repositioned near their first mention, captions expanded, error bands added, and ↑/↓ indicators included for all metrics. Acronyms are introduced where first used.

We thank all reviewers once again for their insights, which helped improve both clarity and rigor of the work.

---

### Author Response · Authors · 2025-12-03
**Summary for Area Chair**

Dear Area Chair and Senior Area Chair,

We sincerely thank all reviewers for their constructive feedback and recognition of our work's contributions. To facilitate the meta-review process, we briefly summarize the changes made during the author–reviewer discussion in response to the reviewers’ comments.

---

**1. Experimental details, tuning, and non-IID setup**
- We fully specified the experimental protocol in **Section 4.1, Appendix F/G/H**:
  - Per-method hyperparameter grids and tuning (loss weights, LR, batch size, communication rounds, local epochs, clipping bounds), with a shared trial budget and a clear model selection rule (**R5t55, R8ZW3**).
  - Fixed **90/10 local validation splits** per client, stratified by task and sensitive cohorts, used only for UFM and validation metrics, never for testing or attacks (**R5t55**).
  - Detailed **Dirichlet non-IID construction**, including $\alpha$, resulting cohort proportions, and justification for equal client sizes (Section 4.2, Appendix H) (**R5t55, R8ZW3**).
  - Full **fine-tuning schedule** and rationale for the balanced CARLA evaluation set in **Appendix E.2** (**R8ZW3**).
- All main tables and Figure 2 now show **mean±std over 3 seeds** and ↑/↓ indicators (**R6gjm, R5t55**).

---

**2. Threat model, MIA/AIA, and corrected DP baselines**
- **Threat model**: **Appendix A.4** now explicitly states cross-silo FL with an honest-but-curious server observing cleartext updates; no secure aggregation is used (**R5t55**).
- **MIA/AIA**: We describe attack-reserve splits, white-box score-based MIA, and AIA on frozen features/updates in detail and now report **TPR@FPR and macro-F1** in **Appendix H**, not just success rates (**R5t55**).
- **DP baseline fix** (main concern from **R5t55**, including the “Brief comment on DP”):
  - Switched from batch-aggregate clipping to **per-example DP-SGD with Poisson subsampling and PRV accounting**, aligning the privacy unit with sample-level MIA (**Appendix F “Augmentations and DP clipping”, Appendix G “DP setup summary”**).
  - We now compute end-to-end **$\epsilon$** for given (clip norm $C$, sampling rate $q$, noise multiplier $\sigma$), and report **$(\epsilon, \delta, q, C, \text{steps})$** in tables.
  - **All DP baselines were recomputed** (main tables and Figure 2) with this corrected implementation.
  - Fine-tuning on private data also uses per-example DP-SGD and is composed in PRV; synthetic CARLA adaptation is excluded (Appendix E.2, **R5t55**).

---

**3. Theory: UFM and global fairness**
- To address the requested formal link between local UFM and global DI/EOP (**R5t55**), we added **Lemma B.3, Proposition B.4, Corollary B.5** in **Appendix B**.
- These show that controlling per-client UFM reduces an explicit upper bound on global group-parity gaps under mixture lifting and multi-round aggregation.
- **Section 3.1** now states this claim precisely and cites these results, replacing the earlier informal argument.

---

**4. Sensitive attributes and label-free UFM variants**
- Reviewers **Rqpfy** and **R8ZW3** highlighted reliance on sensitive labels. In response:
  - **Section 4.5** now discusses realistic domains where client-side sensitive labels exist and stay local, while the server only sees a clipped scalar UFM.
  - **Appendix J.2/J.3** introduce three **label-free UFM variants** (AF-UFM-Cluster, AF-UFM-Slices, AF-UFM-Proxy) that use feature clusters, evidential vacuity quantiles, or non-sensitive metadata.
  - On Adult and TweetEval, label-free variants retain **>90% of fairness/privacy gains** of labeled-S RESFL with **<2% mAP loss**, and show smooth degradation under missing/noisy labels.
  - **Section 5** now explicitly lists fully unsupervised and verifiable UFM as future work.

---

**5. Computational overhead and scalability**
- We clarified in **Section 4.1 and Appendix F/H** (**Rqpfy, R8ZW3**):
  - The evidential head **replaces** softmax, and the GRL adds only a sign inversion; the adversary is a small MLP removed at inference. This yields a **constant-factor** training overhead rather than a different asymptotic.
  - Communication adds only one clipped scalar UFM per client per round.
  - **8-client IID and non-IID results** (Appendix H, referenced in Section 4.2) confirm scaling beyond 4 clients.

---

**6. Framing, related work, and presentation fixes**
- We corrected earlier mentions of secure aggregation: the current implementation sends updates and UFMs in cleartext and lists secure aggregation only as future work (**R5t55**).
- Figures 1–2 are moved near first mention, use larger fonts, shaded error bands, and clear metric directions (**R6gjm**).
- The README in the code now clearly indicates which algorithms/utilities (e.g., CHFL) are *not* used in reported experiments (**R5t55**).

---

We once again thank you for your time and effort throughout the review process.

Best regards,

The Authors

---

### Meta-Review · Area_Chair_tCZH · 2026-01-07

**Summary:**

Reviewers (6gjm, qpfy, 8ZW3, 5t55) acknowledged the importance of RESFL's goal to jointly optimize privacy, fairness, and utility, but raised substantial concerns regarding experimental validity and deployment practicality. The most severe critique came from Reviewer 5t55, who identified a fundamental flaw in the Differential Privacy (DP) baseline implementation—specifically the use of batch-level rather than per-example clipping—and noted a lack of proper privacy accounting. Additionally, Reviewers qpfy and 8ZW3 questioned the real-world feasibility of the method due to its reliance on accessible sensitive attribute labels and the potential computational overhead for clients, while Reviewer 6gjm noted presentation issues and counterintuitive ablation results.

**Reviewer Concerns:**

The authors provided a thorough rebuttal that appears to address the primary technical and practical concerns. They re-implemented the DP baselines using per-example DP-SGD with Poisson subsampling and PRV accounting, seemingly resolving Reviewer 5t55’s comment regarding the privacy mechanisms. They also countered the "sensitive label" limitation by introducing label-free variants (using clustering or proxies) which they showed retain over 90% of the fairness/privacy gains, and added theoretical proofs linking local UFM to global fairness; as a result, there appear to be no major technical concerns remaining outstanding.

**Reviewer Scores:**

Reviewers qpfy and 8ZW3 (currently 6) would likely upgrade their scores to 8, as their main reservation regarding the practicality of sensitive attributes was directly addressed by the successful label-free experiments. Reviewer 5t55 (currently 2) may also increase their score had they confirmed the validity of the re-run DP baselines and the new privacy accounting, as their rejection was predicated on the now-corrected experimental flaws. Reviewer 6gjm (currently 6) would likely remain a 6, given that their presentation concerns were minor and addressed.

---

### Decision · Program_Chairs · 2026-01-26

Accept (Poster)